# Revenue Maximization Under Sequential Price Competition Via The Estimation Of $s$-Concave Demand Functions

**Daniele Bracale** [*], **Moulinath Banerjee & Yuekai Sun**
Department of Statistics
University of Michigan
Ann Arbor, MI, USA
{dbracale,moulib,yuekai}@umich.edu

**Cong Shi**
Department of Management
University of Miami
Miami, USA
congshi@bus.miami.edu

## Abstract

We consider price competition among multiple sellers over a selling horizon of $T$ periods. In each period, sellers simultaneously offer their prices (which are made public) and subsequently observe their respective demand (not made public). The demand function of each seller depends on all sellers' prices through a private, unknown, and nonlinear relationship. We propose a dynamic pricing policy that uses semi-parametric least-squares estimation and show that when the sellers employ our policy, their prices converge at a rate of $O(T^{-1/7})$ to the Nash equilibrium prices that sellers would reach if they were fully informed. Each seller incurs a regret of $O(T^{5/7})$ relative to a dynamic benchmark policy. A theoretical contribution of our work is proving the existence of equilibrium under shape-constrained demand functions via the concept of $s$-concavity and establishing regret bounds of our proposed policy. Technically, we also establish new concentration results for the least squares estimator under shape constraints. Our findings offer significant insights into dynamic competition-aware pricing and contribute to the broader study of non-parametric learning in strategic decision-making.

## 1 Introduction

Pricing plays a central role in competitive markets, where firms continuously adjust prices in response to demand fluctuations and rival strategies. A major challenge in competition-aware pricing lies in inferring rivals' pricing behavior from limited observations (Li et al., 2024). Firms cannot easily estimate price sensitivity through controlled experiments since competitors do not coordinate to hold prices constant while one firm tests different price values. Although existing sequential pricing algorithms yield low regret and converge toward a Nash Equilibrium (NE), they often rely on a linear demand framework (Kirman, 1975; Li et al., 2024), or nonlinear approaches restricted to a fixed parametric family (Goyal et al., 2023), limiting applicability. Because nonlinear demand better reflects reality (Gallego et al., 2006; Wan et al., 2022; 2023), we adopt a flexible semiparametric model with unknown parametric and nonparametric components.

Over $T$ periods, $N$ sellers set prices simultaneously; each seller's demand depends on both their own and rivals' prices, an effect amplified beyond linearity. *Sellers observe competitors' prices but not competitors' realized demand.* For analysis, we assume that all sellers use the same algorithm – a realistic simplification: for instance, Pri (2025) describes a pricing service used by numerous local hotels in Colorado. These hotels, while independently operated, utilize the same class of algorithmic tools provided by the platform to set their prices. Importantly, although they observe competitors' prices, they do not share underlying demand information with one another, as we assume in this work.

We design a tuning-free pricing policy purely based on shape constrained estimation that (i) maximizes each seller's revenue and (ii) guarantees convergence to the NE, the pricing configuration that would arise under full information (iii) attains sublinear regret against a dynamic benchmark, defined as the worst-case gap between a seller's average revenue under our proposed pricing policy (where neither

---

[*]Reference author

model parameters nor competitors' demands are known) and an optimal policy in hindsight (which assumes fixed competitor prices and full knowledge of the demand model).

**The Role of Shape Constraints in This Work.** In this paper, we assume that the mean demand for a seller $i \in [N] = \{1, 2, \ldots, N\}$, given the price vector $(p_i, \mathbf{p}_{-i})$ (where $\mathbf{p}_{-i}$ denotes the competitors' prices), follows a single index model of the form $\mathbb{E}[y_i | p_i, \mathbf{p}_{-i}] = \psi_i(-\beta_i p_i + \langle \boldsymbol{\gamma}_i, \mathbf{p}_{-i} \rangle)$, where the demand link function $\psi_i$ is assumed to be *both monotone and s-concave* ($\langle \cdot, \cdot \rangle$ denotes the euclidean inner product) and $\beta_i > 0$, which generalizes the linear demand setting in Li et al. (2024), where $\psi_i$ is linear (that is monotone and log-concave, i.e. $s$-concave with $s = 0$). Below, we summarize the motivation and relevance of these shape constraints, whose importance will become further evident throughout the paper.

Monotonicity aligns with economic intuition, where the standard form of demand is decreasing in the seller's own price (Li et al., 2024; Friesz et al., 2012; Kirman, 1975), and consequently, since $\beta_i > 0$, the functions $\psi_i$ must be increasing to preserve decreasing demand with respect to $p_i$. Indeed, in this case, $\partial_{p_i} \mathbb{E}[y_i | p_i, \mathbf{p}_{-i}] < 0$.

The $s$-concavity assumption of $\psi_i$ also arises naturally as a generalization of the commonly used log-concavity (recovered when $s = 0$), and plays a crucial role in guaranteeing convergence analysis towards equilibrium (see Section 3.1). We emphasize that $s$-concavity, being a higher-order smoothness constraint than monotonicity, drives the convergence rate results.

## 2  RELATED LITERATURE AND CONTRIBUTIONS

**Sequential Price Competition with Demand Learning.** Classic price competition with *known* demand goes back to Cournot (1838); Bertrand (1883), with later variants including multinomial logit (Gallego et al., 2006; Aksoy-Pierson et al., 2013; Gallego & Wang, 2014), fixed-point methods for mixed logit (Morrow & Skerlos, 2011), and multi-epoch competition (Gallego & Hu, 2014; Federgruen & Hu, 2015; Chen & Chen, 2021). However, over the last two decades, *demand learning* for competitive pricing has advanced substantially.

In this paper, we study sequential price competition with demand learning. Early work (Kirman, 1975) analyzed symmetric duopolies with linear demand. Li et al. (2024) achieved optimal $\sqrt{T}$ regret for asymmetric linear demand, $\lambda_i(\mathbf{p}) = \alpha_i - \beta_i p_i + \sum_{j \neq i} \gamma_{ij} p_j$, with unknown sensitivities. However, nonlinear demand is more realistic and widely used (Gallego et al., 2006; Wan et al., 2022; 2023). For instance, Gallego et al. (2006) study $\lambda_i(\mathbf{p}) = a_i(p_i) / \left( \sum_j a_j(p_j) + \kappa \right)$ for *known* increasing $a_i$ and $\kappa \in (0, 1]$. In contrast, we analyze an *unknown* monotone single-index model

$$\lambda_i(\mathbf{p}) = \psi_i(-\beta_i p_i + \sum_{j \neq i} \gamma_{ij} p_j) = \psi_i(\langle \boldsymbol{\theta}_i, \mathbf{p} \rangle), \tag{1}$$

where $\boldsymbol{\theta}_i$ has $-\beta_i$ in coordinate $i$ and $\gamma_{ij}$ elsewhere, and both $\boldsymbol{\theta}_i$ and $\psi_i$ are unknown. This strictly generalizes Li et al. (2024): setting $\psi_i(u) = u + \alpha_i$ recovers their linear model.

**Monotone Single-Index Models.** By Equation (1), we have $N$ single-index models $\mathbb{E}(y_i | \mathbf{p}) = \psi_i(\langle \boldsymbol{\theta}_i, \mathbf{p} \rangle)$, $i \in \mathcal{N} = \{1, 2, \ldots, N\}$, with unknown $\boldsymbol{\theta}_i \in \mathbb{R}^N$ and unknown links $\psi_i$. We assume $\psi_i$ is nondecreasing, which makes demand $\lambda_i$ nonincreasing in its own price $p_i$, a standard assumption in the economics literature (Birge et al., 2024; Li et al., 2024). Monotonicity also permits fully data-driven, tuning-free nonparametric estimators (Balabdaoui et al., 2019). The monotone SIM is identifiable under mild conditions: $\|\boldsymbol{\theta}_i\|_2 = 1$, and $\mathbf{p}$ has a strictly positive density on its domain (see Balabdaoui et al. (2019, Prop. 5.1)). Balabdaoui et al. (2019) also propose a joint estimator $(\widehat{\boldsymbol{\theta}}_{i,n}, \widehat{\psi}_{i,n})$ with $L^2$ error of order $n^{-1/3}$ for $\widehat{\psi}_i(\langle \widehat{\boldsymbol{\theta}}_i, \cdot \rangle)$, and study the normalized linear estimator

$$\widetilde{\boldsymbol{\theta}}_{i,n} = \frac{\widehat{\boldsymbol{\theta}}_{i,n}}{\|\widehat{\boldsymbol{\theta}}_{i,n}\|_2}, \quad \widehat{\boldsymbol{\theta}}_{i,n} \in \arg\min_{\boldsymbol{\theta} \in \mathbb{R}^N} \sum_{t=1}^n \left( y_i^{(t)} - \langle \boldsymbol{\theta}, \mathbf{p}^{(t)} - \bar{\mathbf{p}} \rangle \right)^2, \quad \bar{\mathbf{p}} = \frac{1}{n} \sum_{t=1}^n \mathbf{p}^{(t)}, \tag{2}$$

which remains consistent under elliptically symmetric $\mathbf{p}^{(t)}$ (Brillinger, 2012). In this work, we also assume that $\mathbf{p}^{(t)}$ follows an elliptically symmetric distribution during the exploration phase. However, for our purpose, an $L^2$ convergence alone for $\widehat{\psi}_i(\langle \widehat{\boldsymbol{\theta}}_i, \cdot \rangle)$ is insufficient to establish a regret bound, but a uniform (supremum-norm) rate suffices and is derived in this paper.

**Connection with NE, virtual valuation, and $s$-concavity.** We propose a sequential price competition online algorithm that provides sublinear regret. The first step to achieve this goal is to establish

the existence and uniqueness of a Nash Equilibrium $\mathbf{p}^*$, which is the fixed point of an operator $\mathbf{\Gamma}$ (see Section 3.1 for more details). A necessary condition for its existence is that each seller's *virtual valuation*

$$\varphi_i(u) = u + \psi_i(u)/\psi_i'(u) \tag{3}$$

is increasing with derivative bounded below: $\varphi_i'(u) \geq c_i$ for some $c_i > 0$ for all $i \in \mathcal{N}$. This mirrors the $N = 1$ literature (monopolistic setting) such as Fan et al. (2024); Chen & Farias (2018); Cole & Roughgarden (2014); Golrezaei et al. (2019); Javanmard (2017); Javanmard & Nazerzadeh (2019), which often assume log-concavity of $\psi_i$ and $1 - \psi_i$ (implying $c_i \geq 1$).

Our key observation *links the NE and shape constraints* as follows: $\varphi_i'(u) \geq c_i$ iff $\psi_i$ is $(c_i - 1)$-concave (Proposition 3.5; see the section below for definition of $s$-concavity), with log-concavity recovered as a special case when $c_i = 1$. This connection allows us to estimate $\psi_i$ via nonparametric least squares under $s$-concavity – yielding a fully data-driven, tuning-free estimation procedure.

$s$**-Concavity.** A technical contribution of this work is the study of the uniform rate of convergence of the least-squares estimator of a unidimensional $s$-concave regression function. As defined in Han & Wellner (2016), a unidimensional function $\psi : \mathcal{U} \to (0, \infty), \mathcal{U} \subset \mathbb{R}$, is said to be $s$-concave for some $s \in \mathbb{R}$, and we write $\psi \in \mathcal{F}_s(\mathcal{U})$ if $\psi((1 - \lambda)u_0 + \lambda u_1) \geq M_s(\psi(u_0), \psi(u_1); \lambda)$, for all $u_0, u_1 \in \mathcal{U}$ and $\lambda \in (0, 1)$, where

$$M_s(y_0, y_1; \lambda) \triangleq \begin{cases} ((1 - \lambda)y_0^s + \lambda y_1^s)^{1/s}, & s \neq 0, y_0, y_1 > 0 \\ 0, & s < 0, y_0 = y_1 = 0 \\ y_0^{1-\lambda} y_1^\lambda, & s = 0. \end{cases}$$

This notion generalizes concavity ($s = 1$) and log-concavity (which holds for $s = 0$, in the sense that $\lim_{s \to 0} M_s(y_0, y_1; \lambda) = M_0(y_0, y_1; \lambda)$). These classes are nested since $\mathcal{F}_s(\mathcal{U}) \subset \mathcal{F}_0(\mathcal{U}) \subset \mathcal{F}_r(\mathcal{U})$, if $-\infty < r < 0 < s < \infty$. The class of log-concave densities has been extensively studied: see Bobkov & Madiman (2011); Dümbgen & Rufibach (2009); Cule & Samworth (2010); Borzadaran & Borzadaran (2011); Bagnoli & Bergstrom (2006); while Han & Wellner (2016); Doss & Wellner (2016); Chandrasekaran et al. (2009); Koenker & Mizera (2010) deal with general $s$-concavity. It is easy to see that such functions $\psi$ have the form $\psi = (\phi_+)^{1/s}$ for some concave function $\phi$ if $s > 0$, where $x_+ = \max\{0, x\}, \psi = e^\phi$ for some concave function $\phi$ if $s = 0$, and $\psi = (\phi_+)^{1/s}$ for some convex function $\phi$ if $s < 0$. Then, $\psi$ has the following representation: $\psi = h_s \circ \phi$ where $\phi : \mathcal{U} \to \mathbb{R}$ is concave and

$$h_s(x) = d_s^{-1}(x) = \begin{cases} e^x, & s = 0, x \in \mathbb{R}, \\ (-x)^{1/s}, & s < 0, x < 0, \\ x^{1/s}, & s > 0, x > 0, \end{cases} \qquad d_s(y) = \begin{cases} \log(y), & s = 0, y > 0, \\ -y^s, & s < 0, y > 0, \\ y^s, & s > 0, y > 0. \end{cases} \tag{4}$$

Different from previous results, we study the convergence rate for the more general class $\mathcal{F}_h$ (see Appendix H for details) that contains functions of the form $\psi = h \circ \phi : \mathcal{U} \to \mathbb{R}$, where $h : \mathbb{R} \to (0, \infty)$ is a *known* increasing function in $\mathrm{C}^2$ and $\phi : \mathcal{U} \to \mathbb{R}$ is an *unknown* concave function. Specifically, we consider i.i.d. observations $(U_i, Y_i) \sim (U, Y)$ for $i = 1, 2, \ldots, n$ where $Y_1, \ldots, Y_n$ are noisy representations of the mean function $\psi_0(u) = \mathbb{E}(Y|U = u) = h(\phi_0(u))$ and $U_1 \leq U_2 \leq \cdots \leq U_n$ are contained in $\mathcal{U}$. We study the uniform convergence rate of $\widehat{\psi}_n = h \circ \widehat{\phi}_n$, where $\widehat{\phi}_n$ is the LSE

$$\widehat{\phi}_n \in \operatorname{argmin}_{\phi \text{ concave}} \sum_{i=1}^n (Y_i - h \circ \phi(U_i))^2.$$

The uniform convergence of $\widehat{\psi}_n$ to $\psi_0$ plays a crucial role in establishing the convergence of our proposed pricing strategy to the Nash Equilibrium, as we will see later in the paper.

## 2.1 SUMMARY OF KEY CONTRIBUTIONS

We now provide a summary of our key contributions.

**A novel semiparametric pricing policy for nonlinear mean demand.** We extend the standard approach to estimating the mean demand function of a firm $\lambda_i(\cdot)$ by introducing a monotone single index model $\lambda_i(\mathbf{p}) = \psi_i(\langle \boldsymbol{\theta}_i, \mathbf{p} \rangle)$, where $\psi_i$ is increasing and $s_i$-concave for some $s_i > -1$, providing substantially more flexibility than previous parametric models (Li et al., 2024; Kachani et al., 2007; Gallego et al., 2006).

$s$**-concave mean demand functions.** In many existing works, necessary conditions for the existence of the NE are derived by assuming that the virtual valuation function defined in Equation (3) is increasing, often by invoking log-concavity of the mean demand function. We show how all such assumptions can be cast under the more general framework of $s$-concavity. This reformulation allows the development of a fully data-driven, tuning-parameter-free algorithm using shape constraints.

**Regret upper bound and convergence to equilibrium.** We establish an upper bound on the total expected regret and analyze the convergence to the NE for a general exploration length of order $\tau \propto T^{\xi}$ for $\xi \in (0,1)$ (Theorem 5.4). Our results reveal the existence of an optimal choice of $\xi$ that minimizes the total expected regret for each seller, leading to a regret of order $\widetilde{\mathcal{O}}(N^{3/2}T^{5/7})$, where $\widetilde{\mathcal{O}}$ excludes log factors (Remark 5.6). Moreover, we show that by the end of the selling horizon, the joint prices set by sellers converge to Nash equilibrium prices at a rate of $\widetilde{\mathcal{O}}(N^{3/4}T^{-1/7})$.

**Concentration inequality for $s$-concave regression functions in the supremum norm.** Our work involves establishing a concentration inequality for the nonparametric LSE, under the supremum norm, for a large class of shape constraints that includes $s$-concavity (Appendix H). As a minor contribution, we derive a concentration inequality for the parametric component $\boldsymbol{\theta}_i$ of the monotone single index model (Proposition D.3), while previous results show convergence in probability or distribution (Balabdaoui et al., 2019).

**General Notation.** We use $\|\cdot\|_1, \|\cdot\|_2$ (or $\|\cdot\|$),$\|\cdot\|_{\infty}$ for the $L^1$, Euclidean, and sup norms, respectively, and $\langle \mathbf{u}, \mathbf{v} \rangle = \mathbf{u}^{\top}\mathbf{v}$ for the inner product. The unit sphere in $\mathbb{R}^N$ is $\mathbb{S}_{N-1} = \{\mathbf{x} \in \mathbb{R}^N : \|\mathbf{x}\|_2 = 1\}$. We write $\tilde{O}(\cdot)$ to suppress logarithmic factors and use $\lesssim$ to hide absolute constants. For functions, $\mathrm{C}^m(\Omega)$ denotes $m$-times continuously differentiable $f : \Omega \to \mathbb{R}$. A Lipschitz function $f$ has constant $L_f > 0$ with respect to prespecified norms. $f$ is a contraction if $L_f < 1$. If $f \in \mathrm{C}^2(\Omega)$, $\Omega \subseteq \mathbb{R}$, it is $\mu$-strongly convex if $f''(x) \geq \mu > 0$. Finally, following Delmas et al. (2024, Pag. 86), a random vector $Z \sim \mathscr{E}(\mathbf{m}, \Lambda, g)$ has elliptically symmetric distribution with location $\mathbf{m}$, scale matrix $\Lambda \succ 0$ (positive definite), and density generator $g : [0, \infty) \to [0, \infty)$, if $f_Z(\mathbf{z}) \propto g\left((\mathbf{z} - \mathbf{m})^{\top}\Lambda^{-1}(\mathbf{z} - \mathbf{m})\right)$.

## 3 PROBLEM FORMULATION

We adopt a problem setup similar to that of Li et al. (2024). We consider $N$ sellers, each selling a single type of product with unlimited inventories over a selling horizon of $T$ periods. We use $t \in \mathcal{T} \triangleq \{1, 2, \ldots, T\}$ to index time periods and $i \in \mathcal{N} \triangleq \{1, 2, \ldots, N\}$ to index sellers. At the beginning of each period, each seller *simultaneously* selects their price. For seller $i, p_i^{(t)} \in \mathcal{P}_i \triangleq [\underline{p}_i, \bar{p}_i]$, denotes the price that seller $i$ offers in period $t$, with price bounds $\underline{p}_i < \bar{p}_i$ and $\underline{p}_i, \bar{p}_i \in [0, +\infty)$. Let $\mathbf{p}_{-i}^{(t)} \triangleq (p_j^{(t)})_{j \in \mathcal{N} \setminus \{i\}}$ denote the competitor prices at time $t$, $\mathbf{p}^{(t)} \triangleq (p_j^{(t)})_{j \in \mathcal{N}}$ denote the joint prices' vector, and let $\mathcal{P} \triangleq \prod_{i \in \mathcal{N}} [\underline{p}_i, \bar{p}_i]$. *The vector $\mathbf{p}^{(t)}$ is made public at time $t$, then observed by all the sellers.* A common knowledge is also the set

$$\mathcal{U} \triangleq [-p_{\max}, p_{\max}], \quad p_{\max} = \left(\sum_{i \in \mathcal{N}} \bar{p}_i^2\right)^{1/2}. \tag{5}$$

*The demand $y_i^{(t)}$ of seller $i$ in period $t$ is observed by seller $i$ and is kept private, i.e., not shared among competitors.* The individual demand $y_i^{(t)}$ depends on the offered prices of all sellers, $\mathbf{p}^{(t)} \in \mathcal{P}$, and follows a nonlinear model:

$$y_i^{(t)} = \lambda_i(\mathbf{p}^{(t)}) + \varepsilon_i^{(t)}, \quad \lambda_i(\mathbf{p}^{(t)}) \triangleq \psi_i(-\beta_i p_i^{(t)} + \langle \boldsymbol{\gamma}_i, \mathbf{p}_{-i}^{(t)} \rangle) = \psi_i(\langle \boldsymbol{\theta}_i, \mathbf{p}^{(t)} \rangle), \quad t \in \mathcal{T}, \tag{6}$$

where $\{\varepsilon_i^{(t)}\}_{t \in \mathcal{T}}$ are (zero mean) $\sigma_i$-sub-gaussian demand noises following independent and identical distributions ($\varepsilon_i^{(t)}$ and $\varepsilon_j^{(t)}$ can be correlated with $i \neq j$, $i, j \in \mathcal{N}$), and $\boldsymbol{\theta}_i$ is an unknown vector of dimension $N$ with $i$-th entry equal to $-\beta_i$ and the remaining entries being the values of $\boldsymbol{\gamma}_i \in \mathbb{R}^{N-1}$, ordered. The parameter vector $\boldsymbol{\gamma}_i$ measures how seller $i$'s demand is affected by competitor prices. We assume that the parameter space is such that the average demand $\lambda_i$ is non-negative and $\partial_{p_i}\lambda_i < 0$ among all values of $\{\boldsymbol{\theta}_i, \psi_i\}_{i \in \mathcal{N}}$; a similar assumption is found in Birge et al. (2024); Li et al. (2024). The above conditions hold if $\psi_i, \psi_i' > 0$ and $\beta_i > 0$, which are explicitly assumed in Assumption 3.1.

**Assumption 3.1.** *For every $i \in \mathcal{N}$, $\psi_i : \mathcal{U} \to [\underline{B}_{\psi_i}, \bar{B}_{\psi_i}]$, where: $\psi_i \in \mathrm{C}^2(\mathcal{U})$ is unknown; $0 < \underline{B}_{\psi_i} < \bar{B}_{\psi_i} < \infty$ are known; $0 < \underline{B}_{\psi_i'} \leq \psi_i' \leq \bar{B}_{\psi_i'}$ and $|\psi_i''| \leq B_{\psi_i''}$, where $\underline{B}_{\psi_i'}, \bar{B}_{\psi_i'}, B_{\psi_i''} > 0$*

*are not necessarily known. We also assume*

$$\beta_i \geq \underline{\beta}_i, \quad \|\boldsymbol{\theta}_i\|_2^2 = \beta_i^2 + \|\boldsymbol{\gamma}_i\|_2^2 = 1, \quad i.e. \quad \boldsymbol{\theta}_i \in \mathbb{S}_{N-1},$$

*where $\underline{\beta}_i \in (0, 1]$ is an unknown constant.*

The constants $\underline{B}_{\psi_i}, \bar{B}_{\psi_i}$ are known by the seller $i$ (they are used to compute the optimization problem in (11)), however, firms can estimate them easily from historical sales data and operational capacity limits; moreover loose bounds suffice and do not affect the rates in Theorem 5.4. The condition $\|\boldsymbol{\theta}_i\|_2 = 1$ guarantees the differentiability of the monotone single index model (Chmielewski, 1981; Balabdaoui et al., 2019). Note that Equation (6) is well defined, that is $\langle \boldsymbol{\theta}_i, \mathbf{p}^{(t)} \rangle \in \mathcal{U}$, indeed $\|\boldsymbol{\theta}_i\|_2 = 1 \implies |\langle \boldsymbol{\theta}_i, \mathbf{p} \rangle| \leq \|\mathbf{p}\|_2 \leq p_{\max}, \forall \mathbf{p} \in \mathcal{P}$. Regarding $\boldsymbol{\gamma}_i$, while many applications assume $\gamma_{ij} > 0$, this restriction is not required for our algorithm or theoretical guarantees - our results remain valid when $\gamma_{ij} > 0$. We allow $\gamma_{ij}$ to be negative to capture settings with negative cross-effects, such as vertically differentiated products, where raising a competitor's price may reduce my demand if consumers perceive their product as higher quality.

**Individual Regret.** Seller $i \in \mathcal{N}$ aims to design a policy $\{p_i^{(t)}\}_{t \in \mathcal{T}}$ that maximizes their individual (cumulative) revenue

$$\mathrm{R}_i(T) \triangleq \mathbb{E} \sum_{t=1}^{T} p_i^{(t)} y_i^{(t)} = \mathbb{E} \sum_{t=1}^{T} \mathrm{rev}_i(p_i^{(t)} \mid \mathbf{p}_{-i}^{(t)}), \quad \mathrm{rev}_i(p_i | \mathbf{p}_{-i}) \triangleq p_i \psi_i(-\beta_i p_i + \langle \boldsymbol{\gamma}_i, \mathbf{p}_{-i} \rangle).$$

Maximizing a seller's revenue can be reframed as minimizing their regret. Each seller competes with a dynamic optimal sequence of prices in hindsight while assuming that the other sellers would not have responded differently if this sequence of prices had been offered. Under such a dynamic benchmark, the objective of each seller $i \in \mathcal{N}$ is to minimize the following regret metric in hindsight:

$$\mathrm{Reg}_i(T) \triangleq \left[ \mathbb{E} \sum_{t=1}^{T} \mathrm{rev}_i(\Gamma_i(\mathbf{p}_{-i}^{(t)}) \mid \mathbf{p}_{-i}^{(t)}) \right] - \mathrm{R}_i(T), \ \Gamma_i(\mathbf{p}_{-i}) \in \underset{p_i \in \mathcal{P}_i}{\mathrm{argmax}} \ \mathrm{rev}_i(p_i \mid \mathbf{p}_{-i}), \quad (7)$$

where $\Gamma_i : \mathcal{P}_{-i} \to \mathcal{P}_i$ is denoted as the $i$-th seller's *Best Response Map*, and $\mathbf{p}_{-i} \triangleq (p_j)_{j \in \mathcal{N} \setminus \{i\}}$.

**Nash Equilibrium (NE).** A Nash equilibrium $\mathbf{p}^* = (p_i^*)_{i \in \mathcal{N}} \in \mathcal{P}$ is defined as a price vector under which unilateral deviation is not profitable for any seller. Specifically,

$$\mathrm{rev}_i(p_i^* \mid \mathbf{p}_{-i}^*) \geq \mathrm{rev}_i(p_i \mid \mathbf{p}_{-i}^*), \quad \forall p_i \in \mathcal{P}_i, \quad \forall i \in [N],$$

or, equivalently, $\mathbf{p}^*$ is a solution to the following fixed point equation:

$$\mathbf{p}^* = \boldsymbol{\Gamma}(\mathbf{p}^*) = (\Gamma_1(\mathbf{p}_{-1}^*), \dots, \Gamma_N(\mathbf{p}_{-N}^*)), \quad \boldsymbol{\Gamma} : \mathcal{P} \to \mathcal{P}, \quad (8)$$

where $\boldsymbol{\Gamma}$ is called *Best Response Operator*, and its components $\Gamma_i$ are defined in (7).

## 3.1 MAIN ASSUMPTIONS

Before presenting the main assumptions, we clarify why our framework requires two key properties: the best-response map must be contractive, and each seller's revenue function must be strongly concave. These conditions ensure well-posedness of the equilibrium and enable the stability and convergence guarantees developed later.

The objective of this paper is to design an algorithm that guarantees sublinear regret for every seller, i.e., $\mathrm{Reg}_i(T) = o(T)$ for every $i \in \mathcal{N}$. Our policy consists of two phases: an exploration phase, in which each seller learns their individual best-response map $\hat{\Gamma}_i$ by consistently estimating the parameters $(\boldsymbol{\theta}_i, \psi_i)$, and an exploitation phase, in which, at each round $t$, seller $i$ sets their price according to $p_i^{(t)} = \hat{\Gamma}_i(\mathbf{p}_{-i}^{(t-1)})$, or equivalently $\mathbf{p}^{(t)} = \hat{\boldsymbol{\Gamma}}(\mathbf{p}^{(t-1)})$. The link between regret and equilibrium is straightforward: the individual regret is controlled if the iterates converge to the NE. In fact, $\mathrm{Reg}_i(T) \lesssim T\mathbb{E}\|\mathbf{p}^{(T)} - \mathbf{p}^*\|_2^2$, were $\mathbf{p}^\star$ is a NE (see (37) for a detailed bound). By the triangle inequality, we have

$$\|\mathbf{p}^{(t)} - \mathbf{p}^\star\| \leq \|\hat{\boldsymbol{\Gamma}}(\mathbf{p}^{(t-1)}) - \boldsymbol{\Gamma}(\mathbf{p}^{(t-1)})\| + \|\boldsymbol{\Gamma}(\mathbf{p}^{(t-1)}) - \boldsymbol{\Gamma}(\mathbf{p}^\star)\| \leq \|\hat{\boldsymbol{\Gamma}} - \boldsymbol{\Gamma}\|_\infty + L_{\boldsymbol{\Gamma}}\|\mathbf{p}^{(t-1)} - \mathbf{p}^\star\|,$$

where $L_{\boldsymbol{\Gamma}}$ is the Lipschitz constant of $\boldsymbol{\Gamma}$ and $\|\mathbf{F}\|_\infty = \sup_{\mathbf{p} \in \mathcal{P}} \|\mathbf{F}(\mathbf{p})\|$. Therefore, convergence to $\mathbf{p}^\star$ requires two ingredients: $(i)$ **Contraction**: $0 < L_{\boldsymbol{\Gamma}} < 1$ which ensures that deviations from the

NE shrink geometrically; $(ii)$ **Consistent estimation for** $\|\hat{\Gamma} - \Gamma\|_\infty$, which follows from consistent estimation of $(\boldsymbol{\theta}_i, \psi_i)$ and strong concavity of $\mathrm{rev}_i(\cdot \mid \mathbf{p}_{-i})$ uniformly in $\mathbf{p}_{-i}$, for each seller $i \in \mathcal{N}$. Putting these together, we obtain (see (39) for a formal inequality):

$$\mathbb{E}\|\mathbf{p}^{(T)} - \mathbf{p}^\star\| \lesssim \underbrace{L_{\boldsymbol{\Gamma}}^{(T-1)}\mathbb{E}\|\mathbf{p}^{(0)} - \mathbf{p}^\star\|}_{\text{shrinks under contraction}} + \underbrace{\text{estimation error } \mathbb{E}\|\boldsymbol{\Gamma} - \hat{\boldsymbol{\Gamma}}\|_\infty}_{\text{vanishes under strong concavity}} \quad \text{as } T \to \infty. \quad (9)$$

Additionally, strong concavity ensures the existence of the NE, while the contraction property of the best response map implies the uniqueness of the NE.

**Assumption 3.2.** *For every $i \in \mathcal{N}$, $\mathrm{rev}_i(\cdot \mid \mathbf{p}_{-i})$ is strongly concave in $\mathcal{P}_i$, uniformly on $\mathbf{p}_{-i} \in \mathcal{P}_{-i}$[1].*

By twice differentiability of $\psi_i$, Assumption 3.2 is guaranteed if $\mu_i \triangleq 2\beta_i \underline{B}_{\psi_i'} - \beta_i^2 \bar{p}_i B_{\psi_i''} > 0$. Indeed, since by Assumption 3.1 $|\psi_i''| \leq B_{\psi_i''}$ and $\psi_i' \geq B_{\psi_i'}$, we have

$$-\partial_p^2 \mathrm{rev}_i(p \mid \mathbf{p}_{-i}) \geq 2\beta_i \underline{B}_{\psi_i'} - \beta_i^2 \bar{p}_i B_{\psi_i''} := \mu_i.$$

Assumption 3.2 guarantees that the best-response map $\Gamma_i$ in (7) is well defined for every $\mathbf{p}_{-i}$. More importantly, it guarantees the existence of the Nash equilibrium, in line with standard results in competitive games (Scutari et al., 2014; Li et al., 2024; Tsekrekos & Yannacopoulos, 2024). Instances where Assumption 3.2 is satisfied include linear demand models (Li et al., 2024), concave demand specifications, and a class of $s_i$-concave demand functions with $s_i > -1$. A more detailed discussion of these and further examples is provided in Appendix I.1. The next result is an immediate consequence of Theorem 3, in Scutari et al. (2014).

**Lemma 3.3** (Existence of NE). *Under Assumptions (3.1) and (3.2), there exists a $\mathbf{p}^* \in \mathcal{P}$ satisfying the fixed point equation in (8).*

By Assumption 3.2, the map $\Gamma_i(\mathbf{p}_{-i})$ can be recovered by solving the first order conditions (FOCs), $\partial_{p_i} \mathrm{rev}_i(p_i \mid \mathbf{p}_{-i}) = 0$, and projecting the solution onto $\mathcal{P}_i$. We now determine a shape constraint assumption that is sufficient to satisfy the FOCs. Specifically, we can show that $\Gamma_i$ can be written as (see the proof of Lemma 3.7 for the derivation)

$$\Gamma_i(\mathbf{p}_{-i}) = \Pi_{\mathcal{P}_i} g_i(\langle \boldsymbol{\gamma}_i, \mathbf{p}_{-i} \rangle)/\beta_i, \quad g_i(u) \triangleq u - \varphi_i^{-1}(u), \quad \varphi_i(u) \triangleq u + \psi_i(u)/\psi_i'(u),$$

provided $\varphi_i^{-1}$ exists. Here $\Pi_{\mathcal{P}_i}$ is the projection into $\mathcal{P}_i$ and $\varphi_i$ is called *virtual valuation function* of firm $i$. For this purpose, we introduce Assumption 3.4, which makes the $\varphi_i$'s invertible.

**Assumption 3.4.** *$\forall i \in \mathcal{N}$, there exists a constant $c_i > 0$, known to seller $i$, such that $\varphi_i' \geq c_i$.*

Assumption 3.4 covers the linear demand model by Li et al. (2024) for which $\psi_i(u) = \alpha_i + u \implies \varphi_i(u) = 2u + \alpha_i \implies \psi_i' \geq 2$ and can also be found in several works with $N = 1$ (monopolistic setting); for example, Assumption 2.1 in Fan et al. (2024), Assumption 1 in Chen & Farias (2018), Equation 1 in Cole & Roughgarden (2014) and Golrezaei et al. (2019); Javanmard (2017); Javanmard & Nazerzadeh (2019) which assume $\psi_i$ and $1 - \psi_i$ to be log-concave (which specifically implies $c_i \geq 1$), where $\psi_i$ is a survival function.

We now present an equivalent formulation of Assumption 3.4 in terms of $s$-concavity, a condition that will allow us to estimate $\psi_i$ via shape constraints (the proof is provided in Appendix B). Examples of $s$-concave functions can be found in Appendix L.

**Proposition 3.5.** *For every $i \in \mathcal{N}$, $\varphi_i' \geq c_i$ if and only if $\psi_i$ is $(c_i - 1)$-concave.*

We now establish a sufficient condition under which the operator $\boldsymbol{\Gamma} = (\Gamma_i)_{i \in [N]}$ is a contraction in $\mathcal{P}$.

**Assumption 3.6.** *$\sup_{i \in \mathcal{N}} \|g_i'\|_\infty \|\boldsymbol{\gamma}_i\|_1/\beta_i < 1$.*

Assumption 3.6 generalizes Assumption 2 in Li et al. (2024) (in their case $\psi_i(u) = u + \alpha_i \implies g_i(u) = (u + \alpha_i)/2 \implies g_i' = 1/2$, matching the contraction constant $L_{\boldsymbol{\Gamma}}$ in Equation 30 of their work). Similar assumptions are found in Kachani et al. (2007). The proof of the following result is relegated to Appendix C. In practice, Assumption 3.6 states that the influence of competitors' prices on seller $i$'s optimal response is sufficiently small relative to the sensitivity to its own price.

---

[1] This means that there exists $\xi_i > 0$ independent of $\mathbf{p}_{-i}$ such that for all $x, y \in \mathcal{P}_i$ and all $\mathbf{p}_{-i} \in \mathcal{P}_{-i}$, $\mathrm{rev}_i(y \mid \mathbf{p}_{-i}) - \mathrm{rev}_i(x \mid \mathbf{p}_{-i}) \leq \partial_x \mathrm{rev}_i(x \mid \mathbf{p}_{-i})(y - x) - \frac{\xi_i}{2}\|y - x\|^2$.

**Lemma 3.7.** *Let Assumptions 3.2, 3.4 and 3.6 hold. Then,* $\mathbf{\Gamma} = (\Gamma_i)_{i \in [N]} : \mathcal{P} \to \mathcal{P}$ *is a contraction with contraction constant* $L_{\mathbf{\Gamma}} := \sup_{i \in \mathcal{N}} \|g_i'\|_\infty \|\boldsymbol{\gamma}_i\|_1 / \beta_i < 1$. *Consequently, the NE* $\mathbf{p}^*$ *is unique.*

**Remark 3.8.** *In the specific case where the* $\psi_i s$ *are* $s_i$*-concave for some* $s_i > -1/2$ *(i.e.* $c_i > 1/2$, *which happens for example if the* $\psi_i$ *are log-concave, i.e.* $c_i = 1$, *or concave, i.e.* $c_i = 2$*), then*

$$g_i'(u) = 1 - 1/\varphi_i'(\varphi_i^{-1}(u)) \geq 1 - 1/c_i > -1, \quad g_i'(u) = 1 - 1/\varphi_i'(\varphi_i^{-1}(u)) < 1, \quad \Rightarrow \|g_i'\|_\infty < 1.$$

*In this case Assumption 3.6 reduces to* $\beta_i > \|\boldsymbol{\gamma}_i\|_1$, *but since* $\beta_i = (1 - \|\boldsymbol{\gamma}_i\|_2^2)^{1/2}$, *the condition becomes* $\|\boldsymbol{\gamma}_i\|_1^2 + \|\boldsymbol{\gamma}_i\|_2^2 < 1$. *But since* $\|\boldsymbol{\gamma}_i\|_2 \leq \|\boldsymbol{\gamma}_i\|_1$, *a sufficient condition is* $\|\boldsymbol{\gamma}_i\|_1 < 1/\sqrt{2}$.

# 4 PROPOSED ALGORITHM

Our Algorithm 1 consists of an initial exploration phase for parameter estimation, followed by an exploitation phase where sellers set prices based on the learned demand model. In the exploration phase, each seller $i \in \mathcal{N}$ samples prices $p_i^{(t)}$ from a distribution $\mathscr{D}_i$ for $t = 1, 2, \ldots, \tau$. The common exploration length is $\tau \propto T^\xi$ for some $\xi \in (0, 1)$, to be specified later (Remark 5.6).

**Remark 4.1** (The common exploration phase). *See Appendix I.2 for a clear discussion of why sellers have no incentive to alter the exploration phase.*

The random prices $p_i^{(t)}$ charged by different sellers in the exploration phase are not necessarily independent, and we will use $\mathscr{D}$ from now on to denote the joint distribution of the price vector in the exploration phase. Recall that at every time $t$, *each firm* $i$ *observes their own (random) demand value* $y_i^{(t)}$ *while the prices* $\mathbf{p}^{(t)} = (p_1^{(t)}, p_2^{(t)}, \ldots, p_N^{(t)})^\top$ *are made public.* At the end of the exploration phase, firm $i$ estimates $(\boldsymbol{\theta}_i, \psi_i)$ using data $\{(\mathbf{p}^{(t)}, y_i^{(t)})\}_{t \leq \tau}$. More precisely, each firm $i$ chooses a proportion $\kappa_i$ of initial data points in the exploration phase $\mathcal{T}_i^{(1)} = \{1, 2, \ldots, \kappa_i \tau\}$ to estimate $\boldsymbol{\theta}_i$, and subsequent time points $\mathcal{T}_i^{(2)} = \{\kappa_i \tau + 1, \ldots, \tau\}$ to estimate $\psi_i$. For simplicity of notation we define $n_i^{(1)} = |\mathcal{T}_i^{(1)}| = \tau \kappa_i$ and $n_i^{(2)} = |\mathcal{T}_i^{(2)}| = \tau(1 - \kappa_i)$.

**Remark 4.2** (Need for two different phases for model estimation). *Owing to space constraints, we defer to Appendix I.3.*

**Remark 4.3** (Selection of $\tau$ and $\kappa_i$). *For every seller* $i \in \mathcal{N}$, *the parameter* $\kappa_i$ *can be chosen to minimize that seller's* total *expected regret. By Theorem 5.4 there is a unique optimum, denoted* $\kappa_i^\star = \kappa_i^\star(N, \tau) \in (0, 1)$ *that is characterized by Equation* (14). *Choosing* $\kappa_i = \kappa_i^\star$ *improves only the leading constant of the regret but leaves the convergence rate unchanged. For the exploration horizon* $\tau$ *we set* $\tau \propto T^\xi$ *for some* $\xi > 0$. *Remark 5.6 argues that* $\xi^\star = 5/7$ *is the unique value that minimizes the total expected regret across all sellers. These values* $\kappa_i^*$ *and* $\xi^\star$ *are "optimal" only in the sense that they aim to minimize the regret constant and the exponent of* $T$ *respectively within our derived upper bound; we do not claim minimax optimality of the regret rate itself.*

(1) **Exploration phase part 1**: $\mathcal{T}_i^{(1)}$. Each seller $i$ estimates $(-\widetilde{\beta}_i, \widetilde{\boldsymbol{\gamma}}_i) = \widetilde{\boldsymbol{\theta}}_i \triangleq \widehat{\boldsymbol{\theta}}_i / \|\widehat{\boldsymbol{\theta}}_i\|_2$, where

$$\widehat{\boldsymbol{\theta}}_i = \operatorname*{argmin}_{\boldsymbol{\theta} \in \mathbb{R}^N} \left\{ \mathscr{L}_i(\boldsymbol{\theta}) \triangleq \sum_{t \in \mathcal{T}_i^{(1)}} (y_i^{(t)} - \langle \boldsymbol{\theta}, \mathbf{p}^{(t)} - \bar{\mathbf{p}} \rangle)^2 \right\}, \quad \bar{\mathbf{p}} = \Sigma_{t \in \mathcal{T}_i^{(1)}} \mathbf{p}^{(t)} / |\mathcal{T}_i^{(1)}|. \quad (10)$$

(2) **Exploration phase part 2**: $\mathcal{T}_i^{(2)}$. Each seller $i$ defines $w_i^{(t)} \triangleq \langle \widetilde{\boldsymbol{\theta}}_i, \mathbf{p}^{(t)} \rangle$ for $t \in \mathcal{T}_i^{(2)}$ (note that $w_i^{(t)} \in \mathcal{U}$ because $\|\widetilde{\boldsymbol{\theta}}_i\|_2 = 1$, where $\mathcal{U}$ is defined in (5)) and the estimator $\widehat{\psi}_{i, \widetilde{\boldsymbol{\theta}}_i} = h_{s_i} \circ \widehat{\phi}_{i, \widetilde{\boldsymbol{\theta}}_i}$, where $s_i = c_i - 1$ (known by Assumption 3.4), and

$$\widehat{\phi}_{i, \widetilde{\boldsymbol{\theta}}_i} \in \operatorname{argmin}_{\phi \in \mathcal{H}_i} \sum_{t \in \mathcal{T}_i^{(2)}} (y_i^{(t)} - h_{s_i}(\phi(w_i^{(t)})))^2, \quad (11)$$

where $h_{s_i}$ is defined in (4). The class $\mathcal{H}_i$ consists of all functions $\phi : \mathcal{U} \to [\underline{B}_{\phi_i}, \bar{B}_{\phi_i}]$ that are both monotone non-decreasing and concave. The bounds $\underline{B}_{\phi_i}$ and $\bar{B}_{\phi_i}$ are chosen so that $[\underline{B}_{\psi_i}, \bar{B}_{\psi_i}] = h_{s_i}([\underline{B}_{\phi_i}, \bar{B}_{\phi_i}])$, and are therefore known, indeed $\underline{B}_{\psi_i}, \bar{B}_{\psi_i}$ (defined in Assumption 3.1) are known. A detailed justification of this estimator in (11) is provided in Appendix D.2.

At the end of the exploration phase, seller $i$ obtains an estimator of their revenue function:

$$\widehat{\text{rev}}_i(p_i|\mathbf{p}_{-i}^{(t-1)}) \triangleq p_i\widehat{\psi}_{i,\widetilde{\boldsymbol{\theta}}_i}(-\widetilde{\beta}_i p_i + \langle\widetilde{\boldsymbol{\gamma}}_i, \mathbf{p}_{-i}^{(t-1)}\rangle), \quad \forall p_i \in \mathcal{P}_i, \qquad (12)$$

where $\mathbf{p}_{-i}^{(t-1)}$ are the prices of all the other firms at time $t-1$, that are made public. A price $p_i^{(t)}$ is then offered by firm $i$ as a maximizer of (12) over $\mathcal{P}_i$. We present the full procedure in Algorithm 1 and a visual representation of the exploration-exploitation scheme in Figure 9 in Appendix K. A summary of the information available to each seller to run Algorithm 1 can be found in Appendix A.

---

**Algorithm 1** SPE-BR (Semi-Parametric Estimation then Best-Response)

---

**Input:** the (joint) distributions of the exploration phase $\mathscr{D}$.
**Output:** Price $p_i^{(t)}$ that seller $i \in \mathcal{N}$ offers in period $t \in \mathcal{T} \triangleq \{1, 2, \ldots, T\}$.
**for** $t \leq \tau$ **do**
    Sample $\mathbf{p}^{(t)} = (p_1^{(t)}, p_2^{(t)}, \ldots, p_N^{(t)}) \sim \mathscr{D}$, and let $y_i^{(t)}$ denote seller $i$'s demand as in (6).
**end for**
Each seller $i \in \mathcal{N}$ constructs an estimator $(\widetilde{\boldsymbol{\theta}}_i, \widehat{\psi}_{i,\widetilde{\boldsymbol{\theta}}_i}(\cdot))$ using data $\{(\mathbf{p}^{(t)}, y_i^{(t)})\}_{t=1,2,\ldots,\tau}$ as defined in (10) and (11)
**for** $t = \tau + 1, \tau + 2, \ldots, T$ **do**
    (1) Each seller $i \in \mathcal{N}$ offers a price $p_i^{(t)} = \text{argmax}_{p_i \in \mathcal{P}_i} \; p_i\widehat{\psi}_{i,\widetilde{\boldsymbol{\theta}}_i}(-\widetilde{\beta}_i p_i + \langle\widetilde{\boldsymbol{\gamma}}_i, \mathbf{p}_{-i}^{(t-1)}\rangle)$.
    (2) After all prices have been posted, each seller observes their demand $y_i^{(t)}$ as in (6).
**end for**

---

## 5 REGRET ANALYSIS

We begin by analyzing, for each seller $i \in \mathcal{N}$, the convergence of $\widetilde{\boldsymbol{\theta}}_i$ and $\widehat{\psi}_{i,\widetilde{\boldsymbol{\theta}}_i}$ from the exploration phase. An informal overview is given in Section 5.1, while formal results appear in Appendix D – Appendix D.1 for $\widetilde{\boldsymbol{\theta}}_i$ and Appendix D.2 for $\widehat{\psi}_{i,\widetilde{\boldsymbol{\theta}}_i}$. We then study NE convergence and the regret bound in Section 5.2.

### 5.1 ESTIMATION OF THE MODEL PARAMETERS

In the single-index setting, the linear estimator in (10) attains a $\sqrt{n}$ rate when covariates (here, joint prices $\mathbf{p}^{(t)}$) are elliptically distributed (Balabdaoui et al., 2019; Brillinger, 2012). We impose this elliptical law during exploration to ensure consistency and, unlike prior work, we derive a concentration inequality for (10) (Proposition D.3), which underpins our regret bound.

**Assumption 5.1.** $\mathscr{D} \equiv \mathscr{E}(\mathbf{m}, \Lambda, g)$ with $c_{\min}\mathbb{I} \preccurlyeq \Lambda \preccurlyeq c_{\max}\mathbb{I}$ for some $0 < c_{\min} < c_{\max} < \infty$, and $g$ satisfying $g(x + y) = g(x)g(y)$ for all $x, y \geq 0$.

Assumption 5.1 covers Gaussian and, more generally, elliptically symmetric laws with $g(x) = a^{-\gamma x}$ ($a > 0, \gamma > 0$). This yields sub-Gaussian samples, crucial for the concentration result in Proposition D.3. We defer to Remark D.1 and Remark D.2 for a detailed motivation on Assumption 5.1.

**Proposition 5.2** (Informal version of Proposition D.3). *Under Assumptions 3.1 and 5.1,* $\|\widetilde{\boldsymbol{\theta}}_i - \boldsymbol{\theta}_i\|_2 = O\big((N\log n_i^{(1)}/n_i^{(1)})^{1/2}\big)$.

**Theorem 5.3** (Informal version of Theorem D.6). *Let Assumptions 3.1, 3.4 and 5.1 hold. Then, for every compact $K \subset \mathcal{U}$ we have* $\mathbb{E}[\sup_{u \in K}|\widehat{\psi}_{i,\widetilde{\boldsymbol{\theta}}_i}(u) - \psi_i(u)|] = O\big((\log(n_i^{(2)})/n_i^{(2)})^{2/5}\big)$.

### 5.2 CONVERGENCE TO NASH EQUILIBRIUM AND INDIVIDUAL REGRET BOUNDS.

We are now ready to prove our main result for a fixed value $\xi \in (0, 1)$, which, we recall, represents the proportion of (common) exploration length $\tau \propto T^\xi$.

**Theorem 5.4.** *Suppose that Assumptions 3.1, 3.2, 3.4, 3.6 and 5.1 hold. Then, Algorithm 1 produces a policy such that, for each $i \in \mathcal{N}$:*

*(1)* **Individual sub-linear regret***:* $\text{Reg}_i(T) = O(T^\xi + T^{1-2\xi/5}N^{3/2}(\log T)^{2/5})$.

*(2)* **Convergence to NE**: $\mathbb{E}\left[\|\mathbf{p}^{(T)} - \mathbf{p}^*\|_2^2\right] = O(N^{3/2}T^{-2\xi/5}(\log T)^{2/5})$. *More precisely, we have*

$$\mathbb{E}[\|\mathbf{p}^{(t)} - \mathbf{p}^*\|_2^2] \lesssim N^{3/2}T^{-2\xi/5}(\log T)^{2/5} + L_{\boldsymbol{\Gamma}}^{2(t-\tau-1)}, \quad t \geq \tau + 1. \tag{13}$$

*Moreover, for each $i \in \mathcal{N}$, there exists a unique $\kappa_i^\star \in (0,1)$ that minimizes the seller's $i$ regret, and satisfies the implicit equation:*

$$4\mathscr{C}_i\tau^{1/10}\kappa_i^{3/2} = 5\mathscr{B}_iN^{1/2}(1-\kappa_i)^{7/5}. \tag{14}$$

*where $\mathscr{B}_i$ and $\mathscr{C}_i$ are defined in (17) and (21), respectively.*

**Remark 5.5.** *Since $\mathscr{C}_i$ and $\mathscr{B}_i$ depend on unknown quantities, it is typically difficult for seller $i$ to compute $\kappa_i$ exactly. However, any positive choice of $\mathscr{C}_i$ and $\mathscr{B}_i$ guarantees $\kappa_i \in (0,1)$, while the regret and NE convergence rates remain unaffected, with only the leading constants changing.*

**Remark 5.6** (Optimal value of $\xi$ and comparison with monopolistic results)**.** *By Theorem 5.4, the value of $\xi$ that minimizes the expected regrets $\mathrm{Reg}_i(T)$ is the value that equalizes the exponents of $T$ in the regret upper bound: $T^\xi + T^{1-2\xi/5}$ (ignoring logarithmic factors). This yields $\xi = 5/7$. Consequently, with this choice of $\xi$, we have that for each $i \in \mathcal{N}$*

$$\mathrm{Reg}_i(T) = O(N^{3/2}T^{5/7}(\log T)^{2/5}), \quad and$$
$$\mathbb{E}\|\mathbf{p}^{(T)} - \mathbf{p}^*\|_2 \leq (\mathbb{E}\|\mathbf{p}^{(T)} - \mathbf{p}^*\|_2^2)^{1/2} = O(N^{3/4}T^{-1/7}(\log T)^{1/5}).$$

*For $N = 1$ (i.e. in a monopolistic setting), our regret upper bound matches that of Fan et al. (2024), who estimate the link function via a kernel-based method. Their result is stated for binary outcomes $y_i$, whereas our framework accommodates continuous $y_i$; nevertheless, our estimation procedure can be straightforwardly adapted to the binary-response setting, yielding the same regret rate as in the continuous case. Moreover, relative to the kernel approach in Fan et al. (2024), our method offers the advantage that is fully data-driven and requires no bandwidth selection.*

**Remark 5.7** (Mispecifying $s_i$)**.** *As established in Remark G.3, if the sellers choose parameters $s_i' \leq s_i$ satisfying $\sup_{i \in \mathcal{N}} |1 - 1/(s_i'+1)| \, \|\gamma_i\|_1/\beta_i < 1$, then the convergence rates of the regret and the NE are preserved. This phenomenon is also confirmed empirically in Section 6.*

**Remark 5.8** (Algorithmic collusion)**.** *Our results indicate that when all sellers employ the same class of learning algorithm that we propose, the dynamics do not give rise to collusive behavior: the learning process converges to the pure Nash equilibrium, and prices do not drift upward in a coordinated way. This is beneficial from a consumer perspective, as it prevents the emergence of a price increase. However, if some sellers deviate and adopt different algorithms or learning rules, then the interaction dynamics may create conditions under which algorithmic collusion can emerge.*

## 6 NUMERICAL EXPERIMENTS

We evaluate the performance of Algorithm 1 in markets with $N \in \{2,4,6\}$ sellers. A detailed description of the simulation setup is provided in Appendix J; here we present the main setup together with a discussion of the results. For each seller $i \in \mathcal{N} = \{1,2,\ldots,N\}$, the price vector $\mathbf{p}^{(t)}$ is supported on $\mathcal{P} = [0,3]^{\times N}$. For every $N$, we examine the convergence behavior as the contraction constant $L_{\boldsymbol{\Gamma}}$ varies – specifically, when $L_{\boldsymbol{\Gamma}} \approx 0$, $L_{\boldsymbol{\Gamma}} \approx 0.5$, and $L_{\boldsymbol{\Gamma}} \approx 1$. The link functions are 0-concave and their price sensitivities are generated such that $\boldsymbol{\theta}_i = (-\beta_i, \boldsymbol{\gamma}_i)$ has unitary $L^2$-norm and satisfies Assumption 3.6. A fixed point exists and is unique by Lemma 3.7 and is recovered using a simple root-finding algorithm. The demand noise of each firm follows a uniform distribution on $[-0.05, 0.05]$. For every $T \in [100, 400, 800, 1600, 3200]$, we apply Algorithm 1. We independently repeat the simulation 30 times to obtain average performances and 95% of confidence intervals. In Figure 5 and Figure 6 we summarize the results. The panels show: (i) the convergence of the estimators $\boldsymbol{\theta}_i$ and $\psi_i$; (ii) convergence of prices to the Nash equilibrium $\mathbf{p}^\star$ and the expected cumulative regret plotted on log–log axes (with fitted slopes) to reveal the empirical rate. Across runs, the observed rates are consistently faster – i.e., exhibit smaller slopes than the theoretical upper bound. Regarding the minimization problem in (11) under the class of monotone increasing and $s$-concave functions, we note that designing efficient algorithms specifically tailored to $s$-concavity remains an open and largely unexplored area. Nonetheless, in this work, we leverage a fast optimization solver to approximate the solution. Additionally, in Appendix J.2 we provide simulations under different exploration phases across the sellers and in Appendix J.3 simulations showing how mispecifying $s_i$ affects the regret and NE convergence rates.

## 7 FUTURE RESEARCH DIRECTIONS.

There are several promising directions for future research. One is to design an algorithm allowing each seller to independently set their exploration length, which is not permitted in our setting as outlined in Remark 4.1. For this purpose, a key theoretical goal is to obtain *uniform* convergence for the joint estimator $(\boldsymbol{\theta}_i, \psi_i)$. While this would not change regrets or NE rates (it will only improve constants), it could yield practical gains: currently, $\boldsymbol{\theta}_i$ uses $\kappa_i \tau$ samples and $\psi_i$ the remaining $(1 - \kappa_i)\tau$, whereas a joint estimator would leverage the full $\tau$. Existing results are limited: Balabdaoui et al. (2019) shows only $L^2$ convergence under monotonicity (we additionally require $s_i$-concavity), and Kuchibhotla et al. (2021) provides uniform convergence under convexity/concavity ($s_i = 1$) with only $O_P$ rates. Our setting instead calls for non-asymptotic *supremum-norm* concentration for the joint estimator, i.e., tail bounds as in (52).

Another challenge is estimating the $s_i$'s themselves. While we assume them to be known, in practice, this may not be the case, and developing a theory for this harder setting remains to be explored.

Finally, an ambitious direction is to design algorithms that bypass the classic exploration-exploitation trade-off. Here, each seller $i$ would use the entire history of public prices and private demands $\{(\mathbf{p}^{(s)}, y_i^{(s)}\}_{s \leq t}$, ensuring $\mathbf{p}^{(T)} \to \text{NE}$ as $T \to \infty$. The main challenge is avoiding *incomplete learning* (Keskin & Zeevi, 2018), which can yield poor parameter estimates and higher regrets.

## REPRODUCIBILITY STATEMENT

In the Supplementary Materials, we provide a `.zip` archive with all the code required to reproduce the results and figures in this paper. The archive contains three top–level folders whose names match the corresponding figure labels in the PDF. Inside each folder, we include Python notebooks with step-by-step instructions that describe how to run the experiments and regenerate the figure. Running the notebooks as instructed will recreate the plots and save them to the corresponding folders' directory named "PLOTS".

## 8 DETAILS OF LLM USAGE

We used generative AI tools when preparing the manuscript to polish our sentences and correct potential typos; we remain responsible for all opinions, findings, and conclusions or recommendations expressed in the paper.

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

# Appendix of
# Revenue Maximization Under Sequential Price Competition Via The Estimation Of $s$-Concave Demand Functions

## Appendix Contents

# A    SUMMARY OF INDIVIDUAL INFORMATION

In this section, we summarize the information each seller must know to implement Algorithm 1.

Table 1: Values known by seller $i \in \mathcal{N}$

| Known (or observed if specified) | Description |
| --- | --- |
| $T$ | Time horizon |
| $y_i$ (observed) | Demand realization of seller $i$ |
| $\mathbf{p}_{-i}$ (observed) | Competitors' prices |
| $\mathcal{P}_i = [\underline{p}_i, \bar{p}_i]$ | Price domain of seller $i$ |
| $\mathcal{U} = [-p_{\max}, p_{\max}]$ | Domain of $\psi_i : \mathcal{U} \to [\underline{B}_{\psi_i}, \bar{B}_{\psi_i}]$ |
| $[\underline{B}_{\psi_i}, \bar{B}_{\psi_i}]$ | Range of $\psi_i : \mathcal{U} \to [\underline{B}_{\psi_i}, \bar{B}_{\psi_i}]$ |
| $c_i$ (or equivalently $s_i = c_i - 1$) | $c_i$: lower bound of $\varphi_i'$. $s_i$: concavity parameter of $\psi_i$. |
| $\mathscr{D}$ | Joint price distribution during exploration |
| $\tau$ | Length of the common exploration phase |

Table 2: Values unknown by seller $i \in \mathcal{N}$

| Unknown (or unobserved if specified) | Description |
| --- | --- |
| $(\boldsymbol{\theta}_i, \psi_i)$ | Semi-parametric population parameter |
| $\underline{\beta}_i$ | Lower bound of the price sensitivity $\beta_i$ |
| $\boldsymbol{y}_{-i}$ (unobserved) | Demand realization of the $i$-th competitors |
| $\sigma_i$ | Variance of the $i$-th demand noise $\varepsilon_i$ |
| $\mathcal{P}_{-i} = [\underline{p}_i, \bar{p}_i]$ | Price domains of the $i$-th competitors |
| $\underline{B}_{\psi_i'}, \bar{B}_{\psi_i'}$ | Lower and upper bound of $\psi_i'$ |
| $B_{\psi_i''}$ | Upper bound of $|\psi_i''|$ |
| $\{[\underline{B}_{\psi_j}, \bar{B}_{\psi_j}]\}_{j \neq i}$ | Ranges of $i$-th competitors demands $\{\psi_j\}_{j \neq i}$ |
| $\{c_j\}_{j \neq i}$ | Lower bounds of competitors $\{\varphi_i'\}_{j \neq i}$ |

# B   PROOF OF PROPOSITION 3.5

Recall the definition

$$\varphi_i(u) = u + \frac{\psi_i(u)}{\psi_i'(u)}, \quad u \in \mathcal{U}.$$

We have $\varphi_i' = 1 + \frac{\psi_i' \cdot \psi_i' - \psi_i \cdot \psi_i''}{(\psi_i')^2} = 2 - \frac{\psi_i \cdot \psi_i''}{(\psi_i')^2} \geq c_i$ iff $\psi_i \cdot \psi_i'' + (s_i - 1)(\psi_i')^2 \leq 0$ with $s_i = c_i - 1$.
The statement follows by Lemma B.1.

**Lemma B.1.** *Let $\psi$ be a positive function defined in an interval $(a, b)$ that is twice continuously differentiable. Then $\psi$ is s-concave iff $\psi \cdot \psi'' + (s - 1)(\psi')^2 \leq 0$ in $(a, b)$.*

*Proof of Lemma B.1.* As for $s = 0$ is a known result, we prove it for $s \neq 0$. A function $\psi$ is $s$-concave if and only if $d_s \circ \psi$ is concave, where $d_s$ is defined in (4). Then $\psi$ is $s$-concave if and only if

$$\frac{d^2}{du^2} d_s(\psi(u)) = \frac{d}{du} \left[ |s| \psi^{s-1}(u) \psi'(u) \right] = |s| \frac{\psi(u)\psi''(u) + (s-1)(\psi'(u))^2}{\psi^{2-s}(u)} \leq 0, \quad u \in (a, b).$$

$\square$

# C   PROOF OF LEMMA 3.7

We first find an analytic form of $\Gamma_i(\mathbf{p}_{-i}^*)$. We have

$$\lambda_i(\mathbf{p}) = \psi_i(\boldsymbol{\theta}_i(\mathbf{p})), \quad \boldsymbol{\theta}_i(\mathbf{p}) = -\beta_i p_i + \boldsymbol{\gamma}_i^\top \mathbf{p}_{-i}.$$

First note that, by strong concavity of $p_i \to \mathrm{rev}_i(p_i \mid \mathbf{p}_{-i})$ (see Assumption 3.2), the best response map is the projection onto $\mathcal{P}_i$ or the value $p_i^*$ that solves the first order condition. More specifically

$$\Gamma_i(\mathbf{p}_{-i}^*) = \Pi_{\mathcal{P}_i} p_i^*,$$

where $\Pi_{\mathcal{P}_i}$ is the projection into $\mathcal{P}_i$ and $p_i^*$ solves the first order condition of (8), that is

$$\frac{\partial}{\partial p_i} \mathrm{rev}_i(p_i|\mathbf{p}_{-i}) = 0. \tag{15}$$

where we that $\mathrm{rev}_i(p_i|\mathbf{p}_{-i}) = p_i \psi_i(\boldsymbol{\theta}_i(\mathbf{p}))$. Solving Equation (15) we have

$$
\begin{aligned}
\psi_i(\boldsymbol{\theta}_i(\mathbf{p}^*)) - \beta_i p_i^* \psi_i'(\boldsymbol{\theta}_i(\mathbf{p}^*)) = 0 &\Leftrightarrow \beta_i p_i^* - \frac{\psi_i(\boldsymbol{\theta}_i(\mathbf{p}^*))}{\psi_i'(\boldsymbol{\theta}_i(\mathbf{p}^*))} = 0 \\
&\Leftrightarrow \varphi_i(\boldsymbol{\theta}_i(\mathbf{p}^*)) = \boldsymbol{\gamma}_i^\top \mathbf{p}_{-i}^* \\
&\overset{Assumption\ 3.4}{\Leftrightarrow} \boldsymbol{\theta}_i(\mathbf{p}^*) = \varphi_i^{-1}(\boldsymbol{\gamma}_i^\top \mathbf{p}_{-i}^*) \\
&\Leftrightarrow p_i^* = \frac{\boldsymbol{\gamma}_i^\top \mathbf{p}_{-i}^* - \varphi_i^{-1}(\boldsymbol{\gamma}_i^\top \mathbf{p}_{-i}^*)}{\beta_i} \\
&\Leftrightarrow p_i^* = \frac{g_i(\boldsymbol{\gamma}_i^\top \mathbf{p}_{-i}^*)}{\beta_i}.
\end{aligned}
$$

Thus

$$\Gamma_i(\mathbf{p}_{-i}^*) = \Pi_{\mathcal{P}_i} \frac{g_i(\boldsymbol{\gamma}_i^\top \mathbf{p}_{-i}^*)}{\beta_i}. \tag{16}$$

We now compute an upper bound of the Lipschitz constant of the Best Response Operator $\boldsymbol{\Gamma} \triangleq (\Gamma_i)_{i \in \mathcal{N}}$ on the compact space $\mathcal{P}$. This will allow us to ensure contractiveness when this constant is

strictly less than 1. For any $\mathbf{p}, \mathbf{p}' \in \mathcal{P}$, we have

$$
\begin{aligned}
\|\mathbf{\Gamma}(\mathbf{p}) - \mathbf{\Gamma}(\mathbf{p}')\|_\infty &= \sup_{i \in \mathcal{N}} |\Gamma_i(\mathbf{p}_{-i}) - \Gamma_i(\mathbf{p}'_{-i})| \\
&\leq \sup_{i \in \mathcal{N}} \left| \frac{g_i(\boldsymbol{\gamma}_i^\top \mathbf{p}_{-i}) - g_i(\boldsymbol{\gamma}_i^\top \mathbf{p}'_{-i})}{\beta_i} \right| \\
&\leq \sup_{i \in \mathcal{N}} \frac{1}{\beta_i} \|g_i'\|_\infty \left| \boldsymbol{\gamma}_i^\top \mathbf{p}_{-i} - \boldsymbol{\gamma}_i^\top \mathbf{p}'_{-i} \right| \\
&\leq \sup_{i \in \mathcal{N}} \frac{1}{\beta_i} \|g_i'\|_\infty \|\boldsymbol{\gamma}_i\|_1 \|\mathbf{p}_{-i} - \mathbf{p}'_{-i}\|_\infty \\
&\leq \|\mathbf{p} - \mathbf{p}'\|_\infty \sup_{i \in \mathcal{N}} \|g_i'\|_\infty \frac{\|\boldsymbol{\gamma}_i\|_1}{\beta_i} \\
&= L_{\mathbf{\Gamma}} \|\mathbf{p} - \mathbf{p}'\|_\infty,
\end{aligned}
$$

where

$$
\|g_i'\|_\infty = \sup_{v \in V} |g_i(v)|, \quad V \triangleq \varphi_i(\mathcal{U}).
$$

In the second inequality, we used the contraction property of the projection operator, and where

$$
L_{\mathbf{\Gamma}} \triangleq \sup_{i \in \mathcal{N}} \|g_i'\|_\infty \frac{\|\boldsymbol{\gamma}_i\|_1}{\beta_i},
$$

which is strictly less than 1 by Assumption 3.6.

## D  FORMAL ESTIMATION OF THE MODEL PARAMETERS IN SECTION 5.1

In this section, we formally explain the estimation procedure of the model parameters $(\boldsymbol{\theta}_i, \psi_i)$, for $i \in \mathcal{N}$.

### D.1  ESTIMATION OF THE PARAMETRIC COMPONENT.

In a single-index model, the linear estimator in (10) converges to the true $\boldsymbol{\theta}_i$ at a $\sqrt{n}$ rate provided that the covariates (in this case, the joint prices $\mathbf{p}^{(t)}$) follow an elliptical distribution (Balabdaoui et al., 2019; Brillinger, 2012). We adopt the same elliptical assumption on the (joint) distribution of the prices during exploration, to ensure the consistency of the estimator in (10). In contrast to earlier work, we establish a concentration inequality for this estimator (Proposition D.3), which serves as a key tool in deriving our regret upper bound.

More specifically, in Assumption 5.1 we assume that $\mathscr{D} = \mathscr{E}(\mathbf{m}, \Lambda, g)$, where $c_{\min} \mathbb{I} \preccurlyeq \Lambda \preccurlyeq c_{\max} \mathbb{I}$ for some $0 < c_{\min} < c_{\max} < \infty$ and the density generator $g$ is such that $g(x+y) = g(x)g(y)$ for all $x, y \geq 0$, i.e.

$$
f_{\mathbf{p}}(\mathbf{p}) \propto g((\mathbf{p} - \mathbf{m})^T \Lambda^{-1} (\mathbf{p} - \mathbf{m})).
$$

**Remark D.1.** *It should be noted that from a purely theoretical standpoint, we do not restrict the domain of the price vector* $\mathbf{p}$ *in the exploration phase to* $\mathcal{P}$*, but this has essentially no implication for the application of the pricing algorithm in practice. To ensure that the $i$'th seller's exploration prices can, realistically, never lie outside* $\mathcal{P}_i = [\underline{p}_i, \bar{p}_i]$*, all they need is to take* $\mathscr{D}_i$ *be a normal distribution (for example) centered at the midpoint of this interval with variance* $\sigma_i^2$ *taken to be an adequately small fraction of the length of* $\mathcal{P}_i$*. If the sellers function independently, as is generally the case, the joint distribution is certainly elliptically symmetric. More generally, view the parameters* $\Lambda$*,* $\mathbf{m}$*, and* $g$ *as concentrating the mass of* $\mathscr{D}$ *in* $\mathcal{P}$ *with overwhelmingly high probability. Theoretically, some proposed price in many thousands of time steps could lie outside* $\mathcal{P}$*, but realistically, the procedure entails no financial consequence for them. Letting* $\mathscr{D}$ *belong to an infinitely supported elliptical distribution allows the implementation of a simple consistent linear regression-based estimation to learn* $\boldsymbol{\theta}_i$ *which satisfies the concentration inequality proved in Proposition D.3 below, and is also important for certain conditioning arguments involved in estimating appropriate surrogates for* $\psi_i$ *in the following subsection, while compromising nothing in terms of the core conceptual underpinnings of the problem.*

**Remark D.2.** *Assumption 5.1 holds for a wide range of distributions, including Gaussian distributions and, more generally, all elliptically symmetric distributions for which $g(x) = a^{-\gamma x}$, for $a > 0$ and $\gamma > 0$. Before describing Assumption 5.1, we emphasize that selecting an appropriate distribution for the design points (here, the exploration prices) is standard in exploration–exploitation algorithms. This choice is crucial because it ensures that the learner can perform exploration design and obtain a consistent estimator. For example, both Fan et al. (2024) and Luo et al. (2024) employ uniformly randomized prices precisely to guarantee consistency of their estimators. We now describe the components of Assumption 5.1:*

1. *The requirement that exploration prices follow an elliptically symmetric distribution is used to guarantee the consistency of the estimator of $\boldsymbol{\theta}_i$ in the single-index model. In particular, this assumption is invoked in Lemma E.2 to derive the normal equations (the same assumption can be found in Balabdaoui et al. (2019) and Brillinger (2012)). If the exploration distribution were not elliptically symmetric, the resulting estimator could fail to be consistent.*

2. *As we will discuss in Appendix D.2, the property that $g(x + y) = g(x)g(y)$ (satisfied, for example, by Gaussian distributions) supports the estimation of $\psi_i$. It ensures that $\psi_{i,\boldsymbol{\theta}}(u)$ (defined in (18)) depends on $u$ solely through the argument of $\psi_i$ rather than that of $g$ (see Equation (29)). This decoupling guarantees that $\psi_{i,\boldsymbol{\theta}}$ inherits key properties from $\psi_i$, such as smoothness and adherence to the prescribed shape constraints, as highlighted in Proposition D.4. In turn, these properties are crucial for establishing consistency of the estimator of $\psi_i$.*

We denote

$$\Sigma = \mathrm{var}(\mathbf{p}),$$

which is proportional to $\Lambda$. Then, there exists $0 < \varsigma_{\min} < \infty$ such that $\varsigma_{\min} \preccurlyeq \Sigma$. These constants will appear in Proposition D.3, which proof is provided in Appendix E.

**Proposition D.3.** *Under Assumption 3.1 and Assumption 5.1, there exist a positive constant $c$ (that depend solely on the variance proxy $\sigma_x$ of the sub-Gaussian random variable $\mathbf{x}^{(t)} = \mathbf{p}^{(t)} - \bar{\mathbf{p}}$) such that, for $n_i^{(1)}$ sufficiently large, with probability at least $1 - 2e^{-c\varsigma_{\min}^2 n_i^{(1)}/16} - \frac{2}{n_i^{(1)}}$, the estimator in (10) satisfies*

$$\|\widetilde{\boldsymbol{\theta}}_i - \boldsymbol{\theta}_i\|_2 \leq \mathscr{B}_i \sqrt{\frac{N \log n_i^{(1)}}{n_i^{(1)}}}.$$

*where*

$$\mathscr{B}_i := 4\sqrt{2} \frac{\sigma_x(\lambda_i \sigma_x + \sigma_{y_i})}{\lambda_i \varsigma_{\min}}. \tag{17}$$

*where $\lambda_i := \frac{\mathrm{cov}\left(\psi_i(\boldsymbol{\theta}_i^\top \mathbf{x}), \boldsymbol{\theta}_i^\top \mathbf{x}\right)}{\mathrm{var}(\boldsymbol{\theta}_i^\top \mathbf{x})} > 0$, for $\mathbf{x} \sim \mathbf{x}^{(t)} = \mathbf{p}^{(t)} - \bar{\mathbf{p}}$ and $\sigma_{y_i}$ is the variance proxy of the the sub-Gaussian random variables $y_i^{(t)}$.*

## D.2 ESTIMATION OF THE LINK FUNCTION VIA S-CONCAVITY.

This section provides a uniform convergence result of $\widehat{\psi}_{i,\boldsymbol{\theta}}$ to $\psi_i$ for any fixed $\boldsymbol{\theta} \in \mathbb{S}_{N-1}$. *In this section, all the results must be considered as conditioned on $\boldsymbol{\theta} = \widetilde{\boldsymbol{\theta}}_i$ estimated using data in $\mathcal{T}_i^{(1)}$ independent of data in $\mathcal{T}_i^{(2)}$. For simplicity of notation, we re-index $\mathcal{T}_i^{(2)}$ as $[n] = \{1, 2, \ldots, n\}$.* To estimate $\psi_i$ we would need to know $\boldsymbol{\theta}_i$ in advance, indeed we remind that $\mathbb{E}(y_i^{(t)} \mid \mathbf{p}^{(t)}) = \psi_i(\langle \boldsymbol{\theta}_i, \mathbf{p}^{(t)} \rangle)$. However, our knowledge is limited to an approximation $\boldsymbol{\theta}$ of $\boldsymbol{\theta}_i$, and the observable design points are $w_i^{(t)} \triangleq \langle \boldsymbol{\theta}, \mathbf{p}^{(t)} \rangle$. Note that

$$w_i^{(t)} \in \mathcal{U}$$

because $\|\widetilde{\boldsymbol{\theta}}_i\|_2 = 1$ implies $|w_i^{(t)}| \leq \|\mathbf{p}^{(t)}\|_2 \leq p_{\max}$. This implies that, with data $\{(w_i^{(t)}, y_i^{(t)})\}_{t \in [n]}$, we are only able to estimate

$$\psi_{i,\boldsymbol{\theta}}(u) \triangleq \mathbb{E}(y_i^{(t)} \mid w_i^{(t)} = u), \quad u \in \mathcal{U}. \tag{18}$$

We now construct an estimator of $\psi_{i,\boldsymbol{\theta}}$ and establish, in Theorem D.6, a uniform convergence result over any bounded subset $K \subset \mathcal{U}$ (where $\mathcal{U}$ is defined in (5)). Our approach draws on techniques from Mösching & Dümbgen (2020); Dümbgen et al. (2004), which study uniform convergence in the context of shape-constrained estimation. A key assumption in these works – one that does not require smoothness of the unknown nonparametric function – is that the density of the design points, $f_w$, is bounded away from zero on $\mathcal{U}$. This ensures that the design points $w_i^{(t)} \in \mathcal{U}$ are asymptotically dense within every subinterval of $\mathcal{U}$, which in turn is sufficient for establishing uniform convergence of the estimator on such intervals. In our setting, $f_w$ is supported on all of $\mathbb{R}$, and thus it is bounded away from zero on any bounded set $K$. Consequently, the uniform convergence result applies to any bounded subset $K \subset \mathcal{U}$.

Before establishing the convergence result, we must first characterize the shape constraints satisfied by the family of functions $\{\psi_{i,\boldsymbol{\theta}}(\cdot) : \boldsymbol{\theta} \in \mathbb{S}_{N-1}\}$. In Proposition D.4 (with proof deferred to Appendix F), we derive a closed-form expression for $\psi_{i,\boldsymbol{\theta}}(\cdot)$ and show that both the monotonicity and the $s_i = c_i - 1$-concavity of the original function $\psi_i$ are preserved by $\psi_{i,\boldsymbol{\theta}}$, uniformly over $\boldsymbol{\theta} \in \mathbb{S}_{N-1}$.

**Proposition D.4.** *Under Assumption 3.1 and Assumption 5.1 we have the following properties:*

*(1)* $\psi_{i,\boldsymbol{\theta}}(u) = \int_{\mathbb{R}^{N-1}} \psi_i \left( \langle \boldsymbol{\theta}_i, \mathbf{m}_u \rangle + \boldsymbol{\theta}_i^\top A \boldsymbol{\alpha} \right) \tilde{g}(\boldsymbol{\alpha}) d\boldsymbol{\alpha}$, $u \in \mathbb{R}$, $\boldsymbol{\theta} \in \mathbb{S}_{N-1}$, where*

$$\tilde{g}(\boldsymbol{\alpha}) = g(\|\boldsymbol{\alpha}\|_2^2) / \int_{\mathbb{R}^{N-1}} g(\|\boldsymbol{\delta}\|_2^2) d\boldsymbol{\delta}, \quad \mathbf{m}_u = \mathbf{m} + \Lambda \boldsymbol{\theta} \frac{(u - \langle \boldsymbol{\theta}, \mathbf{m} \rangle)}{\boldsymbol{\theta}^\top \Lambda \boldsymbol{\theta}},$$

*and $A$ is such that $AA^\top = \Lambda$.*

*(2)* $\psi_{i,\boldsymbol{\theta}} \in \mathrm{C}^2(\mathcal{U})$ and $0 < \underline{B}_{\psi_i} \leq \psi_{i,\boldsymbol{\theta}} \leq \bar{B}_{\psi_i}$ uniformly in $\boldsymbol{\theta} \in \mathbb{S}_{N-1}$.

*(3)* $|\psi_i(u) - \psi_{i,\boldsymbol{\theta}}(u)| \lesssim \|\boldsymbol{\theta} - \boldsymbol{\theta}_i\|_2$ for all $u \in \mathcal{U}$ and $\boldsymbol{\theta} \in \mathbb{S}_{N-1}$.

*(4)* $\psi_{i,\boldsymbol{\theta}}$ is monotone increasing in $\mathcal{U}$ for all $\boldsymbol{\theta}$ such that $\|\boldsymbol{\theta} - \boldsymbol{\theta}_i\|_2 < \frac{\boldsymbol{\theta}_i^\top \Lambda \boldsymbol{\theta}_i}{c_{\max}}$. Moreover $|\psi'_{i,\boldsymbol{\theta}}|$ is bounded uniformly in $\boldsymbol{\theta} \in \mathbb{S}_{N-1}$ and there exists $L > 0$ independent of $\boldsymbol{\theta}$, such that $|\psi'_{i,\boldsymbol{\theta}}(u) - \psi'_{i,\boldsymbol{\theta}}(v)| \leq L|u - v|$ for all $u, v \in \mathcal{U}$, $\boldsymbol{\theta} \in \mathbb{S}_{N-1}$.

*(5)* If also Assumption 3.4 holds (i.e. $\psi_i$ is $s_i$-concave for some $s_i > -1$ as it arises from Proposition 3.5) then $\psi_{i,\boldsymbol{\theta}}$ is $s_i$-concave for any $\boldsymbol{\theta} \in \mathbb{S}_{N-1}$.

By definition of $s$-concavity, from point *(5)* in Proposition D.4 we have that for every $\boldsymbol{\theta} \in \mathbb{S}_{N-1}$ there exists a concave function $\phi_{i,\boldsymbol{\theta}}$ that $\psi_{i,\boldsymbol{\theta}} = h_{s_i} \circ \phi_{i,\boldsymbol{\theta}}$, where $h_{s_i}$ is defined in (4), and $s_i = c_i - 1$ is known by Assumption 3.4. We could then estimate $\phi_{i,\boldsymbol{\theta}}(u)$ using LSE under concavity restriction, that is

$$\widehat{\phi}_{i,\boldsymbol{\theta}} \in \operatorname{argmin}_{\phi \in \mathcal{S}_i} \sum_{t \in [n]} (y_i^{(t)} - h_{s_i} \circ \phi(w_i^{(t)}))^2, \quad \mathcal{S}_i = \{\phi : \mathcal{U} \to [\underline{B}_{\phi_i}, \bar{B}_{\phi_i}] \text{ concave}\}, \quad (19)$$

where we recall $w_i^{(t)} \triangleq \langle \boldsymbol{\theta}, \mathbf{p}^{(t)} \rangle$. The interval $[\underline{B}_{\phi_i}, \bar{B}_{\phi_i}]$ is such that $[\underline{B}_{\psi_i}, \bar{B}_{\psi_i}] = h_{s_i}([\underline{B}_{\phi_i}, \bar{B}_{\phi_i}])$. Note that $[\underline{B}_{\phi_i}, \bar{B}_{\phi_i}]$ is known because $\underline{B}_{\psi_i}$ and $\bar{B}_{\psi_i}$ are known by Assumption 3.1 (recll that, by point *(2)* in Proposition D.4, the bounds $\underline{B}_{\psi_i}, \bar{B}_{\psi_i}$ are inherited from $\psi_i$ to $\psi_{i,\boldsymbol{\theta}}$). We want to highlight that for any fixed $\boldsymbol{\theta} \in \mathbb{S}_{N-1}$, the problem (19) is not infinite dimensional, indeed we only need to recover $n$ variables $(\phi_i(w_1^{(t)}), \phi_i(w_2^{(t)}), \ldots, \phi_i(w_n^{(t)})) \in [\underline{B}_{\phi_i}, \bar{B}_{\phi_i}]$. A full discussion on the optimization problem (19) can be found in Appendix H.

**Remark D.5.** *The problem in (19) serves purely as a theoretical intermediate step for establishing the regret upper bound in Theorem 5.4. In practice, we do not compute $\hat{\psi}_{i,\boldsymbol{\theta}}$ for every $\boldsymbol{\theta}$. As discussed in Algorithm 1, we only solve (19) once, with $\boldsymbol{\theta} = \tilde{\boldsymbol{\theta}}_i$, to obtain $\hat{\psi}_{i,\tilde{\boldsymbol{\theta}}_i}$. Moreover, by point (4) in Proposition D.4, the function $\psi_{i,\boldsymbol{\theta}}$ is increasing (for $\boldsymbol{\theta}$ sufficiently close to $\boldsymbol{\theta}_i$). Since $h_{s_i}$ is also increasing, it follows that $\phi_{i,\boldsymbol{\theta}}$ inherits this monotonicity. Nonetheless, we did not include this monotonicity constraint in the definition of $\mathcal{S}_i$ in (19). This omission is justified because the theoretical convergence rate in Theorem D.6, and consequently the regret upper bound in Theorem 5.4, is driven entirely by the concavity constraint, which imposes a higher-order smoothness condition than monotonicity. However, we do enforce monotonicity in the algorithmic implementation in Algorithm 1.*

We are now prepared to demonstrate the convergence of $\widehat{\psi}_{i,\boldsymbol{\theta}}$ to $\psi_{i,\boldsymbol{\theta}}$, where

$$\widehat{\psi}_{i,\boldsymbol{\theta}} \triangleq h_{s_i} \circ \widehat{\phi}_{i,\boldsymbol{\theta}}.$$

**Theorem D.6.** *Let Assumptions 3.1, 3.4 and 5.1 hold. Then, by Proposition 3.5 and Proposition D.4, $\psi_{i,\boldsymbol{\theta}}$ is $s_i$-concave, where $s_i = c_i - 1 > -1$. Let $m = \pi n$, where $\pi = \int_{\mathcal{U}} f_w(u)du$ is the proportion of data points $\{w_i^{(t)}\}_{t \in [n]}$ lying in the set $\mathcal{U}$ as $n \to \infty$. For every $\gamma > 4$ there exists $m_0$, a constant $\mathscr{C}_i > 0$ and $\widetilde{\mathscr{C}}_i > 0$ depending only on constants in the assumptions such that*

$$\mathbb{P}\left\{ \sup_{\boldsymbol{\theta} \in \mathbb{S}_{N-1}} \sup_{u \in \mathcal{U}_m} |\widehat{\psi}_{i,\boldsymbol{\theta}}(u) - \psi_{i,\boldsymbol{\theta}}(u)| \leq \mathscr{C}_i(\log(m)/m)^{2/5} \right\} \geq 1 - 1/m^{\gamma-2}, \quad m \geq m_0, \quad (20)$$

*where $\mathcal{U}_m = \{u \in \mathcal{U} : [u \pm \delta_m] \subset \mathcal{U}\}$, with $\delta_m = \widetilde{\mathscr{C}}_i(\log(m)/m)^{1/5}$. More specifically,*

$$\mathscr{C}_i = \max\left\{ \frac{4^3 L_i}{3} \bar{B}_{\psi_i}, \frac{4\sqrt{7}\bar{B}_{\psi_i}\gamma\sigma_i\sqrt{\bar{B}_{\psi_i}^2 + \underline{B}_{\psi_i}^2}}{\underline{B}_{\psi_i}^2 (C_{\mathscr{D}}/8)^{1/2}} \right\}, \quad (21)$$

*where $L_i := \sup_{\boldsymbol{\theta} \in \mathbb{S}_{N-1}} \sup_{u,v \in \mathcal{U}, u \neq v} \frac{\phi'_{i,\boldsymbol{\theta}}(u) - \phi'_{i,\boldsymbol{\theta}}(v)}{u-v}$, $C_{\mathscr{D}} > 0$ is such that $\inf_{\boldsymbol{\theta} \in \mathbb{S}_{N-1}} \inf_{u \in \mathcal{U}} f_{\boldsymbol{\theta}^\top \mathbf{p}}(u) > C_{\mathscr{D}}$, and $\sigma_i$ is the sub-gaussian variance proxy of the error $\varepsilon_i$.*

*Moreover, for $\boldsymbol{\theta} = \widetilde{\boldsymbol{\theta}}_i$ we have*

$$\mathbb{E}\left[ \sup_{u \in \mathcal{U}_m} |\widehat{\psi}_{i,\widetilde{\boldsymbol{\theta}}_i}(u) - \psi_i(u)| \right] = O((\log(m)/m)^{2/5}). \quad (22)$$

*Proof of Theorem D.6.* We first prove the concentration inequality in (20).

Let $m = \pi n$ be the asymptotic fraction of points lying in $\mathcal{U}$, where $n$ is the sample size and $\pi = \int_{\mathcal{U}} f_w(w) \, dw$. Define the empirical fraction of points in $\mathcal{U}$ as the random quantity

$$\hat{\pi}_n = \frac{1}{n} \sum_{t=1}^{n} \mathbf{1}\{w_i^{(t)} \in \mathcal{U}\}.$$

By Lemma H.9, the event

$$\mathscr{Q} = \left\{ \frac{\pi}{\hat{\pi}_n} > 1 - \epsilon_n \right\},$$

holds with probability at least $1 - 1/2(n+2)^\gamma$ for some $\gamma > 0$, provided that $n$ is sufficiently large and $\epsilon_n \to 0$ as $n \to \infty$. Now, we apply Theorem H.5, using Remark H.6, with the following substitutions:

$$\psi_0 \leftarrow \psi_{i,\boldsymbol{\theta}}, \quad \widehat{\psi} \leftarrow \widehat{\psi}_{i,\boldsymbol{\theta}}, \quad \alpha \leftarrow 2, \quad s \leftarrow s_i, \quad h \leftarrow h_{s_i} = d_{s_i}^{-1},$$

where $h_{s_i}$ is defined in (4). Intersecting $\mathscr{Q}$ with the high-probability event in Theorem H.5 yields the desired result. The only remaining step is to verify that the assumptions of Theorem D.6 imply those of Theorem H.5.

(1) $\psi_{i,\boldsymbol{\theta}}$ is $s_i$-concave for some $s_i > -1$. According to Proposition D.4, this condition is satisfied provided $\psi_i$ is $s_i$-concave for some $s_i > -1$, which holds by Proposition 3.5 and Assumption 3.4, with $s_i = c_i - 1$.

(2) Assumption H.2 holds uniformly in $\boldsymbol{\theta} \in \mathbb{S}_{N-1}$ because the design points are drawn from an i.i.d. elliptically symmetric distribution with density generator $g > 0$ and then $f_{\mathbf{p}} > 0$ on any bounded interval. Thus, $\inf_{\boldsymbol{\theta} \in \mathbb{S}_{N-1}} \inf_{u \in \mathcal{U}} f_{\boldsymbol{\theta}^\top \mathbf{p}}(u) \geq C_{\mathscr{D}}$ for some $C_{\mathscr{D}} > 0$.

(3) Assumption H.3 holds because $y_i^{(t)}$ are sub-gaussian (indeed $\psi_i$ is bounded and $\varepsilon_i^{(t)}$ are sub-gaussian) and $\psi_{i,\boldsymbol{\theta}}$ is bounded uniformly in $\boldsymbol{\theta} \in \mathbb{S}_{N-1}$ by Proposition D.4.

(4) We prove that Assumption H.4 holds uniformly in $\boldsymbol{\theta}$. Recall that $\phi_{i,\boldsymbol{\theta}} = d_{s_i} \circ \psi_{i,\boldsymbol{\theta}}$ (where $d_{s_i} \triangleq h_{s_i}^{-1}$ is defined in (4)). Then $\psi'_{i,\boldsymbol{\theta}}(u) = d'_{s_i}(\psi_{i,\boldsymbol{\theta}}(u))\psi'_{i,\boldsymbol{\theta}}(u)$. For every $u, v \in \mathcal{U}$ and $\boldsymbol{\theta} \in \mathbb{S}_{N-1}$ we have

$$|\psi'_{i,\boldsymbol{\theta}}(u) - \psi'_{i,\boldsymbol{\theta}}(v)| = |d'_{s_i}(\psi_{i,\boldsymbol{\theta}}(u))\psi'_{i,\boldsymbol{\theta}}(u) - d'_{s_i}(\psi_{i,\boldsymbol{\theta}}(v))\psi'_{i,\boldsymbol{\theta}}(v)|$$
$$\leq |d'_{s_i}(\psi_{i,\boldsymbol{\theta}}(u)) - d'_{s_i}(\psi_{i,\boldsymbol{\theta}}(v))||\psi'_{i,\boldsymbol{\theta}}(u)| + |\psi'_{i,\boldsymbol{\theta}}(u) - \psi'_{i,\boldsymbol{\theta}}(v)||d'_{s_i}(\psi_{i,\boldsymbol{\theta}}(v))|.$$

The first component $|d'_{s_i}(\psi_{i,\boldsymbol{\theta}}(u)) - d'_{s_i}(\psi_{i,\boldsymbol{\theta}}(v))| \leq L_{d'}|u - v|$ by Appendix H.1, where $L_{d'} = \sup_x d''_{s_i}(x)$ depends only on $s_i$ and bound of $\psi_i$. The component $|\psi'_{i,\boldsymbol{\theta}}(u)|$ is bounded uniformly by Proposition D.4 point *(4)*. The other component $|\psi'_{i,\boldsymbol{\theta}}(u) - \psi'_{i,\boldsymbol{\theta}}(v)| \leq L|u - v|$ for some $L > 0$ independent of $\boldsymbol{\theta}$ by Proposition D.4 point *(4)* and $|d'_{s_i}(\psi_{i,\boldsymbol{\theta}}(v))|$ is uniformly bounded by Appendix H.1. This proves that $\psi'_{i,\boldsymbol{\theta}}$ is Lipschitz uniformly in $\boldsymbol{\theta} \in \mathbb{S}_{N-1}$.

To prove (22) we apply the triangular inequality

$$|\widehat{\psi}_{i,\widetilde{\boldsymbol{\theta}}_i}(u) - \psi_i(u)| = |\widehat{\psi}_{i,\widetilde{\boldsymbol{\theta}}_i}(u) - \psi_{i,\widetilde{\boldsymbol{\theta}}_i}(u)| + |\psi_{i,\widetilde{\boldsymbol{\theta}}_i}(u) - \psi_i(u)|$$

and for the first component we apply the concentration inequality in (20) while for the second we use point *(3)* of Proposition D.4

$$|\psi_{i,\widetilde{\boldsymbol{\theta}}_i}(u) - \psi_i(u)| \lesssim \|\widetilde{\boldsymbol{\theta}}_i - \boldsymbol{\theta}_i\|_2$$

and then the concentration inequality in Proposition D.3, while retaining the dominating term $O((\log(m)/m)^{2/5})$. □

# E  PROOF OF PROPOSITION D.3

For simplicity, we redefine $\mathcal{T}_i^{(1)}$ as $\{1, 2, \ldots, n\}$, $\mathbf{x}^{(t)} = \mathbf{p}^{(t)} - \bar{\mathbf{p}}$. The loss function $\mathscr{L}_i(\boldsymbol{\theta})$ is defined as

$$\mathscr{L}_i(\boldsymbol{\theta}) = \frac{1}{n} \sum_{t=1}^{n} \left( y_i^{(t)} - \boldsymbol{\theta}^\top \mathbf{x}^{(t)} \right)^2.$$

Then the gradient and Hessian of $\mathscr{L}_i(\boldsymbol{\theta})$ is given by

$$\nabla_{\boldsymbol{\theta}} \mathscr{L}_i(\boldsymbol{\theta}) = \frac{1}{n} \sum_{t=1}^{n} 2 \left( \boldsymbol{\theta}^\top \mathbf{x}^{(t)} - y_i^{(t)} \right) \mathbf{x}^{(t)},$$

$$\nabla_{\boldsymbol{\theta}}^2 \mathscr{L}_i(\boldsymbol{\theta}) = \nabla_{\boldsymbol{\theta}}^2 \mathscr{L}_i = \frac{1}{n} \sum_{t=1}^{n} 2 \mathbf{x}^{(t)} \mathbf{x}^{(t)\top}.$$

Let $\widehat{\boldsymbol{\theta}}_i$ be the global minimizer of $\mathscr{L}_i(\boldsymbol{\theta})$. We do a Taylor expansion of $\mathscr{L}_i(\widehat{\boldsymbol{\theta}}_i)$ around $\lambda_i \boldsymbol{\theta}_i$, with $\lambda_i$ to be determined:

$$\mathscr{L}_i(\widehat{\boldsymbol{\theta}}_i) - \mathscr{L}_i(\lambda_i \boldsymbol{\theta}_i) = \left\langle \nabla \mathscr{L}_i(\lambda_i \boldsymbol{\theta}_i), \widehat{\boldsymbol{\theta}}_i - \lambda_i \boldsymbol{\theta}_i \right\rangle + \frac{1}{2} \left\langle \widehat{\boldsymbol{\theta}}_i - \lambda_i \boldsymbol{\theta}_i, \nabla_{\boldsymbol{\theta}}^2 \mathscr{L}_i \cdot (\widehat{\boldsymbol{\theta}}_i - \lambda_i \boldsymbol{\theta}_i) \right\rangle.$$

As $\widehat{\boldsymbol{\theta}}_i$ is the global minimizer of loss $\mathscr{L}_i(\boldsymbol{\theta})$, we have $\mathscr{L}_i(\widehat{\boldsymbol{\theta}}_i) \leq \mathscr{L}_i(\lambda_i \boldsymbol{\theta}_i)$, that is

$$\left\langle \nabla \mathscr{L}_i(\lambda_i \boldsymbol{\theta}_i), \widehat{\boldsymbol{\theta}}_i - \lambda_i \boldsymbol{\theta}_i \right\rangle + \frac{1}{2} \left\langle \widehat{\boldsymbol{\theta}}_i - \lambda_i \boldsymbol{\theta}_i, \nabla_{\boldsymbol{\theta}}^2 \mathscr{L}_i \cdot (\widehat{\boldsymbol{\theta}}_i - \lambda_i \boldsymbol{\theta}_i) \right\rangle \leq 0,$$

which implies

$$\left\langle \widehat{\boldsymbol{\theta}}_i - \lambda_i \boldsymbol{\theta}_i, \frac{1}{n} \sum_{t=1}^{n} \mathbf{x}^{(t)} \mathbf{x}^{(t)\top} (\widehat{\boldsymbol{\theta}}_i - \lambda_i \boldsymbol{\theta}_i) \right\rangle \leq \left\langle \nabla_{\boldsymbol{\theta}} \mathscr{L}_i(\lambda_i \boldsymbol{\theta}_i), \lambda_i \boldsymbol{\theta}_i - \widehat{\boldsymbol{\theta}}_i \right\rangle$$

$$\leq \left\| \nabla_{\boldsymbol{\theta}} \mathscr{L}_i(\lambda_i \boldsymbol{\theta}_i) \right\|_2 \left\| \lambda_i \boldsymbol{\theta}_i - \widehat{\boldsymbol{\theta}}_i \right\|_2$$

$$\leq \sqrt{N} \left\| \nabla_{\boldsymbol{\theta}} \mathscr{L}_i(\lambda_i \boldsymbol{\theta}_i) \right\|_\infty \left\| \lambda_i \boldsymbol{\theta}_i - \widehat{\boldsymbol{\theta}}_i \right\|_2.$$

By Lemma E.1, the LHS satisfies

$$\left\langle \widehat{\boldsymbol{\theta}}_i - \lambda_i \boldsymbol{\theta}_i, \frac{1}{n} \sum_{t=1}^{n} \mathbf{x}^{(t)} \mathbf{x}^{(t)\top} (\widehat{\boldsymbol{\theta}}_i - \lambda_i \boldsymbol{\theta}_i) \right\rangle \geq \frac{\varsigma_{\min}}{2} \left\| \lambda_i \boldsymbol{\theta}_i - \widehat{\boldsymbol{\theta}}_i \right\|_2^2,$$

w.p. at least $1 - 2e^{-c\varsigma_{\min}^2 n/16}$ (for $n$ sufficiently large) for certain constants $c > 0$ that depends only on the variance proxy $\sigma_x$ of the sub-Gaussian random variable $\mathbf{x}^{(t)}$, and by Lemma E.2 the RHS satisfies

$$\sqrt{N} \left\| \nabla_{\boldsymbol{\theta}} \mathscr{L}_i(\lambda_i \boldsymbol{\theta}_i) \right\|_\infty \left\| \lambda_i \boldsymbol{\theta}_i - \widehat{\boldsymbol{\theta}}_i \right\|_2 \leq \mathscr{B}_i^{**} \sqrt{\frac{N \log n}{n}} \left\| \lambda_i \boldsymbol{\theta}_i - \widehat{\boldsymbol{\theta}}_i \right\|_2,$$

with probability at least $1 - \frac{2N}{n}$ (as long as $n \geq N$), where $\mathscr{B}_i^{**}$ is a constant defined in (23) that depends solely on $\lambda_i$ and the variance proxies of the sub-Gaussian random variables $y_i^{(t)}$ and $\mathbf{x}^{(t)}$. Putting the two inequalities together we get with probability at least $1 - 2e^{-c\varsigma_{\min}^2 n/16} - \frac{2N}{n}$ that, for $n$ sufficiently large

$$\left\| \lambda_i \boldsymbol{\theta}_i - \widehat{\boldsymbol{\theta}}_i \right\|_2 \leq \mathscr{B}_i^* \sqrt{\frac{N \log n}{n}},$$

where $\mathscr{B}_i^* = 2 \frac{\mathscr{B}_i}{\varsigma_{\min}}$. Now we conclude. Recall that $\lambda_i > 0$ where $\lambda_i$ is defined in (25). Let

$$\mathcal{W}_n = \left\{ \boldsymbol{\theta} : \left\| \lambda_i \boldsymbol{\theta}_i - \boldsymbol{\theta} \right\|_2 \leq \mathscr{B}_i^* \sqrt{\frac{N \log n}{n}} \right\}.$$

Let $n$ be large enough so that $\mathscr{B}_i^* \sqrt{\frac{N \log n}{n}} \le \lambda_i$: this guarantees that any $\boldsymbol{\theta} \in \mathcal{W}_n$ is such that $\|\boldsymbol{\theta}\|_2 \ne 0$, which come from the fact that $\|\lambda_i \boldsymbol{\theta}_i\|_2 = \lambda_i$. Then for $\widehat{\boldsymbol{\theta}}_i \in \mathcal{W}_n$ we can write

$$
\begin{aligned}
\|\boldsymbol{\theta}_i - \widetilde{\boldsymbol{\theta}}_i\|_2 = \left\| \boldsymbol{\theta}_i - \frac{\widehat{\boldsymbol{\theta}}_i}{\|\widehat{\boldsymbol{\theta}}_i\|_2} \right\|_2 &= \left\| \frac{\lambda_i \boldsymbol{\theta}_i}{\lambda_i} - \frac{\widehat{\boldsymbol{\theta}}_i}{\|\widehat{\boldsymbol{\theta}}_i\|_2} \right\|_2 \\
&\le \left\| \frac{\lambda_i \boldsymbol{\theta}_i - \widehat{\boldsymbol{\theta}}_i}{\lambda_i} \right\|_2 + \left\| \widehat{\boldsymbol{\theta}}_i \left( \frac{1}{\lambda_i} - \frac{1}{\|\widehat{\boldsymbol{\theta}}_i\|_2} \right) \right\|_2 \\
&= \frac{\|\lambda_i \boldsymbol{\theta}_i - \widehat{\boldsymbol{\theta}}_i\|_2}{\lambda_i} + \|\widehat{\boldsymbol{\theta}}_i\|_2 \left| \frac{1}{\lambda_i} - \frac{1}{\|\widehat{\boldsymbol{\theta}}_i\|_2} \right| \\
&= \frac{\|\lambda_i \boldsymbol{\theta}_i - \widehat{\boldsymbol{\theta}}_i\|_2}{\lambda_i} + \|\widehat{\boldsymbol{\theta}}_i\|_2 \left| \frac{\lambda_i - \|\widehat{\boldsymbol{\theta}}_i\|_2}{\|\widehat{\boldsymbol{\theta}}_i\|_2 \lambda_i} \right| \\
&= \frac{\|\lambda_i \boldsymbol{\theta}_i - \widehat{\boldsymbol{\theta}}_i\|_2}{\lambda_i} + \frac{|\|\lambda_i \boldsymbol{\theta}_i\|_2 - \|\widehat{\boldsymbol{\theta}}_i\|_2|}{\lambda_i} \\
&\le 2 \frac{\|\lambda_i \boldsymbol{\theta}_i - \widehat{\boldsymbol{\theta}}_i\|_2}{\lambda_i},
\end{aligned}
$$

and, as $\widehat{\boldsymbol{\theta}}_i \in \mathcal{W}_n$ with probability at least $1 - 2e^{-c\varsigma_{\min}^2 n/16} - \frac{2N}{n}$, for $n$ sufficiently large, this implies that with the same probability bound

$$
\|\widetilde{\boldsymbol{\theta}}_i - \boldsymbol{\theta}_i\|_2 \le \mathscr{B}_i \sqrt{\frac{N \log n}{n}},
$$

where

$$
\mathscr{B}_i = 2 \frac{\mathscr{B}_i^*}{\lambda_i} = 4 \frac{\mathscr{B}_i^{**}}{\lambda_i \varsigma_{\min}} = 4\sqrt{2} \frac{\sigma_x (\lambda_i \sigma_x + \sigma_{y_i})}{\lambda_i \varsigma_{\min}}.
$$

This concludes the proof.

**Lemma E.1.** *There exist $c > 0$ such that with probability at least $1 - 2e^{-c\varsigma_{\min}^2 n/16}$ we have $(\varsigma_{\min}/2)\,\mathbb{I} \preccurlyeq \Sigma_n$ for $n$ sufficiently large, where*

$$
\Sigma_n \triangleq \frac{1}{n} \sum_{t=1}^{n} \mathbf{x}^{(t)} \mathbf{x}^{(t)\top}.
$$

*Proof of Lemma E.1.* To get a lower bound of the minimum eigenvalue of $\Sigma_n$ we use the following remark 5.40 of Vershynin (2010): let $A$ be a $n \times N$ matrix whose rows $A_i$ are independent sub-gaussian random vectors in $\mathbb{R}^N$ with second moment matrix $\Sigma$ (non necessarily isotropic), then there exist $c, C > 0$ depending only on the sub-gaussian norm $\sigma_x$, such that for every $r > 0$ we have $\|\frac{1}{n} \sum_{i=1}^{n} A_i A_i^\top - \Sigma\| \le \max\{\delta, \delta^2\}$ with probability at least $1 - 2e^{-cr^2}$, where $\delta = C\sqrt{\frac{N}{n}} + \frac{r}{\sqrt{n}}$. In our case, since $\Sigma_n$ is an average of $n$ i.i.d. random matrices with mean $\Sigma = \mathbb{E}\left[\mathbf{x}^{(t)} \mathbf{x}^{(t)\top}\right]$ and that $\{\mathbf{x}^{(t)}\}$ are sub-Gaussian random vectors (recall that $\mathbf{x}^{(t)}$ is sub-gaussian because the generator $g$ satisfies $g(x + y) = g(x)g(y)$ for all $x, y$), then there exist $c, C > 0$ such that for each $r > 0$ with probability at least $1 - 2e^{-cr^2}$,

$$
\|\Sigma_n - \Sigma\| \le \max\left\{\delta, \delta^2\right\}, \text{ where } \delta \triangleq C\sqrt{\frac{N}{n}} + \frac{r}{\sqrt{n}}.
$$

Here $c, C$ are both constants that are only related to $\sigma_x$ sub-Gaussian norm of $\mathbf{x}^{(t)}$. Now, choosing $r = \varsigma_{\min}\sqrt{n}/4$, then as long as $n \ge N16C^2/\varsigma_{\min}^2$ we have $\delta = C\sqrt{\frac{N}{n}} + \frac{\varsigma_{\min}}{4} \le \frac{\varsigma_{\min}}{4} + \frac{\varsigma_{\min}}{4} = \frac{\varsigma_{\min}}{2}$. Thus with probability at least $1 - 2e^{-c\varsigma_{\min}^2 n/16}$ we have

$$
\|\Sigma_n - \Sigma\| \le \max\left\{\delta, \delta^2\right\} \le \max\left\{\frac{\varsigma_{\min}}{2}, \frac{\varsigma_{\min}^2}{4}\right\},
$$

and since $\varsigma_{\min}\mathbb{I} \preccurlyeq \Sigma \preccurlyeq \varsigma_{\max}\mathbb{I}$ we get $\varsigma_{\min}\mathbb{I} - \Sigma_n \preccurlyeq \Sigma - \Sigma_n \preccurlyeq \varsigma_{\max}\mathbb{I} - \Sigma_n \preccurlyeq \max\left\{\varsigma_{\min}/2, \varsigma_{\min}^2/4\right\}$, and considering the first and last term we get

$$\left(\varsigma_{\min} + \max\left\{\frac{\varsigma_{\min}}{2}, \frac{\varsigma_{\min}^2}{4}\right\}\right)\mathbb{I} \preccurlyeq \Sigma_n,$$

and since $\varsigma_{\min} + \max\left\{\frac{\varsigma_{\min}}{2}, \frac{\varsigma_{\min}^2}{4}\right\} \geq \varsigma_{\min}/2$ we get $(\varsigma_{\min}/2)\mathbb{I} \preccurlyeq \Sigma_n$. $\qquad\square$

**Lemma E.2.** *With probability at least* $1 - \frac{2}{n}$, *for* $n$ *sufficiently large, we have*

$$\|\nabla_{\boldsymbol{\theta}}\mathscr{L}_i^{(n)}(\boldsymbol{\theta}_i)\|_\infty \leq \mathscr{B}_i^{**}\sqrt{\log(n)/n},$$

*where*

$$\mathscr{B}_i^{**} = \sqrt{2}\sigma_x\left(\lambda_i\sigma_x + \sigma_{y_i}\right) \tag{23}$$

*is a constant that depends solely on* $\lambda_i$ *(defined in (24)) and the variance proxies* $\sigma_{y_i}$ *and* $\sigma_x$ *of the sub-Gaussian random variables* $y_i^{(t)}$ *and* $\mathbf{x}^{(t)}$, *respectively.*

*Proof of Lemma E.2.* We first prove that for all $t = 1, 2, \ldots, n$,

$$\mathbb{E}\left[(\lambda_i\boldsymbol{\theta}_i^\top\mathbf{x}^{(t)} - y_i^{(t)})\mathbf{x}^{(t)}\right] = 0.$$

Using the known property of elliptically symmetric random variables (see, e.g. Li (1991, page 319, comment following Condition 3.1) or Balabdaoui et al. (2019, Section 9.1)) we have that

$$\mathbb{E}\left[\mathbf{x}^{(t)} \mid \boldsymbol{\theta}_i^\top\mathbf{x}^{(t)}\right] = \boldsymbol{\theta}_i^\top\mathbf{x}^{(t)}b,$$

where $b$ is a vector that has to satisfy $\mathbb{E}\left[\mathbf{x}^{(t)}\boldsymbol{\theta}_i^\top\mathbf{x}^{(t)}\right] = \text{var}(\boldsymbol{\theta}_i^\top\mathbf{x}^{(t)})b$ or equivalently (using $\text{var}(\mathbf{x}^{(t)}) = \Sigma$), $b = \Sigma\boldsymbol{\theta}_i(\boldsymbol{\theta}_i^\top\Sigma\boldsymbol{\theta}_i)^{-1}$. We have

$$\begin{aligned}
\mathbb{E}\left[\lambda_i\mathbf{x}^{(t)}\boldsymbol{\theta}_i^\top\mathbf{x}^{(t)} - \mathbf{x}^{(t)}y_i^{(t)}\right] &= \lambda_i\,\text{var}(\boldsymbol{\theta}_i^\top\mathbf{x}^{(t)})b - \mathbb{E}\left[y_i^{(t)}\mathbf{x}^{(t)}\right] \\
&= \lambda_i\,\text{var}(\boldsymbol{\theta}_i^\top\mathbf{x}^{(t)})b - \mathbb{E}\left[\mathbb{E}\left[y_i^{(t)}\mathbf{x}^{(t)} \mid \boldsymbol{\theta}_i^\top\mathbf{x}^{(t)}\right]\right] \\
&= \lambda_i\,\text{var}(\boldsymbol{\theta}_i^\top\mathbf{x}^{(t)})b - \mathbb{E}\left[\psi_i(\boldsymbol{\theta}_i^\top\mathbf{x}^{(t)})\mathbb{E}\left[\mathbf{x}^{(t)} \mid \boldsymbol{\theta}_i^\top\mathbf{x}^{(t)}\right]\right] \\
&= \lambda_i\,\text{var}(\boldsymbol{\theta}_i^\top\mathbf{x}^{(t)})b - \mathbb{E}\left[\psi_i(\boldsymbol{\theta}_i^\top\mathbf{x}^{(t)})\boldsymbol{\theta}_i^\top\mathbf{x}^{(t)}b\right] \\
&= b\left(\lambda_i\,\text{var}(\boldsymbol{\theta}_i^\top\mathbf{x}^{(t)}) - \text{cov}\left[\psi_i(\boldsymbol{\theta}_i^\top\mathbf{x}^{(t)}), \boldsymbol{\theta}_i^\top\mathbf{x}^{(t)}\right]\right),
\end{aligned}$$

which is equal to 0 as long as we choose

$$\lambda_i \triangleq \frac{\text{cov}\left[\psi_i(\boldsymbol{\theta}_i^\top\mathbf{x}^{(t)}), \boldsymbol{\theta}_i^\top\mathbf{x}^{(t)}\right]}{\text{var}(\boldsymbol{\theta}_i^\top\mathbf{x}^{(t)})}. \tag{24}$$

Note that $\lambda_i > 0$: indeed, as $\psi_i$ is increasing, it implies that, given $\mathbf{z}^{(t)}$ independent copy of $\mathbf{x}^{(t)}$

$$0 < \mathbb{E}\left[\left(\psi_i\left(\boldsymbol{\theta}_i^\top\mathbf{z}^{(t)}\right) - \psi_i\left(\boldsymbol{\theta}_i^\top\mathbf{x}^{(t)}\right)\right)\left(\boldsymbol{\theta}_i^\top\mathbf{z}^{(t)} - \boldsymbol{\theta}_i^\top\mathbf{x}^{(t)}\right)\right] = 2\,\text{cov}\left(\psi_i\left(\boldsymbol{\theta}_i^\top\mathbf{x}^{(t)}\right), \boldsymbol{\theta}_i^\top\mathbf{x}^{(t)}\right). \tag{25}$$

Thus, $\mathbb{E}\left[\nabla_{\boldsymbol{\theta}}\mathscr{L}_i(\lambda_i\boldsymbol{\theta}_i)\right] = 0$, then every entry of $\nabla_{\boldsymbol{\theta}}\mathscr{L}_i(\lambda_i\boldsymbol{\theta}_i)$ is mean zero, i.e.

$$\mathbb{E}\left[(\lambda_i\boldsymbol{\theta}_i^\top\mathbf{x}^{(t)} - y_i^{(t)})\mathbf{x}_i^{(t)}\right] = 0, \quad i = 1, \ldots, N.$$

Now, since $y_i^{(t)}$ is sub-gaussian (because $\varepsilon_i^{(t)}$ is sub-gaussian) with variance proxy denoted as $\sigma_{y_i}$ and $\mathbf{x}^{(t)}$ is sub-gaussian (because the generator $g$ satisfies $g(x + y) = g(x)g(y)$ for all $x, y$) with variance proxy denoted as $\sigma_x$, $\lambda_i\boldsymbol{\theta}_i^\top\mathbf{x}^{(t)} - y_i^{(t)}$ is sub-gaussian, and given that $\mathbf{x}_i^{(t)}$ is sub-gaussian, the product $(\lambda_i\boldsymbol{\theta}_i^\top\mathbf{x}^{(t)} - y_i^{(t)})\mathbf{x}_i^{(t)}$ is sub-exponential. More specifically

$$\begin{aligned}
\|(\lambda_i\boldsymbol{\theta}_i^\top\mathbf{x}^{(t)} - y_i^{(t)})\mathbf{x}_i^{(t)}\|_{\psi_1} &\leq \|\lambda_i\boldsymbol{\theta}_i^\top\mathbf{x}^{(t)} - y_i^{(t)}\|_{\psi_2}\|\mathbf{x}_i^{(t)}\|_{\psi_2} \\
&\leq (\lambda_i\|\boldsymbol{\theta}_i^\top\mathbf{x}^{(t)}\|_{\psi_2} + \|y_i^{(t)}\|_{\psi_2})\|\mathbf{x}_i^{(t)}\|_{\psi_2} \\
&\leq (\lambda_i\sigma_x + \sigma_{y_i})\sigma_x \triangleq \nu_i, \tag{26}
\end{aligned}$$

where $\| \cdot \|_{\psi_p}$ is the Orlicz norm associated to $\psi_p(x) = e^{x^p} - 1$, with $p \geq 1$. For $p = 1$, $\|X\|_{\psi_1} < \infty$ is equivalent to $X$ belonging to the class of sub-exponential distributions, while with $p = 2$, $\|X\|_{\psi_2} < \infty$ is equivalent to $X$ belonging to the class of sub-gaussian distributions. Then $(\lambda_i \boldsymbol{\theta}_i^\top \mathbf{x}^{(t)} - y_i^{(t)}) \mathbf{x}_i^{(t)} \sim \mathrm{SE}(\nu_i^2, \nu_i)$ for $i = 1, \ldots, N$, that is

$$\mathbb{P}\left( \left| 2 \left( \lambda_i \boldsymbol{\theta}_i^\top \mathbf{x}^{(t)} - y_i^{(t)} \right) \mathbf{x}_i^{(t)} \right| \geq u \right) \leq 2 \exp\left( -\frac{1}{2} \min\left\{ \frac{u^2}{\nu_i^2}, \frac{u}{\nu_i} \right\} \right).$$

Now, using that $X_t = 2(\lambda_i \boldsymbol{\theta}_i^\top \mathbf{x}^{(t)} - y_i^{(t)}) \mathbf{x}_i^{(t)} \sim \mathrm{SE}(\nu_i^2, \nu_i)$ are independent for all $t$, with $\mathbb{E}(X_t) = 0$ we get

$$\mathbb{P}\left( \left\| \nabla_{\boldsymbol{\theta}} \mathscr{L}_i^{(n)}(\boldsymbol{\theta}_i) \right\|_\infty \geq u \right) = \mathbb{P}\left( \bigcup_{i=1}^N \left| \frac{1}{n} \sum_{t=1}^n 2(\lambda_i \boldsymbol{\theta}_i^\top \mathbf{x}^{(t)} - y_i^{(t)}) \mathbf{x}_i^{(t)} \right| \geq u \right)$$

$$\leq N\mathbb{P}\left( \left| \frac{1}{n} \sum_{t=1}^n 2(\lambda_i \boldsymbol{\theta}_i^\top \mathbf{x}^{(t)} - y_i^{(t)}) \mathbf{x}_i^{(t)} \right| \geq u \right)$$

$$\leq 2N \exp\left( -\frac{n}{2} \min\left\{ \frac{u^2}{\nu_i^2}, \frac{u}{\nu_i} \right\} \right),$$

By taking $u = \sqrt{2} \nu_i \sqrt{\log n / n}$, for sufficiently large $n$ we have that

$$\min\left\{ \frac{u^2}{\nu_i^2}, \frac{u}{\nu_i} \right\} = \frac{u^2}{\nu_i^2},$$

and then

$$\left\| \nabla_{\boldsymbol{\theta}} \mathscr{L}(\boldsymbol{\theta}_i) \right\|_\infty \leq \sqrt{2} \nu_i \sqrt{\frac{\log n}{n}} = \mathscr{B}_i^{**} \sqrt{\frac{\log n}{n}},$$

with probability at least $1 - 2Ne^{-\log(n)} = 1 - 2\frac{N}{n}$. $\qquad\square$

# F    PROOF OF PROPOSITION D.4

## F.1    PROOF OF *(1)*

We have $\mathbf{p} \sim \mathscr{E}_N(\mathbf{m}, \Lambda, g)$ be an elliptically symmetric random vector with density

$$f_{\mathbf{p}}(\mathbf{p}) = k\,g((\mathbf{p} - \mathbf{m})^\top \Lambda^{-1}(\mathbf{p} - \mathbf{m})),$$

where $\mathbf{m}$ is the location vector, $\Lambda$ is a positive-definite scatter matrix, $g$ is the density generator, and $k$ is a normalizing constant. We wish to find the conditional density of $\mathbf{p}$ given $\boldsymbol{\theta}^\top \mathbf{p} = u$, where $\boldsymbol{\theta} \in \mathbb{R}^N$ is a fixed vector. The constraint defines a hyperplane:

$$H_u = \{\mathbf{p} \in \mathbb{R}^N : \boldsymbol{\theta}^\top \mathbf{p} = u\}.$$

The conditional density satisfies:

$$f_{\mathbf{p}|\boldsymbol{\theta}^\top \mathbf{p}=u}(\mathbf{p}) \propto f_{\mathbf{p}}(\mathbf{p}), \quad \mathbf{p} \in H_u,$$

that is,

$$f_{\mathbf{p}|\boldsymbol{\theta}^\top \mathbf{p}=u}(\mathbf{p}) = \frac{1}{Z(u)}\,g((\mathbf{p} - \mathbf{m})^\top \Lambda^{-1}(\mathbf{p} - \mathbf{m})), \quad \text{for } \mathbf{p} \text{ such that } \boldsymbol{\theta}^\top \mathbf{p} = u,$$

where $Z(u)$ is the normalizing constant. Define the transformation:

$$\mathbf{y} = A^{-1}(\mathbf{p} - \mathbf{m}),$$

where $A$ satisfies $\Lambda = AA^\top$. Then, since $(\mathbf{p} - \mathbf{m})^\top \Lambda^{-1}(\mathbf{p} - \mathbf{m}) = \mathbf{y}^\top \mathbf{y}$, the density becomes:

$$f_{\mathbf{y}}(\mathbf{y}) \propto g(\mathbf{y}^\top \mathbf{y}).$$

The constraint transforms as:

$$u = \boldsymbol{\theta}^\top \mathbf{p} = \boldsymbol{\theta}^\top (\mathbf{m} + A\mathbf{y}) = \boldsymbol{\theta}^\top \mathbf{m} + (A^\top \boldsymbol{\theta})^\top \mathbf{y}.$$

Define $\boldsymbol{v} = A^\top \boldsymbol{\theta}$, so that the constraint simplifies to:

$$\boldsymbol{v}^\top \mathbf{y} = u - \boldsymbol{\theta}^\top \mathbf{m}. \tag{27}$$

Decomposing $\mathbf{y}$ into components parallel and perpendicular to $\boldsymbol{v}$, we write:

$$\mathbf{y} = \frac{\boldsymbol{v}}{\|\boldsymbol{v}\|_2} y_1 + \boldsymbol{\alpha}, \quad \text{with } \boldsymbol{v}^\top \boldsymbol{\alpha} = 0. \tag{28}$$

Then $\mathbf{y}^\top \mathbf{y} = y_1^2 + \|\boldsymbol{\alpha}\|_2^2$. From the constraint in Equation (27):

$$u - \boldsymbol{\theta}^\top \mathbf{m} = \boldsymbol{v}^\top \left( \frac{\boldsymbol{v}}{\|\boldsymbol{v}\|_2} y_1 + \boldsymbol{\alpha} \right) \quad \Rightarrow \quad u - \boldsymbol{\theta}^\top \mathbf{m} = \|\boldsymbol{v}\|_2 y_1 \quad \Rightarrow \quad y_1 = \frac{u - \boldsymbol{\theta}^\top \mathbf{m}}{\|\boldsymbol{v}\|_2}.$$

Since the original density in the $\mathbf{y}$-space is

$$f_{\mathbf{y}}(\mathbf{y}) \propto g(y_1^2 + \|\boldsymbol{\alpha}\|_2^2),$$

when conditioning on $y_1$, the conditional density on the $(N-1)$-dimensional space of $\boldsymbol{\alpha}$ is:

$$f_{\boldsymbol{\alpha}|\boldsymbol{v}^\top \mathbf{y}=u-\boldsymbol{\theta}^\top \mathbf{m}}(\boldsymbol{\alpha}) \propto g\left( \left( \frac{u - \boldsymbol{\theta}^\top \mathbf{m}}{\|\boldsymbol{v}\|_2} \right)^2 + \|\boldsymbol{\alpha}\|_2^2 \right).$$

Now, using the assumption that $g(x + y) = g(x)g(y)$ we have that

$$f_{\boldsymbol{\alpha}|\boldsymbol{v}^\top \mathbf{y}=u-\boldsymbol{\theta}^\top \mathbf{m}}(\boldsymbol{\alpha}) \propto g(\|\boldsymbol{\alpha}\|_2^2). \tag{29}$$

Now, recall that

$$\mathbf{p} = \mathbf{m} + A\mathbf{y} = \mathbf{m} + A\left( \frac{\boldsymbol{v}}{\|\boldsymbol{v}\|_2} \frac{u - \boldsymbol{\theta}^\top \mathbf{m}}{\|\boldsymbol{v}\|_2} + \boldsymbol{\alpha} \right) = \mathbf{m}_u + A\boldsymbol{\alpha}, \tag{30}$$

where we defined the conditional center (the "shift")

$$\mathbf{m}_u = \mathbf{m} + A \frac{\boldsymbol{v}}{\|\boldsymbol{v}\|_2} \frac{u - \boldsymbol{\theta}^\top \mathbf{m}}{\|\boldsymbol{v}\|_2} = \mathbf{m} + \Lambda \boldsymbol{\theta} \frac{(u - \boldsymbol{\theta}^\top \mathbf{m})}{\boldsymbol{\theta}^\top \Lambda \boldsymbol{\theta}}. \tag{31}$$

Thus,

$$
\begin{aligned}
\psi_{i,\boldsymbol{\theta}}(u) &= \mathbb{E}_{(\varepsilon_i, \mathbf{p})}(y_i \mid w_i = u) \\
&= \mathbb{E}_{\mathbf{p}}(\mathbb{E}_{\varepsilon_i}(y_i \mid \mathbf{p}) \mid w_i = u) \\
&= \mathbb{E}_{\mathbf{p}}(\psi_i(\boldsymbol{\theta}_i^\top \mathbf{p}) \mid w_i = u) \\
&= \mathbb{E}_{\mathbf{p}}\left(\psi_i\left(\boldsymbol{\theta}_i^\top \mathbf{p}\right) \mid \mathbf{p}^\top \boldsymbol{\theta} = u\right) \\
&\overset{(30)}{=} \mathbb{E}_{\boldsymbol{\alpha}}\left(\psi_i\left(\boldsymbol{\theta}_i^\top \mathbf{m}_u + \boldsymbol{\theta}_i^\top A\boldsymbol{\alpha}\right) \mid \boldsymbol{v}^\top \mathbf{y} = u - \boldsymbol{\theta}^\top \mathbf{m}\right) \\
&\overset{(29)}{=} \int_{\mathbb{R}^{N-1}} \psi_i\left(\boldsymbol{\theta}_i^\top \mathbf{m}_u + \boldsymbol{\theta}_i^\top A\boldsymbol{\alpha}\right) \frac{g(\|\boldsymbol{\alpha}\|_2^2)}{\int_{\mathbb{R}^{N-1}} g(\|\boldsymbol{\delta}\|_2^2)d\boldsymbol{\delta}} d\boldsymbol{\alpha}
\end{aligned}
\tag{32}
$$

This completes the proof of *(1)*.

### F.2 PROOF OF *(2)*

We first define

$$c(\boldsymbol{\theta}) \triangleq \frac{d}{du}\boldsymbol{\theta}_i^\top \mathbf{m}_u = \frac{\boldsymbol{\theta}_i^\top \Lambda \boldsymbol{\theta}}{\boldsymbol{\theta}^\top \Lambda \boldsymbol{\theta}}, \tag{33}$$

where we recall from Equation (31) that $\mathbf{m}_u = \mathbf{m} + \Lambda \boldsymbol{\theta} \frac{(u - \boldsymbol{\theta}^\top \mathbf{m})}{\boldsymbol{\theta}^\top \Lambda \boldsymbol{\theta}}$. Note that

$$|c(\boldsymbol{\theta})| \le \frac{c_{\min}}{c_{\max}} \sum_{j \in \mathcal{N}} |\boldsymbol{\theta}_{i,j}\boldsymbol{\theta}_j| \le \frac{c_{\min}}{c_{\max}} \|\boldsymbol{\theta}_i\|_2 \|\boldsymbol{\theta}\|_2 \le \frac{c_{\min}}{c_{\max}}, \tag{34}$$

where we used that $\boldsymbol{\theta}_i, \boldsymbol{\theta} \in \mathbb{S}_{N-1}$. By (32) we have

$$\psi_{i,\boldsymbol{\theta}}(u) = \int_{\mathbb{R}^{N-1}} \psi_i\left(\boldsymbol{\theta}_i^\top \mathbf{m}_u + \boldsymbol{\theta}_i^\top A\boldsymbol{\alpha}\right) \tilde{g}(\boldsymbol{\alpha})d\boldsymbol{\alpha}, \quad u \in \mathcal{U}, \ \boldsymbol{\theta} \in \mathbb{S}_{N-1},$$

where $\tilde{g}(\boldsymbol{\alpha}) = \frac{g(\|\boldsymbol{\alpha}\|_2^2)}{\int_{\mathbb{R}^{N-1}} g(\|\boldsymbol{\delta}\|_2^2)d\boldsymbol{\delta}}$ is independent of $u$. Then, for $k = 0, 1, 2$ we can write

$$\psi_{i,\boldsymbol{\theta}}^{(k)}(u) = c(\boldsymbol{\theta})^k \int_{\mathbb{R}^{N-1}} \psi_i^{(k)}\left(\boldsymbol{\theta}_i^\top \mathbf{m}_u + \boldsymbol{\theta}_i^\top A\boldsymbol{\alpha}\right) \tilde{g}(\boldsymbol{\alpha})d\boldsymbol{\alpha}, \tag{35}$$

proving that $\psi_{i,\boldsymbol{\theta}} \in \mathrm{C}^2(\mathcal{U})$ for all $\boldsymbol{\theta}$. By Assumption 3.1 we have that $0 < \underline{B}_{\psi_i} \le \psi_i \le \bar{B}_{\psi_i}$, which immediately implies that $0 < \underline{B}_{\psi_i} \le \psi_{i,\boldsymbol{\theta}} \le \bar{B}_{\psi_i}$ uniformly in $\boldsymbol{\theta}$, proving point *(2)*.

### F.3 PROOF OF *(3)*

We can write

$$
\begin{aligned}
|\psi_{i,\boldsymbol{\theta}}(u) - \psi_i(u)| &= \int_{\mathbb{R}^{N-1}} |\psi_i\left(\boldsymbol{\theta}_i^\top \mathbf{m}_u + \boldsymbol{\theta}_i^\top A\boldsymbol{\alpha}\right) - \psi_i(u)|\, \tilde{g}(\boldsymbol{\alpha})d\boldsymbol{\alpha} \\
&\le \int_{\mathbb{R}^{N-1}} |\boldsymbol{\theta}_i^\top \mathbf{m}_u + \boldsymbol{\theta}_i^\top A\boldsymbol{\alpha} - u|\tilde{g}(\boldsymbol{\alpha})d\boldsymbol{\alpha}.
\end{aligned}
$$

Note that

$$|\boldsymbol{\theta}_i^\top \mathbf{m}_u + \boldsymbol{\theta}_i^\top A\boldsymbol{\alpha} - u| = \left| \boldsymbol{\theta}_i \left( \mathbf{m} + \Lambda\boldsymbol{\theta}\frac{(u - \boldsymbol{\theta}^\top\mathbf{m})}{\boldsymbol{\theta}^\top\Lambda\boldsymbol{\theta}} \right) + \boldsymbol{\theta}_i^\top A\boldsymbol{\alpha} - u \right|$$

$$\stackrel{\boldsymbol{\theta}^\top A\boldsymbol{\alpha}=0 \text{ by } Equation\ (28)}{=} \left| \boldsymbol{\theta}_i \left( \mathbf{m} + \Lambda\boldsymbol{\theta}\frac{(u - \boldsymbol{\theta}^\top\mathbf{m})}{\boldsymbol{\theta}^\top\Lambda\boldsymbol{\theta}} \right) + (\boldsymbol{\theta}_i - \boldsymbol{\theta})^\top A\boldsymbol{\alpha} - u \right|$$

$$\lesssim \|\boldsymbol{\theta}_i - \boldsymbol{\theta}\|_2 \|A\boldsymbol{\alpha}\|_2 + \|\mathbf{m}\|_2 \left\| \boldsymbol{\theta}_i - \frac{\boldsymbol{\theta}_i^\top\Lambda\boldsymbol{\theta}}{\boldsymbol{\theta}^\top\Lambda\boldsymbol{\theta}}\boldsymbol{\theta} \right\|_2 + |u| \left\| \frac{\boldsymbol{\theta}_i^\top\Lambda\boldsymbol{\theta}}{\boldsymbol{\theta}^\top\Lambda\boldsymbol{\theta}} - 1 \right\|_2$$

$$\lesssim \|\boldsymbol{\theta}_i - \boldsymbol{\theta}\|_2 \|\boldsymbol{\alpha}\|_2 + \left\| \boldsymbol{\theta}_i - \boldsymbol{\theta} + \boldsymbol{\theta}\left(1 - \frac{\boldsymbol{\theta}_i^\top\Lambda\boldsymbol{\theta}}{\boldsymbol{\theta}^\top\Lambda\boldsymbol{\theta}}\right) \right\|_2 + \left\| \frac{\boldsymbol{\theta}_i^\top\Lambda\boldsymbol{\theta}}{\boldsymbol{\theta}^\top\Lambda\boldsymbol{\theta}} - 1 \right\|_2$$

$$\lesssim \|\boldsymbol{\theta}_i - \boldsymbol{\theta}\|_2 \|\boldsymbol{\alpha}\|_2 + \|\boldsymbol{\theta}\|_2 \left\| 1 - \frac{\boldsymbol{\theta}_i^\top\Lambda\boldsymbol{\theta}}{\boldsymbol{\theta}^\top\Lambda\boldsymbol{\theta}} \right\|_2 + \left\| \frac{\boldsymbol{\theta}_i^\top\Lambda\boldsymbol{\theta}}{\boldsymbol{\theta}^\top\Lambda\boldsymbol{\theta}} - 1 \right\|_2$$

$$\lesssim \|\boldsymbol{\theta}_i - \boldsymbol{\theta}\|_2 \|\boldsymbol{\alpha}\|_2 + \left\| \frac{(\boldsymbol{\theta}_i - \boldsymbol{\theta})^\top\Lambda\boldsymbol{\theta}}{\boldsymbol{\theta}\Lambda\boldsymbol{\theta}} \right\|_2$$

$$\lesssim \|\boldsymbol{\theta}_i - \boldsymbol{\theta}\|_2 \|\boldsymbol{\alpha}\|_2 + \|\boldsymbol{\theta}_i - \boldsymbol{\theta}\|_2 \left\| \frac{\Lambda\boldsymbol{\theta}}{\boldsymbol{\theta}^\top\Lambda\boldsymbol{\theta}} \right\|_2$$

$$\lesssim \|\boldsymbol{\theta}_i - \boldsymbol{\theta}\|_2 \|\boldsymbol{\alpha}\|_2 + \|\boldsymbol{\theta}_i - \boldsymbol{\theta}\|_2 \left\| \frac{\Lambda\boldsymbol{\theta}}{\boldsymbol{\theta}^\top\Lambda\boldsymbol{\theta}} \right\|_2$$

$$\lesssim \|\boldsymbol{\theta}_i - \boldsymbol{\theta}\|_2 \|\boldsymbol{\alpha}\|_2 + \|\boldsymbol{\theta}_i - \boldsymbol{\theta}\|_2 \frac{c_{\max}\|\boldsymbol{\theta}\|_2}{c_{\min}\|\boldsymbol{\theta}\|_2}$$

$$\lesssim \|\boldsymbol{\theta}_i - \boldsymbol{\theta}\|_2 (c + \|\boldsymbol{\alpha}\|_2),$$

for some $c > 0$, where in the second inequality we used that $u \in \mathcal{U}$, which is a bounded set, and we also used, at different points, that $\|\boldsymbol{\theta}_i\|_2 = \|\boldsymbol{\theta}\|_2 = 1$. We then have

$$|\psi_{i,\boldsymbol{\theta}}(u) - \psi_i(u)| \leq \int_{\mathbb{R}^{N-1}} |\boldsymbol{\theta}_i^\top\mathbf{m}_u + \boldsymbol{\theta}_i^\top A\boldsymbol{\alpha} - u|\tilde{g}(\boldsymbol{\alpha})d\boldsymbol{\alpha}$$

$$\lesssim \|\boldsymbol{\theta}_i - \boldsymbol{\theta}\|_2 \left( c + \int_{\mathbb{R}^{N-1}} \|\boldsymbol{\alpha}\|_2\tilde{g}(\boldsymbol{\alpha})d\boldsymbol{\alpha} \right) \lesssim \|\boldsymbol{\theta}_i - \boldsymbol{\theta}\|_2,$$

where we have used that

$$\int_{\mathbb{R}^{N-1}} \|\boldsymbol{\alpha}\|_2\tilde{g}(\boldsymbol{\alpha})d\boldsymbol{\alpha} = \frac{1}{\int_{\mathbb{R}^{N-1}} g(\|\boldsymbol{\delta}\|_2^2)d\boldsymbol{\delta}} \int_{\mathbb{R}^{N-1}} \|\boldsymbol{\alpha}\|_2 g(\|\boldsymbol{\alpha}\|_2^2)d\boldsymbol{\alpha},$$

is finite because since $g(x + y) = g(x)g(y)$, then $g$ has exponential decay.

### F.4 PROOF OF *(4)*

From (35) we have

$$\psi'_{i,\boldsymbol{\theta}}(u) = c(\boldsymbol{\theta}) \int_{\mathbb{R}^{N-1}} \psi'_i \left( \boldsymbol{\theta}_i^\top\mathbf{m}_u + \boldsymbol{\theta}_i^\top A\boldsymbol{\alpha} \right) \tilde{g}(\boldsymbol{\alpha})d\boldsymbol{\alpha}$$

where $c(\boldsymbol{\theta}) \triangleq \frac{d}{du}\boldsymbol{\theta}_i^\top\mathbf{m}_u = \frac{\boldsymbol{\theta}_i^\top\Lambda\boldsymbol{\theta}}{\boldsymbol{\theta}^\top\Lambda\boldsymbol{\theta}}$ as defined in (33). Then we only need to prove that $\boldsymbol{\theta}_i^\top\Lambda\boldsymbol{\theta} > 0$ for $\boldsymbol{\theta}$ sufficiently close to $\boldsymbol{\theta}_i$. Recall by Assumption 5.1 that $\Lambda \in \mathbb{R}^{N \times N}$ is symmetric positive definite matrix with $c_{\min}\mathbb{I} \preccurlyeq \Lambda \preccurlyeq c_{\max}\mathbb{I}$ for some $0 < c_{\min} \leq c_{\max} < \infty$, and that $\|\boldsymbol{\theta}\|_2 = \|\boldsymbol{\theta}_i\|_2 = 1$. We are interested in the conditions under which

$$\boldsymbol{\theta}^\top\Lambda\boldsymbol{\theta}_i > 0.$$

We begin by writing:

$$\boldsymbol{\theta}^\top\Lambda\boldsymbol{\theta}_i = \boldsymbol{\theta}_i^\top\Lambda\boldsymbol{\theta}_i + (\boldsymbol{\theta} - \boldsymbol{\theta}_i)^\top\Lambda\boldsymbol{\theta}_i.$$

Let $a := \boldsymbol{\theta}_i^\top\Lambda\boldsymbol{\theta}_i$. Since $\Lambda$ is positive definite and $\|\boldsymbol{\theta}_i\| = 1$, we have

$$a \geq c_{\min} > 0.$$

Next, we bound the second term using the Cauchy–Schwarz inequality:

$$|(\boldsymbol{\theta} - \boldsymbol{\theta}_i)^\top \Lambda \boldsymbol{\theta}_i| \leq \|\boldsymbol{\theta} - \boldsymbol{\theta}_i\|_2 \cdot \|\Lambda \boldsymbol{\theta}_i\|_2.$$

Moreover,

$$\|\Lambda \boldsymbol{\theta}_i\|_2 \leq c_{\max} \|\boldsymbol{\theta}_i\|_2 = c_{\max}.$$

Therefore,

$$\boldsymbol{\theta}^\top \Lambda \boldsymbol{\theta}_i \geq a - c_{\max} \|\boldsymbol{\theta} - \boldsymbol{\theta}_i\|_2.$$

To ensure that $\boldsymbol{\theta}^\top \Lambda \boldsymbol{\theta}_i > 0$, it suffices to require:

$$\|\boldsymbol{\theta} - \boldsymbol{\theta}_i\|_2 < \frac{a}{c_{\max}}.$$

We now prove that $|\psi'_{i,\boldsymbol{\theta}}| \leq |c(\boldsymbol{\theta})| \bar{B}_{\psi_i} \overset{(34)}{\leq} \frac{c_{\min}}{c_{\max}} \bar{B}_{\psi_i}$. Lastly, we prove that $\psi'_{i,\boldsymbol{\theta}}$ is Lipschitz uniformly in $\boldsymbol{\theta} \in \mathbb{S}_{N-1}$. For all $u, v \in \mathcal{U}$, and $\boldsymbol{\theta} \in \mathbb{S}_{N-1}$, we have

$$
\begin{aligned}
&|\psi'_{i,\boldsymbol{\theta}}(u) - \psi'_{i,\boldsymbol{\theta}}(v)| \\
&= |c(\boldsymbol{\theta})| \left| \int_{\mathbb{R}^{N-1}} \psi'_i \left( \boldsymbol{\theta}_i^\top \mathbf{m}_u + \boldsymbol{\theta}_i^\top A\boldsymbol{\alpha} \right) \tilde{g}(\boldsymbol{\alpha}) d\boldsymbol{\alpha} - \int_{\mathbb{R}^{N-1}} \psi'_i \left( \boldsymbol{\theta}_i^\top \mathbf{m}_v + \boldsymbol{\theta}_i^\top A\boldsymbol{\alpha} \right) \tilde{g}(\boldsymbol{\alpha}) d\boldsymbol{\alpha} \right| \\
&\overset{Assumption\ 3.1}{\leq} B_{\psi''_i} |c(\boldsymbol{\theta})| \left| \boldsymbol{\theta}_i^\top \mathbf{m}_u - \boldsymbol{\theta}_i^\top \mathbf{m}_v \right| \\
&\overset{(34)}{\leq} B_{\psi''_i} \frac{c_{\min}}{c_{\max}} \left| \boldsymbol{\theta}_i^\top (\mathbf{m}_u - \mathbf{m}_v) \right| \\
&\leq B_{\psi''_i} \frac{c_{\min}}{c_{\max}} \|\mathbf{m}_u - \mathbf{m}_v\|_2 \\
&= B_{\psi''_i} \frac{c_{\min}}{c_{\max}} \left\| \left( \mathbf{m} + \Lambda \boldsymbol{\theta} \frac{(u - \boldsymbol{\theta}^\top \mathbf{m})}{\boldsymbol{\theta}^\top \Lambda \boldsymbol{\theta}} \right) - \left( \mathbf{m} + \Lambda \boldsymbol{\theta} \frac{(v - \boldsymbol{\theta}^\top \mathbf{m})}{\boldsymbol{\theta}^\top \Lambda \boldsymbol{\theta}} \right) \right\|_2 \\
&= B_{\psi''_i} \frac{c_{\min}}{c_{\max}} \frac{\|\Lambda \boldsymbol{\theta}\|}{\boldsymbol{\theta}^\top \Lambda \boldsymbol{\theta}} |u - v| \leq B_{\psi''_i} \frac{c_{\min}^2}{c_{\max}^2} |u - v|.
\end{aligned}
$$

## F.5 PROOF OF (5)

Recall that

$$\psi_{i,\boldsymbol{\theta}}(u) = \int_{\mathbb{R}^{N-1}} \psi_i \left( \boldsymbol{\theta}_i^\top \mathbf{m}_u + \boldsymbol{\theta}_i^\top A\boldsymbol{\alpha} \right) \tilde{g}(\boldsymbol{\alpha}) d\boldsymbol{\alpha} = \int_{\mathbb{R}^{N-1}} \psi_i \left( \xi(\boldsymbol{\alpha}, u) \right) \tilde{g}(\boldsymbol{\alpha}) d\boldsymbol{\alpha},$$

where

$$\xi(\boldsymbol{\alpha}, u) = \boldsymbol{\theta}_i^\top \mathbf{m}_u + \boldsymbol{\theta}_i^\top A\boldsymbol{\alpha}.$$

First note that if $\boldsymbol{\theta} = \boldsymbol{\theta}_i$, we have $\xi(\boldsymbol{\alpha}, u) = u$, since $\boldsymbol{\theta}^\top A\boldsymbol{\alpha} = 0$, and then $\psi_{i,\boldsymbol{\theta}_i} = \psi_i$. Now suppose $\boldsymbol{\theta}_i \neq \boldsymbol{\theta}$. More generally, we will prove that for non-negative functions $h, f_1, f_2$ integrable in $\mathbb{R}$ that satisfy

$$h((1 - \lambda)x + \lambda y) \geq M_s(f_1(x), f_2(y); 1 - \lambda, \lambda), \quad x, y \in \mathbb{R},$$

then

$$H((1 - \lambda)u_1 + \lambda u_2) \geq M_s(F_1(u_1), F_2(u_2); 1 - \lambda, \lambda), \quad u_1, u_2 \in \mathbb{R},$$

where

$$H(u) \triangleq \int h(\xi(\boldsymbol{\alpha}, u)) dP(\boldsymbol{\alpha})$$

$$F_j(u) \triangleq \int f_j(\xi(\boldsymbol{\alpha}, u)) dP(\boldsymbol{\alpha}), \quad j = 1, 2,$$

where $P$ is a probability measure.

**Lemma F.1.** *Let $0 < \lambda < 1$, $-1 < s \leq \infty$, and $f, g$, and let $h$ nonnegative be integrable functions on $\mathbb{R}$ satisfying*

$$h((1-\lambda)x + \lambda y) \geq M_s(f_1(x), f_2(y); 1 - \lambda, \lambda), \quad x, y \in \mathbb{R},$$

*For every fixed $u \in \mathbb{R}$ define $H, F_1$ and $F_2$ as*

$$H(u) \triangleq \int h(\xi(\boldsymbol{\alpha}, u)) dP(\boldsymbol{\alpha})$$

$$F_j(u) \triangleq \int f_j(\xi(\boldsymbol{\alpha}, u)) dP(\boldsymbol{\alpha}), \quad j = 1, 2,$$

*where $\xi(\boldsymbol{\alpha}, u) = \boldsymbol{\alpha}^\top \mathbf{b} + cu + a$ for a non zero vector $\mathbf{b}$, a nonzero real value $c$ and a constant $a$, and $P$ is a probability measure, then*

$$H((1-\lambda)u_1 + \lambda u_2) \geq M_{s/(s+1)}(F_1(u_1), F_2(u_2); 1 - \lambda, \lambda), \quad u_1, u_2 \in \mathbb{R},$$

*where we have defined*

$$M_s(x, y; \lambda, \eta) \triangleq \begin{cases} (\lambda x^s + \eta y^s)^{1/s}, & s \neq 0 \\ x^\lambda y^\eta, & s = 0. \end{cases}$$

*Proof.* We have

$$H((1-\lambda)u_1 + \lambda u_2) = \int h(\xi(\boldsymbol{\alpha}, (1-\lambda)u_1 + \lambda u_2)) dP(\boldsymbol{\alpha}).$$

We can write

$$\begin{aligned} \xi(\boldsymbol{\alpha}, (1-\lambda)u_1 + \lambda u_2) &= \boldsymbol{\alpha}^\top \mathbf{b} + a + c((1-\lambda)u_1 + \lambda u_2) \\ &= (1-\lambda)[cu_1 + a + \boldsymbol{\alpha}^\top \mathbf{b}] + \lambda[cu_2 + a + \boldsymbol{\alpha}^\top \mathbf{b}] \\ &= (1-\lambda)\xi(\boldsymbol{\alpha}, u_1) + \lambda \xi(\boldsymbol{\alpha}, u_2), \end{aligned}$$

and by assumption on $h$ we have that

$$h(\xi(\boldsymbol{\alpha}, (1-\lambda)u_1 + \lambda u_2)) \geq M_s(f_1(\xi(\boldsymbol{\alpha}, u_1)), f_2(\xi(\boldsymbol{\alpha}, u_2)); 1 - \lambda, \lambda),$$

and then

$$\begin{aligned} H((1-\lambda)u_1 + \lambda u_2) &\geq \int M_s(f_1(\xi(\boldsymbol{\alpha}, u_1)), f_2(\xi(\boldsymbol{\alpha}, u_2)); 1 - \lambda, \lambda) dP(\boldsymbol{\alpha}) \\ &= M_s\left(\int f_1(\xi(\boldsymbol{\alpha}, u_1)) dP(\boldsymbol{\alpha}), \int f_2(\xi(\boldsymbol{\alpha}, u_2)) dP(\boldsymbol{\alpha}); 1 - \lambda, \lambda\right) \\ &\overset{(\star)}{\geq} M_s(F_1(u_1), F_2(u_2); 1 - \lambda, \lambda), \end{aligned}$$

where in $(\star)$ we used that for $s > -1$ the map

$$(x, y) \mapsto M_s(x, y; 1 - \lambda, \lambda) = [(1-\lambda)x^s + \lambda y^s]^{1/s}$$

is convex on $(0, \infty) \times (0, \infty)$, and then we can apply Niculescu & Persson (2006, Theorem 3.5.3), which is an extension of the Jensen's inequality for in 2 variables. $\square$

## G   PROOF OF THEOREM 5.4

We compute the regret upper bound of a single firm $i$. Recall the definitions

$$\mathrm{rev}_i(p \mid \mathbf{p}_{-i}) = p\psi_i(-\beta_i p + \boldsymbol{\gamma}_i^\top \mathbf{p}_{-i}), \quad \widehat{\mathrm{rev}}_i(p \mid \mathbf{p}_{-i}) = p\widehat{\psi}_{i,\widetilde{\boldsymbol{\theta}}_i}(-\widetilde{\beta}_i p + \widetilde{\boldsymbol{\gamma}}_i^\top \mathbf{p}_{-i}),$$

so that

$$\mathrm{Reg}_i(T) = \mathbb{E}\sum_{t=1}^T [\mathrm{rev}_i(\Gamma_i(\mathbf{p}_{-i}^{(t)}) \mid \mathbf{p}_{-i}^{(t)}) - \mathrm{rev}_i(p_i^{(t)} \mid \mathbf{p}_{-i}^{(t)})],$$

where

$$\Gamma_i(\mathbf{p}_{-i}^{(t)}) = \mathrm{argmax}_{p_i \in \mathcal{P}_i}\, \mathrm{rev}_i(p_i \mid \mathbf{p}_{-i}^{(t)}), \quad p_i^{(t)} = \mathrm{argmax}_{p_i \in \mathcal{P}_i}\, \widehat{\mathrm{rev}}_i(p_i \mid \mathbf{p}_{-i}^{(t-1)}).$$

By optimality condition of $\Gamma_i$ we have

$$\mathrm{rev}_i(p_i^{(t)} \mid \mathbf{p}_{-i}^{(t)}) = \mathrm{rev}_i(\Gamma_i(\mathbf{p}_{-i}^{(t)}) \mid \mathbf{p}_{-i}^{(t)})$$
$$+ \underbrace{\partial_p \mathrm{rev}_i(p \mid \mathbf{p}_{-i}^{(t)})_{|p=\Gamma_i(\mathbf{p}^{(t)})}(p_i^{(t)} - \Gamma_i(\mathbf{p}_{-i}^{(t)}))}_{\geq 0}$$
$$+ \partial_{p^2} \mathrm{rev}_i(p \mid \mathbf{p}_{-i}^{(t)})_{|p=p'}(p_i^{(t)} - \Gamma_i(\mathbf{p}_{-i}^{(t)}))^2$$

for some $p'$ in the segment between the points $p_i^{(t)}$ and $\Gamma_i(\mathbf{p}_{-i}^{(t)})$. From this, we derive

$$\mathrm{rev}_i(\Gamma_i(\mathbf{p}_{-i}^{(t)}) \mid \mathbf{p}_{-i}^{(t)}) - \mathrm{rev}_i(p_i^{(t)} \mid \mathbf{p}_{-i}^{(t)}) \leq -\partial_{p^2}^2 \mathrm{rev}_i(p \mid \mathbf{p}_{-i}^{(t)})_{|p=p'}(\Gamma_i(\mathbf{p}_{-i}^{(t)}) - p_i^{(t)})^2.$$

Now, note that

$$-\partial_{p^2}^2 \mathrm{rev}_i(p \mid \mathbf{p}_{-i}^{(t)})_{|p=p'} = 2\beta_i\psi_i'(-\beta_i p' + \boldsymbol{\gamma}_i^\top \mathbf{p}_{-i}^{(t)}) - \beta_i^2 p'\psi_i''(-\beta_i p' + \boldsymbol{\gamma}_i^\top \mathbf{p}_{-i}^{(t)})$$
$$\leq 2\beta_i \bar{B}_{\psi_i'} + \beta_i^2 \bar{p}_i B_{\psi_i''},$$

where we used Assumption 3.2: $0 < \underline{B}_{\psi_i'} \leq \psi_i' \leq \bar{B}_{\psi_i'}$ and $|\psi_i''| \leq B_{\psi_i''}$ on $\mathcal{U}$ for some (unknown) $\underline{B}_{\psi_i'}, \bar{B}_{\psi_i'}, B_{\psi_i''} > 0$. This implies

$$\mathrm{rev}_i(\Gamma_i(\mathbf{p}_{-i}^{(t)}) \mid \mathbf{p}_{-i}^{(t)}) - \mathrm{rev}_i(p_i^{(t)} \mid \mathbf{p}_{-i}^{(t)}) \lesssim (\Gamma_i(\mathbf{p}_{-i}^{(t)}) - p_i^{(t)})^2$$
$$\lesssim \|\boldsymbol{\Gamma}(\mathbf{p}^{(t)}) - \mathbf{p}^{(t)}\|_2^2$$
$$\lesssim \|\boldsymbol{\Gamma}(\mathbf{p}^{(t)}) - \mathbf{p}^*\|_2^2 + \|\mathbf{p}^* - \mathbf{p}^{(t)}\|_2^2$$
$$= \|\boldsymbol{\Gamma}(\mathbf{p}^{(t)}) - \boldsymbol{\Gamma}(\mathbf{p}^*)\|_2^2 + \|\mathbf{p}^* - \mathbf{p}^{(t)}\|_2^2$$
$$\lesssim \|\mathbf{p}^{(t)} - \mathbf{p}^*\|_2^2 + \|\mathbf{p}^* - \mathbf{p}^{(t)}\|_2^2$$
$$\lesssim \|\mathbf{p}^{(t)} - \mathbf{p}^*\|_2^2 \tag{36}$$

Observe that this implies

$$\mathrm{Reg}_i(T) \lesssim \mathbb{E}\sum_{t=1}^T \|\mathbf{p}^{(t)} - \mathbf{p}^*\|_2^2. \tag{37}$$

**Lemma G.1.** *If Assumptions in the Theorem hold, for $T$ sufficiently large we have* $\mathbb{E}[\sup_{\mathbf{p}\in\mathcal{P}} |\psi_i(\boldsymbol{\theta}_i^\top \mathbf{p}) - \widehat{\psi}_{i,\widetilde{\boldsymbol{\theta}}_i}(\widetilde{\boldsymbol{\theta}}_i^\top \mathbf{p})|] \lesssim \mathscr{X}_i\sqrt{N}(\frac{\log\tau}{\tau})^{2/5}$, *where* $\mathscr{X}_i = \max\{\mathscr{C}_i, \mathscr{B}_i\}$, *and* $\mathscr{B}_i$ *and* $\mathscr{C}_i$ *are defined in (17) and (21), respectively. Moreover, there exists a unique value* $\kappa_i^\star \in (0,1)$ *that minimizes* $\mathbb{E}[\sup_{\mathbf{p}\in\mathcal{P}} |\psi_i(\boldsymbol{\theta}_i^\top \mathbf{p}) - \widehat{\psi}_{i,\widetilde{\boldsymbol{\theta}}_i}(\widetilde{\boldsymbol{\theta}}_i^\top \mathbf{p})|]$. *This value satisfies the implicit equation:*

$$4\mathscr{C}_i\tau^{1/10}\kappa_i^{3/2} = 5\mathscr{B}_i N^{1/2}(1 - \kappa_i)^{7/5}.$$

**Lemma G.2.** *If Assumptions in the Theorem hold, for $T$ sufficiently large,* $\sup_{\mathbf{p}\in\mathcal{P}} |\psi_i(\boldsymbol{\theta}_i^\top \mathbf{p}) - \widehat{\psi}_{i,\widetilde{\boldsymbol{\theta}}_i}(\widetilde{\boldsymbol{\theta}}_i^\top \mathbf{p})| \leq \epsilon_i$ *implies* $\|\mathbf{p}^{(t)} - \mathbf{p}^*\|_2^2 \lesssim N\sup_{i\in\mathcal{N}}\epsilon_i + L_{\boldsymbol{\Gamma}}^{2(t-\tau-1)}$ *for every* $t \geq \tau + 1$.

By Lemma G.1, Lemma G.2 have that for $t \geq \tau + 1$ with $\tau$ sufficiently large,

$$
\mathbb{E}[\mathrm{rev}_i(\Gamma_i(\mathbf{p}^{(t)}_{-i}) \mid \mathbf{p}^{(t)}_{-i}) - \mathrm{rev}_i(p^{(t)}_i \mid \mathbf{p}^{(t)}_{-i})] \overset{(36)}{\lesssim} \mathbb{E}[\|\mathbf{p}^{(t)} - \mathbf{p}^*\|^2_2]
$$
$$
\overset{Lemma\ G.1, Lemma\ G.2}{\lesssim} \mathscr{X}_i N^{3/2} \left( \frac{\log \tau}{\tau} \right)^{2/5} + L^{2(t-\tau-1)}_{\boldsymbol{\Gamma}},
$$
$$(38)$$

and as a consequence, give that $\tau \propto T^\xi$, for every $t \geq T^\xi + 1$

$$
\mathrm{Reg}_i(T) = \mathbb{E} \sum_{t=1}^{\tau} [\mathrm{rev}_i(\Gamma_i(\mathbf{p}^{(t)}_{-i}) \mid \mathbf{p}^{(t)}_{-i}) - \mathrm{rev}_i(p^{(t)}_i \mid \mathbf{p}^{(t)}_{-i})]
$$
$$
+ \mathbb{E} \sum_{t=\tau+1}^{T} [\mathrm{rev}_i(\Gamma_i(\mathbf{p}^{(t)}_{-i}) \mid \mathbf{p}^{(t)}_{-i}) - \mathrm{rev}_i(p^{(t)}_i \mid \mathbf{p}^{(t)}_{-i})]
$$
$$
\lesssim \tau + (T - \tau)\mathscr{X}_i N^{3/2} \left( \frac{\log \tau}{\tau} \right)^{2/5} + \sum_{t=\tau+1}^{T} L^{2(t-\tau-1)}_{\boldsymbol{\Gamma}}
$$
$$
\lesssim \tau + T \mathscr{X}_i N^{3/2} \left( \frac{\log \tau}{\tau} \right)^{2/5} + \sum_{j=0}^{T-\tau-1} L^{2j}_{\boldsymbol{\Gamma}}
$$
$$
= T^\xi + T \mathscr{X}_i N^{3/2} \left( \frac{\log T^\xi}{T^\xi} \right)^{2/5} + \frac{1 - L^{2(T-\tau)}_{\boldsymbol{\Gamma}}}{1 - L^2_{\boldsymbol{\Gamma}}}
$$
$$
\leq T^\xi + \mathscr{X}_i T^{1-2\xi/5} N^{3/2} (\log T)^{2/5} + 1.
$$

Note that we are retaining the dependence on $\mathscr{X}_i = \mathscr{X}_i(s_i)$ to demonstrate the way that $s_i$ affects the rate of the regret. Substituting back the value $\tau \propto T^\xi$ into Equation (38), we get

$$
\mathbb{E}[\|\mathbf{p}^{(t)} - \mathbf{p}^*\|^2_2] \lesssim \mathscr{X}_i N^{3/2} (\tfrac{\log \tau}{\tau})^{2/5} + L^{2(t-\tau-1)}_{\boldsymbol{\Gamma}}
$$
$$
\lesssim \mathscr{X}_i N^{3/2} (\tfrac{\log T^\xi}{T^\xi})^{2/5} + L^{2(t-T^\xi-1)}_{\boldsymbol{\Gamma}}
$$
$$
= \mathscr{X}_i N^{3/2} T^{-2\xi/5} (\log T)^{2/5} + L^{2(t-T^\xi-1)}_{\boldsymbol{\Gamma}},
$$

and for $t = T$
$$
\mathbb{E}[\|\mathbf{p}^{(T)} - \mathbf{p}^*\|^2_2] \lesssim \mathscr{X}_i N^{3/2} T^{-2\xi/5} (\log T)^{2/5} + L^{2(T-T^\xi-1)}_{\boldsymbol{\Gamma}}.
$$

This completes the proof.

**Remark G.3** (Dependence of the regret and NE convergence rate on $s_i$). *The value of $\xi$ that minimizes the expected regret $\mathrm{Reg}_i(T)$ is obtained by equalizing the exponents of $T$ in the upper bound $T^\xi + T^{1-2\xi/5}$, ignoring logarithmic factors. Solving this balance gives $\xi = 5/7$. Consequently, for each $i \in \mathcal{N}$ we obtain*

$$
\mathrm{Reg}_i(T) = O\left( \mathscr{X}_i N^{3/2} T^{5/7} (\log T)^{2/5} \right),
$$
$$
\mathbb{E}\|\mathbf{p}^{(T)} - \mathbf{p}^*\|^2_2 = O\left( \mathscr{X}_i N^{3/2} T^{-2/7} (\log T)^{2/5} + L^{2T+o(T)}_{\boldsymbol{\Gamma}} \right), \qquad T \to \infty.
$$

*These bounds retain the constants $\mathscr{X}_i$ and $L_{\boldsymbol{\Gamma}}$ because they depend on the parameters $s_1, \ldots, s_N$ (or equivalently $c_1, \ldots, c_N$). Note that the dependence on $\{s_j\}_{j \in [N]}$ enters only through the multiplicative constant $\mathscr{X}_i$ and the additive constant $L_{\boldsymbol{\Gamma}}$, never through the powers of $T$. This makes explicit how the choice of $s_i$ influences the regret and the convergence to the NE. Below we show that if $s_i$ are the optimal-concave parameters (as defined in Definition L.3), i.e.*

$$
s_i = \sup\Big\{ t_i : \sup_{u \in \mathcal{U}_i} [\psi_i(u)\psi''_i(u) + (t_i - 1)(\psi'_i(u))^2] \leq 0 \Big\},
$$

*then any misspecification $s_i' \le s_i$ satisfying*

$$\sup_{i \in \mathcal{N}} \frac{|1 - 1/s_i'+1| \, \|\boldsymbol{\gamma}_i\|_1}{\beta_i} < 1$$

*does not affect the regret or the NE convergence rates.*

(1) **Multiplicative constants $\mathscr{X}_i$.** *The constant $\mathscr{X}_i = \{\mathscr{C}_i, \mathscr{B}_i\}$ contains $\mathscr{B}_i$ and $\mathscr{C}_i$ (defined in (17)–(21)), which are the multiplicative constants in the concentration bounds for*

$$\|\psi_i - \widehat{\psi}_{i,\widetilde{\boldsymbol{\theta}}_i}\|_\infty \qquad and \qquad \|\boldsymbol{\theta}_i - \widetilde{\boldsymbol{\theta}}_i\|_2,$$

*respectively. Any value $s_i' \ne s_i$ does not affect the rate of linear estimator $\widetilde{\boldsymbol{\theta}}_i$, but it could introduce a bias in the estimator $\widehat{\psi}_{i,\widetilde{\boldsymbol{\theta}}_i}$. Precisely, if we choose $s_i' \le s_i$, the inclusion property of Proposition L.1 implies that an $s_i$-optimal-concave function remains $s_i'$-concave. Thus no bias is introduced in the concentration of $\|\psi_i - \widehat{\psi}_{i,\widetilde{\boldsymbol{\theta}}_i}\|_\infty$, whereas a choice $s_i' > s_i$ would. Then for $s_i' \le s_i$, since $\mathscr{B}_i$ and $\mathscr{C}_i$ are uniformly bounded in $s_i$, i.e. $\mathscr{B}_i, \mathscr{C}_i < C$ for some $C > 0$ independent of $s_i$, they do not affect the regret or NE rates.*

(2) **Additive constant $L_{\boldsymbol{\Gamma}}$.** *The constant $L_{\boldsymbol{\Gamma}}$ is the contraction modulus of the best-response map and determines the geometric decay term in the NE convergence. Convergence holds whenever $L_{\boldsymbol{\Gamma}} \in (0,1)$, or equivalently $\sup_{i \in \mathcal{N}} \|g_i'\|_\infty \|\boldsymbol{\gamma}_i\|_1 / \beta_i < 1$. Because $g_i'(u) = 1 - 1/\varphi_i'(\varphi_i^{-1}(u))$, we have*

$$1 - \frac{1}{c_i} < g_i'(u) < 1, \qquad c_i = s_i + 1.$$

*Thus any $s_i'$ satisfying $\sup_{i \in \mathcal{N}} |1-1/s_i'+1| \, \|\boldsymbol{\gamma}_i\|_1 / \beta_i < 1$ preserves contraction. If $L_{\boldsymbol{\Gamma}} \ge 1$, contraction fails and the regret guarantees break down.*

*Combining both components, we conclude that as long as the sellers choose $s_i' \le s_i$ (where $s_i$ are the optimal-concave parameters of $\psi_i$) and $\sup_{i \in \mathcal{N}} |1-1/s_i'+1| \, \|\boldsymbol{\gamma}_i\|_1 / \beta_i < 1$, the convergence rates of both the regret and the NE remain unchanged.*

## G.1 PROOF OF LEMMA G.1

Let $\tau$ be the minimum exploration phase among the $N$ firms and $t \in \{\tau + 1, \tau + 2, \dots, T\}$. Let $n_i^{(1)}$ be the length of the exploration phase to estimate $\boldsymbol{\theta}_i$ and $n_i^{(2)}$ the remaining length for estimating the link function $\psi_i$. Define the event $\mathcal{E}_i = \{\|\widetilde{\boldsymbol{\theta}}_i - \boldsymbol{\theta}_i\| \le R_{n_i^{(1)}}\}$ where $R_{n_i^{(1)}} = \mathscr{B}_i \sqrt{\frac{N \log n_i^{(1)}}{n_i^{(1)}}}$, where $\mathscr{B}_i$ is defined in Proposition D.3, and

$$\mathscr{R}_i(p_i | \mathbf{p}_{-i}) \triangleq |\psi_i(-\beta_i p_i + \boldsymbol{\gamma}_i^\top \mathbf{p}_{-i}) - \widehat{\psi}_{i,\widetilde{\boldsymbol{\theta}}_i}(-\widetilde{\beta}_i p_i + \widetilde{\boldsymbol{\gamma}}_i^\top \mathbf{p}_{-i})|, \quad p_i \in \mathcal{P}_i, \mathbf{p}_{-i} \in \mathcal{P}_{-i},$$

We can write

$$\mathscr{R}_i(p_i | \mathbf{p}_{-i}) = \mathscr{R}_i(p_i | \mathbf{p}_{-i}) \mathbb{I}(\mathcal{E}_i) + \mathscr{R}_i(p_i | \mathbf{p}_{-i}) \mathbb{I}(\mathcal{E}_i^c).$$

**Analyzing the $\mathscr{R}_i(p_i | \mathbf{p}_{-i}) \mathbb{I}(\mathcal{E}_i^c)$:**

By Proposition D.3 we have $\mathbb{E}[\mathscr{R}_i(p_i | \mathbf{p}_{-i}) \mathbb{I}(\mathcal{E}_i^c)] \lesssim \mathbb{P}(\mathcal{E}_i^c) = Q_{n_i^{(1)}} = 2e^{-c_1 \varsigma_{\min}^2 n_i^{(1)}/16} + \frac{2}{n_i^{(1)}}$.

**Analyzing the $\mathscr{R}_i(p_i | \mathbf{p}_{-i}) \mathbb{I}(\mathcal{E}_i)$:**

$$\mathscr{R}_i(p_i | \mathbf{p}_{-i}) \mathbb{I}(\mathcal{E}_i) \le \underbrace{|\widehat{\psi}_{i,\widetilde{\boldsymbol{\theta}}_i}(-\widetilde{\beta}_i p_i + \widetilde{\boldsymbol{\gamma}}_i^\top \mathbf{p}_{-i}) - \psi_{i,\widetilde{\boldsymbol{\theta}}_i}(-\widetilde{\beta}_i p_i + \widetilde{\boldsymbol{\gamma}}_i^\top \mathbf{p}_{-i})| \mathbb{I}(\mathcal{E}_i)}_{=A}$$

$$+ \underbrace{|\psi_{i,\widetilde{\boldsymbol{\theta}}_i}(-\widetilde{\beta}_i p_i + \widetilde{\boldsymbol{\gamma}}_i^\top \mathbf{p}_{-i}) - \psi_{i,\widetilde{\boldsymbol{\theta}}_i}(-\beta_i p_i + \boldsymbol{\gamma}_i^\top \mathbf{p}_{-i})| \mathbb{I}(\mathcal{E}_i)}_{=B}$$

$$+ \underbrace{|\psi_{i,\widetilde{\boldsymbol{\theta}}_i}(-\beta_i p_i + \boldsymbol{\gamma}_i^\top \mathbf{p}_{-i}) - \psi_i(-\beta_i p_i + \boldsymbol{\gamma}_i^\top \mathbf{p}_{-i})| \mathbb{I}(\mathcal{E}_i)}_{=C}.$$

**Analyzing $A$ on $\mathcal{E}_i$:** Define the event $\mathcal{Q}_i = \{\sup_{u \in \mathcal{U}} |\widehat{\psi}_{i,\widetilde{\boldsymbol{\theta}}_i}(u) - \psi_{i,\widetilde{\boldsymbol{\theta}}_i}(u)| \leq \mathscr{C}_i \rho_{n_i^{(2)}}^{2/5}\}$. For $n_i^{(2)}$ sufficiently large, by Theorem D.6 we have that for every $\gamma > 4$

$$\mathbb{P}(\mathcal{Q}_i) \geq 1 - \frac{1}{(n_i^{(2)})^{\gamma - 2}}.$$

Consequently,

$$
\begin{aligned}
\mathbb{E}(A) &= \mathbb{E}(A \mathbb{I}(\mathcal{Q}_i, \cap \mathcal{E}_i)) + \mathbb{E}(A \mathbb{I}(\mathcal{Q}_i^c \cap \mathcal{E}_i)) \\
&\leq \mathbb{E}\left(\sup_{u \in \mathcal{U}} |\widehat{\psi}_{i,\widetilde{\boldsymbol{\theta}}_i}(u) - \psi_{i,\widetilde{\boldsymbol{\theta}}_i}(u)| \mathbb{I}(\mathcal{Q}_i \cap \mathcal{E}_i)\right) + 2\mathbb{P}(\mathcal{Q}_i^c)\mathbb{P}(\mathcal{E}_i) \\
&= \mathscr{C}_i \left(\frac{\log n_i^{(2)}}{n_i^{(2)}}\right)^{2/5} \underbrace{\mathbb{P}(\mathcal{Q}_i \cap \mathcal{E}_i)}_{\leq 1} + 2\mathbb{P}(\mathcal{Q}_i^c) \\
&\leq \mathscr{C}_i \left(\frac{\log n_i^{(2)}}{n_i^{(2)}}\right)^{2/5} + 2\frac{1}{(n_i^{(2)})^{\gamma - 2}} \\
&\leq \mathscr{C}_i \left(\frac{\log n_i^{(2)}}{n_i^{(2)}}\right)^{2/5},
\end{aligned}
$$

where we chose $\gamma > 4$ and $n_i^{(2)}$ sufficiently large.

**Analyzing $B$ on $\mathcal{E}_i$:** By Proposition D.4, $\psi_{i,\widetilde{\boldsymbol{\theta}}_i}$ is Lipschitz, then

$$\mathbb{E}[B\mathbb{I}(\mathcal{E}_i)] \lesssim \|\widetilde{\boldsymbol{\theta}}_i - \boldsymbol{\theta}_i\|_2 \leq R_{n_i^{(1)}} = \mathscr{B}_i \left(N\frac{\log n_i^{(1)}}{n_i^{(1)}}\right)^{1/2}.$$

**Analyzing $C$ on $\mathcal{E}_i$:** By Proposition D.4 we have

$$|\psi_{i,\widetilde{\boldsymbol{\theta}}_i}(u) - \psi_i(u)|\mathbb{I}(\mathcal{E}_i) \lesssim \|\widetilde{\boldsymbol{\theta}}_i - \boldsymbol{\theta}_i\|_2 \leq R_{n_i^{(1)}} = \mathscr{B}_i \left(N\frac{\log n_i^{(1)}}{n_i^{(1)}}\right)^{1/2}.$$

**Combining the terms $\mathscr{R}_i(p_i|\mathbf{p}_{-i})\mathbb{I}(\mathcal{E}_i^c)$ and $\mathscr{R}_i(p_i|\mathbf{p}_{-i})\mathbb{I}(\mathcal{E}_i)$:** We got that

$$
\mathscr{R}_i(p_i|\mathbf{p}_{-i}) \lesssim \underbrace{\mathscr{C}_i \left(\frac{\log n_i^{(2)}}{n_i^{(2)}}\right)^{2/5}}_{\text{Bound of the non parametric error associated with } \psi_i} + \underbrace{\mathscr{B}_i \left(N\frac{\log n_i^{(1)}}{n_i^{(1)}}\right)^{1/2}}_{\text{Bound of the parametric error associated with } \boldsymbol{\theta}_i}.
$$

We now derive the optimal choice of $\kappa_i$. Since $\mathscr{R}_i(p_i|\mathbf{p}_{-i})$ appears as a multiplicative factor in the total regret we want to make $\mathscr{R}_i(p_i|\mathbf{p}_{-i})$ as small as possible. Using that $n_i^{(1)} = \tau\kappa_i$, $n_i^{(2)} = \tau(1-\kappa_i)$, excluding log-factors, we have that

$$\mathscr{R}_i(p_i|\mathbf{p}_{-i}) \lesssim \mathscr{C}_i(\tau(1-\kappa_i))^{-2/5} + \mathscr{B}_i N^{1/2}(\tau\kappa_i)^{-1/2}.$$

We want to minimize the function

$$f(\kappa_i) := \mathscr{C}_i(\tau(1-\kappa_i))^{-2/5} + \mathscr{B}_i N^{1/2}(\tau\kappa_i)^{-1/2}$$

for constants $\tau > 0$, $N > 0$, and $\kappa_i \in (0,1)$. Define for simplicity $A_i = \mathscr{C}_i\tau^{-2/5}$, $B_i = \mathscr{B}_i N^{1/2}\tau^{-1/2}$. Then:

$$f(\kappa_i) = A_i(1-\kappa_i)^{-2/5} + B_i\kappa_i^{-1/2}, \quad f'(\kappa_i) = \frac{2A_i}{5}(1-\kappa_i)^{-7/5} - \frac{B_i}{2}\kappa_i^{-3/2}.$$

Setting $f'(\kappa_i) = 0$ gives:

$$\frac{2A_i}{5}(1-\kappa_i)^{-7/5} = \frac{B_i}{2}\kappa_i^{-3/2} \Leftrightarrow \quad \kappa_i^{3/2} = \frac{5}{4}\frac{B_i}{A_i}\tau^{-1/10}(1-\kappa_i)^{7/5}$$

Substituting back the expressions for $A_i$ and $B_i$:

$$\kappa_i^{3/2} = \frac{5}{4}\frac{\mathscr{B}_i}{\mathscr{C}_i}N^{1/2}\tau^{-1/10}(1-\kappa_i)^{7/5}$$

The optimal $\kappa_i^\star \in (0,1)$ satisfies the implicit equation:

$$\kappa_i^{3/2} = \frac{5}{4}\frac{\mathscr{B}_i}{\mathscr{C}_i}N^{1/2}\tau^{-1/10}(1-\kappa_i)^{7/5}$$

We now prove that there exists a unique $\kappa_i^*$ satisfying this equation. Let $g(\kappa_i) = \kappa_i^{3/2} - \frac{5}{4}\frac{\mathscr{B}_i}{\mathscr{C}_i}N^{1/2}\tau^{-1/10}(1-\kappa_i)^{7/5}$ and note that $g(0) = -\frac{5}{4}\frac{\mathscr{B}_i}{\mathscr{C}_i}N^{1/2}\tau^{-1/10} < 0$, $g(1) = 1$ and $g'(\kappa_i) = \frac{3}{2}\kappa_i^{1/2} + \frac{7}{4}N^{1/2}\frac{\mathscr{B}_i}{\mathscr{C}_i}\tau^{-1/10}(1-\kappa_i)^{2/5} > 0$. Note that, since we are excluding all the constants when upper bounding $\mathscr{R}_i(p_i|\mathbf{p}_{-i})$, the optimal value $\kappa_i^\star$ is independent of $i$ and only depends on the exploration length $\tau$ and number of sellers in the market $N$.

Now, since the optimal $\kappa_i^* \in (0,1)$ is fixed, and $n_i^{(1)} = \frac{1-\kappa_i^\star}{\kappa_i^\star}n_i^{(2)} \propto n_i^{(2)}$ we have that the dominating term is

$$\mathscr{R}_i(p_i|\mathbf{p}_{-i}) \lesssim \mathscr{C}_i\left(\frac{\log(\tau(1-\kappa_i))}{\tau(1-\kappa_i)}\right)^{2/5} + \mathscr{B}_i\left(N\frac{\log(\tau\kappa_i)}{\tau\kappa_i}\right)^{1/2} \lesssim \mathscr{X}_i\sqrt{N}\left(\frac{\log n_i^{(2)}}{n_i^{(2)}}\right)^{2/5}.$$

where $\mathscr{X}_i = \max\{\mathscr{C}_i, \mathscr{B}_i\}$.

## G.2 PROOF OF LEMMA G.2

Recall the definition

$$\mathrm{rev}_i(p \mid \mathbf{p}_{-i}) = p\psi_i(-\beta_i p + \boldsymbol{\gamma}_i^\top \mathbf{p}_{-i}), \quad \widehat{\mathrm{rev}}_i(p \mid \mathbf{p}_{-i}) = p\widehat{\psi}_{i,\widetilde{\boldsymbol{\theta}}_i}(-\widetilde{\beta}_i p + \widetilde{\boldsymbol{\gamma}}_i^\top \mathbf{p}_{-i}),$$

and

$$\Gamma_i(\mathbf{p}_{-i}) = \mathrm{argmax}_{p_i \in \mathcal{P}_i}\,\mathrm{rev}_i(p_i \mid \mathbf{p}_{-i}), \quad \widehat{\Gamma}_i(\mathbf{p}_{-i}) = \mathrm{argmax}_{p_i \in \mathcal{P}_i}\,\widehat{\mathrm{rev}}_i(p_i \mid \mathbf{p}_{-i}).$$

We have

$$\sup_{\mathbf{p}\in\mathcal{P}} |\mathrm{rev}_i(p_i \mid \mathbf{p}_{-i}) - \widehat{\mathrm{rev}}_i(p_i \mid \mathbf{p}_{-i})| \le \bar{p}_i \sup_{\mathbf{p}\in\mathcal{P}} |\psi_i(\boldsymbol{\theta}_i^\top \mathbf{p}) - \widehat{\psi}_{i,\widetilde{\boldsymbol{\theta}}_i}(\widetilde{\boldsymbol{\theta}}_i^\top \mathbf{p})| \le \epsilon_i.$$

Then we have that for every $\mathbf{p} \in \mathcal{P}$,

$$\mathrm{rev}_i(\Gamma_i(\mathbf{p}_{-i}) \mid \mathbf{p}_{-i}) \le \widehat{\mathrm{rev}}_i(\Gamma_i(\mathbf{p}_{-i}) \mid \mathbf{p}_{-i})+\epsilon_i, \quad \mathrm{rev}_i(\widehat{\Gamma}_i(\mathbf{p}_{-i}) \mid \mathbf{p}_{-i}) \ge \widehat{\mathrm{rev}}_i(\widehat{\Gamma}_i(\mathbf{p}_{-i}) \mid \mathbf{p}_{-i})-\epsilon_i,$$

that implies

$$\mathrm{rev}_i(\Gamma_i(\mathbf{p}_{-i}) \mid \mathbf{p}_{-i})-\mathrm{rev}_i(\widehat{\Gamma}_i(\mathbf{p}_{-i}) \mid \mathbf{p}_{-i}) \le \underbrace{\widehat{\mathrm{rev}}_i(\Gamma_i(\mathbf{p}_{-i}) \mid \mathbf{p}_{-i}) - \widehat{\mathrm{rev}}_i(\widehat{\Gamma}_i(\mathbf{p}_{-i}) \mid \mathbf{p}_{-i})}_{\le 0}+2\epsilon_i \le 2\epsilon_i.$$

Now define

$$\mu_i := 2\beta_i\underline{B}_{\psi_i'} - \beta_i^2\bar{p}_i B_{\psi_i''},$$

which is strictly positive by Assumption 3.2. Then we have

$$2\epsilon_i \ge \mathrm{rev}_i(\Gamma_i(\mathbf{p}_{-i}) \mid \mathbf{p}_{-i}) - \mathrm{rev}_i(\widehat{\Gamma}_i(\mathbf{p}_{-i}) \mid \mathbf{p}_{-i}) \ge \frac{\mu_i}{2}(\Gamma_i(\mathbf{p}_{-i}) - \widehat{\Gamma}_i(\mathbf{p}_{-i}))^2.$$

This implies

$$\|\boldsymbol{\Gamma}(\mathbf{p}) - \widehat{\boldsymbol{\Gamma}}(\mathbf{p})\|_2 = \sqrt{\sum_{i\in\mathcal{N}}(\Gamma_i(\mathbf{p}_{-i}) - \widehat{\Gamma}_i(\mathbf{p}_{-i}))^2} \le 2\sqrt{\sum_{i\in\mathcal{N}}\frac{\epsilon_i}{\mu_i}} \le 2\sqrt{N\sup_{i\in\mathcal{N}}\frac{\epsilon_i}{\mu_i}} =: \epsilon, \quad \forall i \in \mathcal{N}.$$

This implies that

$$\sup_{\mathbf{p}\in\mathcal{P}} \|\mathbf{\Gamma}(\mathbf{p}) - \widehat{\mathbf{\Gamma}}(\mathbf{p})\|_2 \leq \epsilon.$$

For every $t \geq \tau + 1$ we have

$$
\begin{aligned}
\|\mathbf{p}^{(t)} - \mathbf{p}^*\|_2 &= \|\widehat{\mathbf{\Gamma}}(\mathbf{p}^{(t-1)}) - \mathbf{p}^*\|_2 \\
&\leq \|\widehat{\mathbf{\Gamma}}(\mathbf{p}^{(t-1)}) - \mathbf{\Gamma}(\mathbf{p}^{(t-1)})\|_2 + \|\mathbf{\Gamma}(\mathbf{p}^{(t-1)}) - \mathbf{\Gamma}(\mathbf{p}^*)\|_2 \\
&\leq \epsilon + L_{\mathbf{\Gamma}}\|\mathbf{p}^{(t-1)} - \mathbf{p}^*\|_2 \\
&\leq \epsilon + L_{\mathbf{\Gamma}}(\epsilon + L_{\mathbf{\Gamma}}\|\mathbf{p}^{(t-2)} - \mathbf{p}^*\|_2) \\
&= \epsilon(1 + L_{\mathbf{\Gamma}}) + L_{\mathbf{\Gamma}}^2\|\mathbf{p}^{(t-2)} - \mathbf{p}^*\|_2 \\
&= \epsilon\left(\sum_{j=0}^{\ell-1} L_{\mathbf{\Gamma}}^j\right) + L_{\mathbf{\Gamma}}^\ell\|\mathbf{p}^{(t-\ell)} - \mathbf{p}^*\|_2, \qquad \ell \in \{1,2,\ldots,t-\tau-1\} \\
&\leq \epsilon\frac{1}{1-L_{\mathbf{\Gamma}}} + L_{\mathbf{\Gamma}}^{(t-\tau-1)}\|\mathbf{p}^{(t-(t-\tau-1))} - \mathbf{p}^*\|_2 \\
&= \epsilon\frac{1}{1-L_{\mathbf{\Gamma}}} + L_{\mathbf{\Gamma}}^{(t-\tau-1)}\|\mathbf{p}^{(\tau+1)} - \mathbf{p}^*\|_2 \\
&\leq \epsilon\frac{1}{1-L_{\mathbf{\Gamma}}} + 2p_{\max}L_{\mathbf{\Gamma}}^{(t-\tau-1)}.
\end{aligned}
\tag{39}
$$

We conclude that, for every $t \geq \tau + 1$,

$$
\begin{aligned}
\|\mathbf{p}^{(t)} - \mathbf{p}^*\|_2^2 &\leq 2\left(\frac{1}{1-L_{\mathbf{\Gamma}}}\right)^2 \epsilon^2 + 8p_{\max}^2 L_{\mathbf{\Gamma}}^{2(t-\tau-1)} \\
&= 8\left(\frac{1}{1-L_{\mathbf{\Gamma}}}\right)^2 N\left(\sup_{i\in\mathcal{N}}\frac{\epsilon_i}{\mu_i}\right) + 8p_{\max}^2 L_{\mathbf{\Gamma}}^{2(t-\tau-1)} \\
&\lesssim N\sup_{i\in\mathcal{N}}\epsilon_i + L_{\mathbf{\Gamma}}^{2(t-\tau-1)},
\end{aligned}
$$

where we have absorbed the constants $8\left(\frac{1}{1-L_{\mathbf{\Gamma}}}\right)^2 \frac{1}{\min_{i\in\mathcal{N}}\mu_i}$ and $8p_{\max}^2$.

## H  NPLS FOR $s$-CONCAVE REGRESSION

Inspired by the work of Dümbgen et al. (2004), we prove a uniform convergence result for the NPLSE of a mean function that can be written as an increasing function $h$ composed with a concave function $\phi_0$. Suppose that we observe $(U_1, Y_1), (U_2, Y_2), \ldots, (U_n, Y_n)$ with fixed numbers $U_1 \leq U_2 \leq \cdots \leq U_n$ and independent random variables $Y_1, Y_2, \ldots, Y_n$ with

$$\mathbb{E}(Y \mid U = u) = \psi_0(u),$$

for unknown function $\psi_0 : \mathcal{U} \to (0, \infty)$, where $\mathcal{U}$ is an known interval containing

$$\mathscr{T} = \{U_1, U_2, \ldots, U_n\}.$$

Assume that

$$\psi_0 = h \circ \phi_0, \tag{40}$$

for some *known* increasing twice differentiable function $h : \mathbb{R} \to (0, \infty)$ and *unknown* concave function $\phi_0 : \mathcal{U} \to \mathbb{R}$. We denote the class of functions $\psi_0$ that can be written as $h \circ \phi_0$ with $\mathcal{F}_h$, which includes log-concave functions ($h(x) = e^x$) and more generally $s$-concave functions for every real $s$ ($h(x) = (-x)^{1/s}$ if $s < 0$ and $h(x) = x^{1/s}$ if $s > 0$).

Now we go back to the general known transformation $h$. We assume that $\psi_0 \in [\underline{B}_\psi, \bar{B}_\psi]$ for some *known* $0 < \underline{B}_\psi < \bar{B}_\psi < \infty$, so that $\mathcal{U} = \{u : \underline{B}_\psi \leq \psi_0(u) \leq \bar{B}_\psi\}$. Let $[\underline{B}_\phi, \bar{B}_\phi] \subset \mathbb{R}$ such that $[\underline{B}_\psi, \bar{B}_\psi] = h([\underline{B}_\phi, \bar{B}_\phi])$. Then $h' \in [\underline{B}_h, \bar{B}_h]$ on $[\underline{B}_\phi, \bar{B}_\phi]$ for some $0 < \underline{B}_h < \bar{B}_h < \infty$, where $\underline{B}_h$ and $\bar{B}_h$ depend on $\underline{B}_\psi$ and $\bar{B}_\psi$. Note that $\underline{B}_h, \bar{B}_h$ are well defined, indeed, by Weierstrass theorem, $h'$ is locally bounded, that is for every compact set $K$ there exists $0 < \underline{B}_h < \bar{B}_h < +\infty$ such that $\underline{B}_h \leq h' \leq \bar{B}_h$ in $K$. Then we can write

$$\widehat{\phi}_n \in \arg\min_{\phi \in \mathcal{C}} \left\{ \mathcal{L}(\phi) \triangleq \sum_{i=1}^n (Y_i - h \circ \phi(U_i))^2 \right\}, \tag{41}$$

$$\widehat{\psi}_n \triangleq h \circ \widehat{\phi}_n,$$

where $\mathcal{C}$ is the set of all concave functions $\phi : \mathcal{U} \to [\underline{B}_\phi, \bar{B}_\phi]$. *Note that the problem is not infinite dimensional, indeed we only need to recover $n$ variables* $(\phi(U_1), \phi(U_2), \ldots, \phi(U_n)) \in [\underline{B}_\phi, \bar{B}_\phi]$. When $h(x) = x$ (or equivalently for 1-concave functions, i.e. concave functions), (41) was studied by Dümbgen et al. (2004), for which $\widehat{\phi}_n$ exists and is unique and doesn't need restrictions on the codomain of $\phi_0$, that is $[\underline{B}_\phi, \bar{B}_\phi]$ can be the whole $\mathbb{R}$. Existence and uniqueness come from the coercivity of the loss $\mathcal{L}$, which is a property that only applies for $h(x) = x$. Since in our case $h$ is not necessarily the identity map, we rely on Weierstrass' theorem (which guarantees the existence of at least a minimum for a continuous function $F : \Omega \subset \mathbb{R}^n \to \mathbb{R}$ defined on a compact set $\Omega$ of $\mathbb{R}^n$). Thus a solution of (41) exists (but might not be unique) because the $n$-dimensional variable $(\phi(U_1), \phi(U_2), \ldots, \phi(U_n)) \in [\underline{B}_\phi, \bar{B}_\phi]$ where $[\underline{B}_\phi, \bar{B}_\phi]$ is compact. Uniqueness is not an issue in our case since we only care about the construction of a consistent estimator. Proposition H.1, which is followed by Lemma H.7, shows the constraints on the codomain of $\phi_0$ when $\psi_0 : \mathcal{U} \to [\underline{B}_\psi, \bar{B}_\psi]$ is $s$-concave for $s \neq 1$.

**Proposition H.1** (Constraints for $s$-concave transformations). *Suppose that $\psi_0$ is $s$-concave, that is $\psi_0 = h_s \circ \psi_0$ for some $s \neq 1$, where $h_s = d_s^{-1}$ as defined in (4). Then*

$$\phi_0 \in [\underline{B}_\phi, \bar{B}_\phi] \triangleq d_s([\underline{B}_\psi, \bar{B}_\psi]) = \begin{cases} [\log(\underline{B}_\psi), \log(\bar{B}_\psi)] & s = 0 \\ [-\underline{B}_\psi^s, -\bar{B}_\psi^s], & s < 0 \\ [\underline{B}_\psi^s, \bar{B}_\psi^s], & s > 0. \end{cases}$$

We now establish the assumptions to guarantee the convergence of the estimator defined in Equation (41). We consider a triangular scheme of observations $U_i = U_{n,i}$ and $Y_i = Y_{n,i}$ but suppress the additional subscript $n$ for notational simplicity. Let $\mathscr{M}_n$ be the empirical distribution of the design points $U_i$, i.e.

$$\mathscr{M}_n(B) \triangleq \frac{1}{n} \sum_{i=1}^n \mathbf{1}\{U_i \in B\}, \quad B \subset \mathbb{R}.$$

We analyze the asymptotic behavior of $\widehat{\psi}_n$ on a fixed compact interval

$$[a, b] \subset \mathcal{U},$$

under certain conditions on $\mathcal{M}$, the probability measure of the design points, and the errors

$$E_i \triangleq Y_i - \psi_0(U_i), \quad i = 1, 2, \ldots, n.$$

**Assumption H.2.** *The probability measure $\mathcal{M}$ of the design points satisfies $\mathcal{M}([u, v]) \geq C_{\mathcal{M}}(v - u)$ for some $C_{\mathcal{M}} > 0$ and for all $u < v$ with $u, v \in \mathcal{U}$.*

**Assumption H.3.** *For some constant $\sigma > 0$,*

$$\max_{i=1,\ldots,n} \mathbb{E} \exp(\mu E_i) \leq \exp(\sigma^2 \mu^2 / 2) \quad \text{for all } \mu \in \mathbb{R}.$$

**Assumption H.4.** *There exists a $\alpha \in [1, 2]$ and $L > 0$ such that for all $u, v \in [a, b]$*

$$\begin{cases} |\phi_0(u) - \phi_0(v)| \leq L|u - v| & \text{if } \alpha = 1 \\ |\phi_0'(u) - \phi_0'(v)| \leq L|u - v|^{\alpha - 1} & \text{if } \alpha > 1. \end{cases} \tag{42}$$

Assumption H.2 holds if the density $f_U$ of the measure $\mathcal{M}$ is bounded away from zero, that is $f_U \geq C_{\mathcal{M}}$, which is a standard assumption for uniform convergence of non-parametric functions (see e.g. Dümbgen et al. (2004); Mösching & Dümbgen (2020)) as well as for $L^2$ convergence (see e.g. Groeneboom & Hendrickx (2018); Balabdaoui et al. (2019)).

**Theorem H.5.** *Suppose that Assumption H.2, Assumption H.3 and Assumption H.4 are satisfied. Let $\widehat{\phi}_n$ be a solution of (41), and let $\widehat{\psi}_n = h \circ \widehat{\phi}_n$. Then, for all $\gamma > 4$ there exists a integer $n_0$ and a real value $\mathscr{C} > 0$ such that*

$$\mathbb{P}\left\{ \sup_{u \in [a+\delta_n, b-\delta_n]} |\psi_0(u) - \widehat{\psi}_n(u)| \leq \mathscr{C}\left(\frac{\log n}{n}\right)^{\alpha/2\alpha+1} \right\} \geq 1 - 2n^{2-\gamma}, \quad n \geq n_0,$$

*where $\mathscr{C}$ depends on $C_{\mathcal{M}}$, $\bar{B}_h$, $\underline{B}_h$, $L$, $\alpha$, $\sigma$ and $\gamma$, and $\delta_n = \widetilde{\mathscr{C}}(\log(n)/n)^{1/2\alpha+1}$ for some $\widetilde{\mathscr{C}} = \widetilde{\mathscr{C}}(L, \alpha)$. More specifically, $\mathscr{C}$ needs to satisfy*

$$\mathscr{C} \geq \max\left\{ \frac{4^{1+\alpha}L}{\alpha}\bar{B}_h, \frac{4\sqrt{7}\bar{B}_h\gamma\sigma\sqrt{\bar{B}_h^2 + \underline{B}_h^2}}{\underline{B}_h^2(C_{\mathcal{M}}/8)^{1/2}} \right\}.$$

The proof can be found in Appendix H.3.

**Remark H.6.** *If $\psi_0$ and $\widehat{\psi}_n$ depends also on a parameter $\boldsymbol{\theta} \in \Theta$, $\psi_{0,\boldsymbol{\theta}}$ and $\widehat{\psi}_{n,\boldsymbol{\theta}}$, for which the domain $\mathcal{U}$ does not depend on $\boldsymbol{\theta}$ and such that Assumption H.2 and Assumption H.4 hold uniformly in $\boldsymbol{\theta} \in \Theta$ and all the constants $C_{\mathcal{M}}, \bar{B}_h, \underline{B}_h, L, \alpha, \gamma, a, b$ are independent of $\boldsymbol{\theta}$, then*

$$\mathbb{P}\left\{ \sup_{u \in [a+\delta_n, b-\delta_n]} \sup_{\boldsymbol{\theta} \in \Theta} |\psi_{0,\boldsymbol{\theta}}(u) - \widehat{\psi}_{n,\boldsymbol{\theta}}(u)| \leq \mathscr{C}\left(\frac{\log n}{n}\right)^{\alpha/2\alpha+1} \right\} \geq 1 - 2n^{2-\gamma}.$$

## H.1 TECHNICAL LEMMA 1

**Lemma H.7.** *Recall the definition*

$$d_s(y) = \begin{cases} \log(y), & s = 0, y > 0 \\ -y^s, & s < 0, y > 0 \\ y^s, & s > 0, y > 0. \end{cases}$$

*Then it holds*

$$0 < C_s'(\underline{B}_\psi) \leq d_s' \leq C_s'(\bar{B}_\psi) \text{ and } C_s''(\underline{B}_\psi) \leq d_s'' \leq C_s''(\bar{B}_\psi), \text{ uniformly on } [\underline{B}_\psi, \bar{B}_\psi],$$

*where*

$$C_s'(\underline{B}_\psi) = \begin{cases} d_s'(\underline{B}_\psi), & s > 1, \\ 1, & s = 1, \\ d_s'(\bar{B}_\psi), & s < 1 \end{cases} \quad \text{and} \quad C_s''(\underline{B}_\psi) = \begin{cases} d_s''(\underline{B}_\psi), & 1 < s < 2 \\ \{0\}, & s = 1 \\ \{2\}, & s = 2 \\ d_s''(\underline{B}_\psi), & s \in (1,2)^c, \end{cases} \tag{43}$$

$C_s'' < 0$ *for* $s < 1$ *and* $C_s'' > 0$ *for* $s > 1$. *As a consequence, for every real s, the function* $h_s = d_s^{-1}$ *is increasing on* $[\underline{B}_\psi, \bar{B}_\psi]$ *with the following bounds*

$$0 < \frac{1}{C_s'(\bar{B}_\psi)} \le h_s' \le \frac{1}{C_s'(\underline{B}_\psi)}.$$

*Proof.*

$$d_s'(y) = \begin{cases} \frac{1}{y}, & s = 0, y > 0 \\ |s| y^{s-1}, & s \ne 0, y \ge 0 \end{cases}, \qquad d_s''(y) = \begin{cases} -\frac{1}{y^2}, & s = 0, y > 0 \\ |s|(s-1)y^{s-2}, & s \ne 0, y \ge 0. \end{cases}$$

**Case 1**. $s > 1$. Then $d_s'$ is increasing and $d_s''$ is increasing for $s > 2$, $d_s'' = 2$ for $s = 2$ and $d_s''$ is decreasing for $s \in (1, 2)$.

**Case 2**. $s = 1$. Then $d_s' = 1$ and $d_s'' = 0$.

**Case 3**. $s < 1$. Then $d_s'$ is decreasing and $d_s''$ increasing.

**Conclusion**. If $0 < \epsilon < u < 1 - \epsilon$, then we

$$d_s' \in \begin{cases} (d_s'(\underline{B}_\psi), d_s'(\bar{B}_\psi), & s > 1 \\ \{1\}, & s = 1 \\ (d_s'(\bar{B}_\psi), d_s'(\underline{B}_\psi)), & s < 1 \end{cases} \quad \text{and} \quad d_s'' \in \begin{cases} (d_s''(\underline{B}_\psi), d_s''(\bar{B}_\psi)), & 1 < s < 2 \\ \{0\}, & s = 1 \\ \{2\}, & s = 2 \\ (d_s''(\bar{B}_\psi), d_s''(\underline{B}_\psi)), & s \in (1, 2)^c, \end{cases}$$

and specifically

$$0 < C_s'(\underline{B}_\psi) \le d_s' \le C_s'(\bar{B}_\psi) \text{ and } C_s''(\underline{B}_\psi) \le d_s'' \le C_s''(\bar{B}_\psi), \text{ uniformly on } [\underline{B}_\psi, \bar{B}_\psi],$$

where

$$C_s'(\underline{B}_\psi) = \begin{cases} d_s'(\underline{B}_\psi), & s > 1, \\ 1, & s = 1, \\ d_s'(\bar{B}_\psi), & s < 1 \end{cases} \quad \text{and} \quad C_s''(\underline{B}_\psi) = \begin{cases} d_s''(\underline{B}_\psi), & 1 < s < 2 \\ \{0\}, & s = 1 \\ \{2\}, & s = 2 \\ d_s''(\bar{B}_\psi), & s \in (1, 2)^c \end{cases}$$

$C_s'' < 0$ for $s < 1$ and $C_s'' > 0$ for $s > 1$. $\qquad \square$

## H.2 TECHNICAL LEMMA 2

Assumption H.2 implies the following Lemma H.8 that will be used to prove the uniform convergence consistency in Theorem H.5.

**Lemma H.8.** *Let $\rho_n \triangleq \frac{\log n}{n}$ and $Leb(\cdot)$ stands for Lebesgue measure and $U_1, U_2, \ldots, U_n$ i.i.d. points with probability measure $M$ that satisfies Assumption H.2, then for a given constant $C_3 > 0$, and for any $\gamma > 2$ there exists $n_0 \in \mathbb{N}$ and a sequence $\epsilon_n = \epsilon_n(\gamma, C_3, \alpha) > 0$, $\epsilon_n \to 0$ such that*

$$\mathbb{P}\left(A_{n,\gamma}\right) > 1 - \frac{1}{2(n+2)^{\gamma-2}}, \quad n \geq n_0,$$

*where $A_{n,\gamma}$ is the event*

$$\inf\left\{\frac{\mathscr{M}_n\left(\mathcal{U}_n\right)}{Leb\left(\mathcal{U}_n\right)} : \mathcal{U}_n \subset \mathcal{U}, Leb\left(\mathcal{U}_n\right) \geq \delta_n := C_3 \rho_n^{1/(2\alpha+1)}\right\} \geq C_{\mathscr{M}}(1 - \epsilon_n).$$

This result immediately follos from the proof of the more general result by Mösching & Dümbgen (2020, Section 4.3) which can be stated as follows:

**Lemma H.9.** *Let $\delta_n > 0$ such that $\delta_n \to 0$ while $n\delta_n/\log(n) \to \infty$ (as $n \to \infty$). Then for every $\gamma > 2$, there exists $n_0 = n_0(\gamma, \delta_n)$ and $\epsilon_n = \epsilon_n(\gamma, \delta_n) > 0$, $\epsilon_n \to 0$ such that*

$$\mathbb{P}\left(\inf\left\{\frac{P_n\left(\mathcal{U}_n\right)}{P\left(\mathcal{U}_n\right)} : \mathcal{U}_n \subset \mathcal{U}, P\left(\mathcal{U}_n\right) \geq \delta_n\right\} \geq 1 - \epsilon_n\right) > 1 - \frac{1}{2(n+2)^{\gamma-2}}, \quad n \geq n_0,$$

*where $P(\cdot), P_n(\cdot)$ are respectively the probability measure and the empirical probability measure of the design points $U_1, U_2, \ldots, U_n$, that is*

$$P(B) \triangleq \int_B f_U(u)du, \quad P_n(B) \triangleq \frac{1}{n}\sum_{t=1}^n \mathbf{1}\{U_i \in B\}, \quad for \; B \subset \mathcal{U},$$

*and*

$$\epsilon_n \triangleq \max\left(c_n/\delta_n, \sqrt{2c_n/\delta_n}\right) + (n\delta_n)^{-1} \to 0,$$

*where $c_n \triangleq \gamma \log(n+2)/(n+1)$. The value $n_0$ is the smallest integer $n$ that satisfies $\epsilon_n < 1$.*

## H.3  PROOF OF THEOREM H.5

First note that for every $\phi \in \mathcal{C}$, since $\phi \in [\underline{B}_\phi, \bar{B}_\phi]$, then $h \circ \phi \in [\underline{B}_\psi, \bar{B}_\psi]$. By assumption we also have $h' \in [\underline{B}_h, \bar{B}_h]$ for some $0 < \underline{B}_h < \bar{B}_h < \infty$. Let $\widehat{\phi}_n$ be a solution of (41), i.e. the LS estimate of $\phi_0$. Then, being $\widehat{\phi}_n$ a minimizer, we have that for any direction $\Delta$ such that $\widehat{\phi}_n + \delta\Delta \in \mathcal{C}$ for $\delta \geq 0$ sufficiently small, $\mathcal{L}(\widehat{\phi}_n + \delta\Delta) \geq \mathcal{L}(\widehat{\phi}_n)$. Hence the directional derivative at $\widehat{\phi}_n$ along direction $\Delta$ satisfies

$$\left.\frac{d}{d\delta}\right|_{\delta=0} \mathcal{L}(\widehat{\phi}_n + \delta\Delta) = \lim_{\delta\downarrow 0} \frac{\mathcal{L}(\widehat{\phi}_n + \delta\Delta) - \mathcal{L}(\widehat{\phi}_n)}{\delta} \geq 0.$$

Fix such a direction $\Delta : \mathbb{R} \to \mathbb{R}$ that will be selected later. We have

$$0 \leq \left.\frac{d}{d\delta}\right|_{\delta=0} \mathcal{L}(\widehat{\phi} + \delta\Delta) = \left.\frac{d}{d\delta}\right|_{\delta=0} \sum_{i=1}^{n} \left(Y_i - h(\widehat{\phi}_n(U_i) + \delta\Delta(U_i))\right)^2$$

$$= 2\sum_{i=1}^{n} \Delta(U_i) h'(\widehat{\phi}_n(U_i)) \left(h(\widehat{\phi}_n(U_i)) - Y_i\right).$$

Adding and subtracting $\psi_0(U_i) = h(\phi_0(U_i))$ we get

$$-\sum_{i=1}^{n} \Delta(U_i) h'(\widehat{\phi}_n(U_i)) E_i \geq \sum_{i=1}^{n} \Delta(U_i) h'(\widehat{\phi}_n(U_i)) \left[h \circ \phi_0(U_i) - h \circ \widehat{\phi}_n(U_i)\right]. \tag{44}$$

In what follows, we apply (44) to a special class of perturbation functions $\Delta$ and write

$$\|\Delta\|_n \triangleq \left(\sum_{i=1}^{n} \Delta(U_i)^2\right)^{1/2}, \quad \|o_h\| \triangleq \sqrt{\underline{B}_h^2 + \bar{B}_h^2}, \text{ where } o_h = (\underline{B}_h, \bar{B}_h)^\top.$$

The next Lemma H.10 is proved in Appendix H.4

**Lemma H.10.** *For an integer $m \geq 0$, let $\mathcal{D}_m$ be the family of all continuous, piece-wise linear functions on $\mathbb{R}$ with at most $m$ knots. Then for any fixed $\gamma > 4$,*

$$B_{m,n,\lambda} \triangleq \left\{ S_n(m) \triangleq \sup_{\Delta \in \mathcal{D}_m} \frac{\left|\sum_{i=1}^{n} \Delta(U_i) h'(\widehat{\phi}_n(U_i)) E_i\right|}{\|\Delta\|_n \|o_h\|} \leq \gamma\sigma(m+1)^{1/2}(\log n)^{1/2} \right\} \quad \text{for all } m \geq 0,$$

*with probability at least $1 - 2n^{\frac{16-\gamma^2}{8}}$.*

The next ingredient for our proof is a result concerning the difference of two concave functions, which can be found in Dümbgen et al. (2004, Lemma 5.2) and we report the proof in Appendix H.5 for completeness.

**Lemma H.11.** *Let $\phi$ satisfy Assumption H.4. There is a universal constant $\widetilde{\mathscr{C}} = \widetilde{\mathscr{C}}(L, \alpha)$ with the following property: for any $\xi > 0$, let $\delta \leq \widetilde{\mathscr{C}} \min\left(b - a, \xi^{1/\alpha}\right)$. Then, for any concave function $\widehat{\phi}$*

$$\sup_{t \in [a,b]} (\widehat{\phi} - \phi) \geq \xi \quad or \quad \sup_{t \in [a+\delta, b-\delta]} (\phi - \widehat{\phi}) \geq \xi,$$

*implies*

$$\inf_{t \in [c,c+\delta]} (\widehat{\phi} - \phi) \geq \xi/4 \quad or \quad \inf_{t \in [c,c+\delta]} (\phi - \widehat{\phi}) \geq \xi/4,$$

*for some $c \in [a, b - \delta]$. More specifically we have $\widetilde{\mathscr{C}}^\alpha = \min\left\{\frac{3\alpha}{4L}, \frac{\alpha}{4^{1+\alpha}L}\right\} = \frac{\alpha}{4^{1+\alpha}L}$.*

Now, we have to show that one of our classes $\mathcal{D}_m$ does indeed contain useful perturbation functions $\Delta$. We denote with $\check{\phi}$ the unique continuous and piecewise linear function with knots in $\mathscr{T} \cap (U_1, U_n)$ such that $\check{\phi}_n = \widehat{\phi}_n$ on $\mathscr{T} = \{U_1, U_2, \ldots, U_n\}$. Thus $\check{\phi}_n$ is one particular LS estimator for $\phi_0$. The following technical Lemma H.12 can be found in Dümbgen et al. (2004, Lemma 3).

**Lemma H.12.** *For $0 < u \le b - a$ let*

$$\widetilde{\mathscr{M}}_n(u) := \min_{c \in [a, b-u]} \mathscr{M}_n[c, c+u].$$

*Suppose that $|\widehat{\phi}_n - \phi_0| \ge \xi$ on some interval $[c, c+\delta] \subset [a, b]$ with length $\delta > 0$. Then there is a function $\Delta \in \mathcal{D}_6$ such that*

$$\check{\phi}_n + \lambda\Delta \text{ is concave for some } \lambda > 0,$$
$$\Delta(\phi_0 - \widehat{\phi}_n) \ge \xi\Delta^2 \quad \text{on } \mathscr{T},$$
$$\|\Delta\|_n^2 \ge n\widetilde{\mathscr{M}}_n(\delta/2)/4. \tag{45}$$

Now we prove the theorem. Consider the event

$$\mathcal{V}_n(\mathscr{C}) = \left\{ \frac{\sup_{[a+\delta_n, b-\delta_n]} |\psi_0 - \widehat{\psi}_n|}{\delta_n^\alpha} \ge \mathscr{C} \right\},$$

for some (random) $\mathscr{C} > 0$ that we aim to prove to be $O_p(1)$. Then we have $|\psi_0 - \widehat{\psi}_n| = |v \circ \phi_0 - v \circ \widehat{\phi}_n| \le \bar{B}_h|\phi_0 - \widehat{\phi}_n|$. It follows that

$$\sup_{u \in [a+\delta_n, b-\delta_n]} |\phi_0 - \widehat{\phi}|(u) \ge \frac{\mathscr{C}}{\bar{B}_h}\delta_n^\alpha.$$

From Lemma H.11, replacing

$$\begin{cases} \delta \leftarrow \delta_n \\ \xi \leftarrow \frac{\mathscr{C}}{\bar{B}_h}\delta_n^\alpha, \end{cases}$$

there is a (random) interval $[c_n, c_n + \delta_n] \subset [a, b]$ on which $|\phi_0 - \widehat{\phi}_n| \ge \frac{\mathscr{C}}{4\bar{B}_h}\delta_n^\alpha$, provided that

- $n$ is sufficiently large to guarantee that $\xi^{1/\alpha} \le b - a$, so that $\min\{b - a, \widetilde{\mathscr{C}}\xi^{1/\alpha}\} = \widetilde{\mathscr{C}}\xi^{1/\alpha}$.

- $\frac{\mathscr{C}}{\bar{B}_h} \ge \widetilde{\mathscr{C}}^{-\alpha}$. This condition comes from the fact that $\delta \le \widetilde{\mathscr{C}}\xi^{1/\alpha}$ that is $\delta_n \le \widetilde{\mathscr{C}}\left(\frac{\mathscr{C}}{\bar{B}_h}\delta^\alpha\right)^{1/\alpha}$.

Because $[c_n, c_n + \delta_n]$ is a sub-interval of $\mathcal{U}$ on length $\delta_n$, Lemma H.8 (which holds by Assumption H.2) guarantees that this interval contains at least one observation $U_i$. This implies Lemma H.12. For any $\Delta \in \mathcal{D}_6$ define $I_\Delta = \{i : \Delta(U_i) < 0\}$. Define also $J = \{i : \phi_0(U_i) - \widehat{\phi}_n(U_i) > 0\}$. Using that $0 < h' \in [\underline{B}_h, \bar{B}_h]$ with $\underline{B}_h, \bar{B}_h > 0$ (which comes from Lemma H.7) we have that

$$0 < \underline{B}_h(\phi_0 - \widehat{\phi}_n) \le h(\phi_0) - h(\widehat{\phi}_n) \le \bar{B}_h(\phi_0 - \widehat{\phi}_n) \quad \text{on points } U_i \text{ with } i \in J,$$
$$\bar{B}_h(\phi_0 - \widehat{\phi}_n) \le h(\phi_0) - h(\widehat{\phi}_n) \le \underline{B}_h(\phi_0 - \widehat{\phi}_n) \le 0 \quad \text{on points } U_i \text{ with } i \in J^c,$$

and as a consequence, using that $0 < h' \in [\underline{B}_h, \bar{B}_h]$ we have

$$0 < \underline{B}_h^2(\phi_0 - \widehat{\phi}_n) \le h'(\widehat{\phi}_n)(h(\phi_0) - h(\widehat{\phi}_n)) \le \bar{B}_h^2(\phi_0 - \widehat{\phi}_n) \quad \text{on points } U_i \text{ with } i \in J,$$
$$\bar{B}_h^2(\phi_0 - \widehat{\phi}_n) \le h'(\widehat{\phi}_n)(h(\phi_0) - h(\widehat{\phi}_n)) \le \underline{B}_h^2(\phi_0 - \widehat{\phi}_n) < 0 \quad \text{on points } U_i \text{ with } i \in J^c. \tag{46}$$

Recall that $o_h = (\underline{B}_h, \bar{B}_h)^\top$. By the definition of $S_n(6)$ and Lemma H.12, there is a (random) function $\Delta \in \mathcal{D}_6$ such that

$$S_n(6) \geq -\frac{\sum_{i=1}^n \Delta\left(U_i\right) h'(\widehat{\phi}_n\left(U_i\right)) E_i}{\|\Delta\|_n \|o_h\|}$$

$$\overset{(44)}{\geq} \frac{\sum_{i=1}^n \Delta\left(U_i\right) h'(\widehat{\phi}_n\left(U_i\right)) \left[h(\phi_0\left(U_i\right)) - h(\widehat{\phi}_n\left(U_i\right))\right]}{\|\Delta\|_n \|o_h\|}$$

$$= (\|\Delta\|_n \|o_h\|)^{-1} \left[\sum_{i \in I_\Delta \cap J} + \sum_{i \in I_\Delta \cap J^c} + \sum_{i \in I_\Delta^c \cap J} + \sum_{i \in I_\Delta^c \cap J^c}\right]$$

$$\overset{(46)}{\geq} (\|\Delta\|_n \|o_h\|)^{-1} \left[u_v^2 \sum_{i \in I_\Delta \cap J} \Delta(U_i)(\phi_0(U_i) - \widehat{\phi}_n(U_i)) + \underline{B}_h^2 \sum_{i \in I_\Delta^c \cap J} \Delta(U_i)(\phi_0(U_i) - \widehat{\phi}_n(U_i)) + \right.$$

$$\left. + \underline{B}_h^2 \sum_{i \in I_\Delta \cap J^c} \Delta(U_i)(\phi_0(U_i) - \widehat{\phi}_n(U_i)) + \bar{B}_h^2 \sum_{i \in I_\Delta^c \cap J^c} \Delta(U_i)(\phi_0(U_i) - \widehat{\phi}_n(U_i))\right]$$

$$\overset{(45)}{\geq} \frac{\mathscr{C}}{4\bar{B}_h} \delta_n^\alpha (\|\Delta\|_n \|o_h\|)^{-1} \left[\bar{B}_h^2 \sum_{i \in I_\Delta \cap J} \Delta(U_i)^2 + \underline{B}_h^2 \sum_{i \in I_\Delta^c \cap J} \Delta(U_i)^2 + \right.$$

$$\left. + \underline{B}_h^2 \sum_{i \in I_\Delta \cap J^c} \Delta(U_i)^2 + \bar{B}_h^2 \sum_{i \in I_\Delta^c \cap J^c} \Delta(U_i)^2\right]$$

$$\geq \frac{\mathscr{C}}{4\bar{B}_h} \delta_n^\alpha \|\Delta\|_n \|o_h\|^{-1} \underline{B}_h^2$$

$$\overset{(45)}{\geq} \frac{\mathscr{C}}{4u_v} \delta_n^\alpha \left(n\widetilde{\mathscr{M}}_n\left(\delta_n/2\right)/4\right)^{1/2} \|o_h\|^{-1} \underline{B}_h^2.$$

Consequently,

$$\mathscr{C} \leq 4\underline{B}_h^{-2} \|o_h\| \delta_n^{-\alpha} \left(n\widetilde{\mathscr{M}}_n\left(\delta_n/2\right)/4\right)^{-1/2} S_n(6).$$

Now consider the event $B_{6,n,m}$ in Lemma H.10 and the event $A_{n,\gamma}$ in Lemma H.8. For $\gamma > 4$ define

$$G_{n,\gamma} = A_{n,\gamma} \cap B_{6,n,\gamma},$$

which holds with probability at least $1 - 2n^{2-\gamma}$. In $G_{n,\gamma}$ we have

$$\mathscr{C} \leq 4\bar{B}_h \|o_h\| \underline{B}_h^{-2} \delta_n^{-\alpha} \left((C_{\mathscr{M}}(1 - \epsilon_n)/8 + o(1))n\delta_n\right)^{-1/2} S_n(6)$$

$$= 4\bar{B}_h \|o_h\| \underline{B}_h^{-2} \left((C_{\mathscr{M}}(1 - \epsilon_n)/8 + o(1))\right)^{-1/2} (\log(n))^{-1/2} S_n(6)$$

$$\lesssim 4\bar{B}_h \|o_h\| \underline{B}_h^{-2} (C_{\mathscr{M}}/8)^{-1/2} (\log(n))^{-1/2} S_n(6),$$

where we used that $o(1)$ is a positive quantity appearing in the denominator. More precisely we have that

$$\mathcal{V}_n(\mathscr{C}) = \left\{\frac{\sup_{[a+\delta_n, b-\delta_n]} |\psi_0 - \widehat{\psi}_n|}{\delta_n^\alpha} \geq \mathscr{C}\right\} \subseteq \left\{S_n(6) \geq \mathscr{C}(4\bar{B}_h)^{-1} \|o_h\|^{-1} \underline{B}_h^2 (C_{\mathscr{M}}/8)^{1/2} (\log n)^{1/2}\right\},$$

then with probability less than $2n^{2-\gamma}$ we have $\sup_{[a+\delta_n, b-\delta_n]} |\psi_0 - \widehat{\psi}_n| \geq C_{\mathscr{C}} \gamma \delta_n^\alpha$ provided (by Lemma H.10)

$$\mathscr{C}(4\bar{B}_h)^{-1} \|o_h\|^{-1} \underline{B}_h^2 (C_{\mathscr{M}}/8)^{1/2} \geq \gamma\sigma\sqrt{7},$$

More precisely, the condition on $\mathscr{C}$ is that $\mathscr{C} \geq \max\left\{\widetilde{\mathscr{C}}^{-\alpha} \bar{B}_h, \frac{\gamma\sigma\sqrt{7}}{(4\bar{B}_h)^{-1}\|o_h\|^{-1}\underline{B}_h^2(C_{\mathscr{M}}/8)^{1/2}}\right\}$, where $\gamma > 4$. Using that $o_h = (\underline{B}_h, \bar{B}_h)^\top$ and $\widetilde{\mathscr{C}}^\alpha = \frac{\alpha}{4^{1+\alpha}L}$ (from Lemma H.11), this condition becomes

$$\mathscr{C} \geq \max\left\{\frac{4^{1+\alpha}L}{\alpha} \bar{B}_h, \frac{4\sqrt{7}\bar{B}_h\gamma\sigma\sqrt{\bar{B}_h^2 + \underline{B}_h^2}}{\underline{B}_h^2(C_{\mathscr{M}}/8)^{1/2}}\right\}$$

## H.4 PROOF OF LEMMA H.10

Assumption H.3 implies the following inequality

$$\mathbb{P}\left\{\frac{\left|\sum_{i=1}^n \zeta\left(U_i\right) h'(\widehat{\phi}_n\left(U_i\right))E_i\right|}{\|\zeta\|_n\|o_h\|}\geq\eta\right\}\leq 2\exp\left(-\frac{\eta^2}{2\sigma^2}\right), \tag{47}$$

for any function $\zeta$ with $\|\zeta\|_n>0$ and arbitrary $\eta\geq 0$. Indeed, since $0<\underline{B}_h\leq h'(\widehat{\phi}_n\left(U_i\right))\leq\bar{B}_h$, we have

$$\sum_{i=1}^n \zeta\left(U_i\right)h'(\widehat{\phi}_n\left(U_i\right))E_i\leq\bar{B}_h\sum_{i\in I}\zeta\left(U_i\right)E_i+\underline{B}_h\sum_{i\in I^c}\zeta\left(U_i\right)E_i, \tag{48}$$

where $I=\{i:\zeta\left(U_i\right)E_i>0\}$ and $I^c=\{i:\zeta\left(U_i\right)E_i\leq 0\}$. We get

$$\mathbb{P}\left\{\frac{\sum_{i=1}^n\zeta\left(U_i\right)h'(\widehat{\phi}_n\left(U_i\right))E_i}{\|\zeta\|_n\|o_n\|}\geq\eta\right\}$$

$$\leq\mathbb{P}\left\{\bar{B}_h\sum_{i\in I}\zeta\left(U_i\right)E_i+\underline{B}_h\sum_{i\in I^c}\zeta\left(U_i\right)E_i\geq\eta\|\zeta\|_n\|o_h\|\right\}$$

$$\leq\exp\left(-\eta\|\zeta\|_n\|o_h\|\right)\mathbb{E}\left[\exp\left\{\bar{B}_h\sum_{i\in I}\zeta\left(U_i\right)E_i+\underline{B}_h\sum_{i\in I^c}\zeta\left(U_i\right)E_i\right\}\right]$$

$$=\exp\left(-\eta\|\zeta\|_n\|o_h\|\right)\prod_{i\in I}\mathbb{E}\left[\exp\left\{\bar{B}_h\zeta\left(U_i\right)E_i\right\}\right]\prod_{i\in I^c}\mathbb{E}\left[\exp\left\{\underline{B}_h\zeta\left(U_i\right)E_i\right\}\right]$$

$$\leq\exp\left(-\eta\|\zeta\|_n\|o_h\|\right)\prod_{i\in I}\left[\exp\left\{\bar{B}_h^2\zeta^2\left(U_i\right)\sigma^2/2\right\}\right]\prod_{i\in I^c}\left[\exp\left\{\underline{B}_h^2\zeta^2\left(U_i\right)\sigma^2/2\right\}\right]$$

$$\leq\exp\left(-\eta\|\zeta\|_n\|o_h\|\right)\exp\left\{\|o_h\|^2\|\zeta\|_n^2\sigma^2/2\right\}$$

$$\leq\exp\left(-\eta^2/(2\sigma^2)\right),$$

where we used that $-\eta x\|o_h\|+\|o_h\|^2 x^2\sigma^2/2$ is maximized at $x=\eta/(\sigma^2\|o_h\|)$. Equivalently we get

$$\mathbb{P}\left\{-\frac{\sum_{i=1}^n\zeta\left(U_i\right)h'(\widehat{\phi}_n\left(U_i\right))E_i}{\|\zeta\|_n\|o_n\|}\geq\eta\right\}\leq\exp\left(-\frac{\eta^2}{2\sigma^2}\right),$$

and

$$\mathbb{P}\left\{\frac{\left|\sum_{i=1}^n\zeta\left(U_i\right)h'(\widehat{\phi}_n\left(U_i\right))E_i\right|}{\|\zeta\|_n\|o_h\|}\geq\eta\right\}\leq 2\exp\left(-\frac{\eta^2}{2\sigma^2}\right).$$

For $1\leq j\leq k\leq n$, let

$$h_{jk}^{(1)}(u):=1\left\{u\in[U_j,U_k]\right\}\frac{u-U_j}{U_k-U_j}\quad\text{and}\quad h_{jk}^{(2)}(u):=1\left\{u\in[U_j,U_k]\right\}\frac{U_k-u}{U_k-U_j},$$

if $U_j<U_k$. Otherwise let $\zeta_{jk}^{(1)}(u):=1\left\{u=U_k\right\}$ and $\zeta_{jk}^{(2)}(u):=0$.

This defines a collection $\Phi$ of at most $n^2$ different nonzero functions $\zeta_{jk}^{(e)}$. Then for any fixed $\gamma_o>2$, (47) implies

$$\mathbb{P}\left\{S_n\leq\gamma_o\sigma(\log n)^{1/2}\right\}\geq 1-\frac{2}{n^{\gamma_o^2/2}},$$

where

$$S_n:=\max_{\zeta\in\Phi}\frac{\left|\sum_{i=1}^n\zeta\left(U_i\right)h'(\widehat{\phi}_n\left(U_i\right))E_i\right|}{\|\zeta\|_n\|o_h\|}.$$

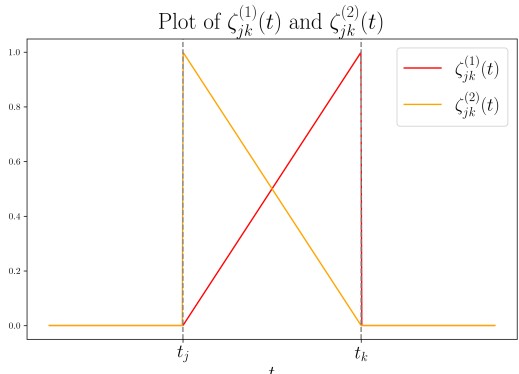

Figure 1

For let $G_n(\zeta) := \|\zeta\|_n^{-1} \sum_{i=1}^n h(U_i) E_i$. Then by (47),

$$\mathbb{P}\left\{ S_n \geq \gamma_o \sigma (\log n)^{1/2} \right\} \leq \sum_{\zeta \in \Phi} \mathbb{P}\left\{ |G_n(\zeta)| \geq \gamma_o \sigma (\log n)^{1/2} \right\} \tag{49}$$

$$\leq 2n^2 \exp\left( -\gamma_o^2 \log(n)/2 \right)$$

$$= 2n^{-\frac{4-\gamma_o^2}{2}}$$

$$\to 0 \text{ as } n \to \infty,$$

because $\gamma_o > 2$. Recall that $\mathcal{D}_m$ is the family of all continuous, piecewise linear functions on $\mathbb{R}$ with at most $m$ knots. For any $\Delta \in \mathcal{D}_m$, there are $m' \leq 2m + 2$ disjoint intervals on which $\Delta$ is either linear and non-negative, or linear and non-positive. For one such interval $B$ with $\mathcal{M}_n(B) > 0$ let $\{U_1, \ldots, U_n\} \cap B = \{U_j, \ldots, U_k\}$. Then

$$\Delta(u) = \Delta(U_k)\, \zeta_{jk}^{(1)}(u) + \Delta(U_j)\, \zeta_{jk}^{(2)}(u) \quad \text{for } u \in [U_j, U_k].$$

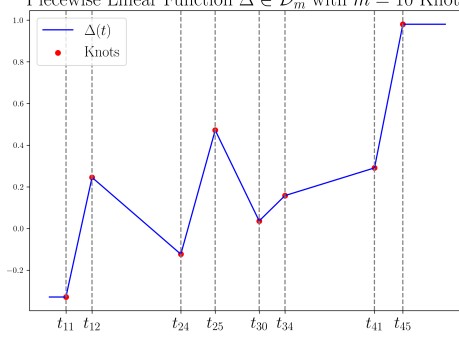

Figure 2

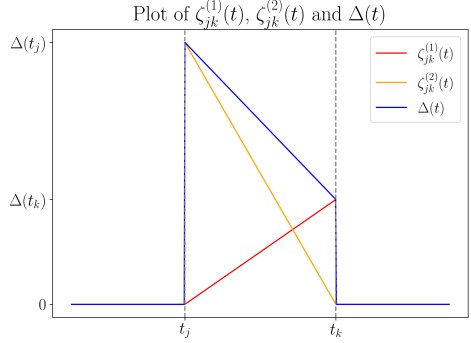

Figure 3

This shows that there are real coefficients $\lambda_1, \ldots, \lambda_{4m+4}$ and functions $\zeta_1, \ldots, \zeta_{4m+4}$ in $\Phi$ such that $\Delta = \sum_{j=1}^{4m+4} \lambda_j \zeta_j$ on $\{U_1, \ldots, U_n\}$, and $\lambda_j \lambda_k \zeta_j \zeta_k \geq 0$ for all pairs $(j, k)$. Consequently,

$$
\frac{\left| \sum_{i=1}^n \Delta(U_i) h'(\widehat{\phi}_n(U_i)) E_i \right|}{\|\Delta\|_n \|o_h\|} \leq \frac{\sum_{j=1}^{4m+4} |\lambda_j| \left| \sum_{i=1}^n \zeta_j(U_i) h'(\widehat{\phi}_n(U_i)) E_i \right|}{\left( \sum_{j=1}^{4m+4} \lambda_j^2 \|\zeta_j\|_n^2 \right)^{1/2} \|o_h\|}
$$

$$
\leq \frac{\sum_{j=1}^{4m+4} |\lambda_j| \|\zeta_j\|_n}{\left( \sum_{j=1}^{4m+4} \lambda_j^2 \|\zeta_j\|_n^2 \right)^{1/2}} S_n
$$

$$
\leq (4m+4)^{1/2} S_n.
$$

The last inequality is due to Cauchy-Schwarz inequality

$$
\sum_{j=1}^{4m+4} |\lambda_j| \|h_j\|_n \times 1 = \left( \sum_{j=1}^{4m+4} \lambda_j^2 \|h_j\|_n^2 \right)^{1/2} \left( \sum_{j=1}^{4m+4} 1^2 \right)^{1/2}.
$$

Thus by (49) we have that

$$
\mathbb{P} \left\{ \sup_{m>0} \sup_{\Delta \in \mathcal{D}_m} \frac{\left| \sum_{i=1}^n \Delta(U_i) h'(\widehat{\phi}_n(U_i)) E_i \right|}{\|\Delta\|_n \|o_h\| (m+1)^{1/2}} \leq 2\gamma_o \sigma (\log n)^{1/2} \right\} \geq 1 - 2n^{-\frac{4-\gamma_o^2}{2}},
$$

Setting $\gamma = 2\gamma_o$ we get the result.

## H.5 Proof of Lemma H.11

We divide the proof in 2 parts. **Part A**: $(\widehat{\phi} - \phi)(t_o) \geq \xi$ for some $t_o \in [a, b]$. **Part B**: $(\widehat{\phi} - \phi)(t_o) \geq \xi$ for some $t_o \in [a + \delta, b - \delta]$.

**Part A**

Suppose that $(\widehat{\phi} - \phi)(t_o) \geq \xi$ for some $t_o \in [a, b]$. Without loss of generality let $t_o \leq (a+b)/2$. Define the linear function

$$
\widetilde{\phi}(\cdot) := \begin{cases} \phi(t_o) & \text{if } \alpha = 1 \\ \phi(t_o) + \phi'(t_o)(\cdot - t_o) & \text{if } \alpha > 1, \end{cases}
$$

Define the concave function $\Phi := \widehat{\phi} - \widetilde{\phi}$. Now, note that by Assumption H.4 we have

$$
|\widetilde{\phi} - \phi| \leq L|t - t_o|^\alpha / \alpha, \tag{50}
$$

Now, since by assumption $\sup_{t \in [a,d]} \widehat{\phi} - \phi \geq \xi$ we have

$$
\Phi(t_o) = \widehat{\phi}(t_o) - \phi(t_o) \geq \xi. \tag{51}
$$

Now let $0 < \delta \leq (b-a)/8$.

- **Step 1.A**. Since $\Phi$ is concave, it follows that if $\Phi(t_o + \delta) \geq \xi/2$ then, joint with (51), $\Phi \geq \xi/2$ on $[t_o, t_o + \delta]$.

- **Step 2.A**. Now assume $\Phi(t_o + \delta) < \xi/2$. By concavity of $\Phi$ we have

$$
\Phi(t_o) \underbrace{\leq}_{\Phi \text{ concave}} \Phi(t_o+\delta) - \delta \Phi'(t_o+\delta) \Rightarrow \Phi'(t_o+\delta) \leq \delta^{-1}(\underbrace{\Phi(t_o+\delta)}_{<\xi/2} - \underbrace{\Phi(t_o)}_{\geq \xi}) < -\delta^{-1}\xi/2.
$$

Consequently, for $t \geq t_o + 3\delta$

$$
\Phi(t) \underbrace{\leq}_{\Phi \text{ concave}} \underbrace{\Phi(t_o+\delta)}_{<\xi/2} + \underbrace{(t - t_o - \delta)}_{\geq 2\delta} \underbrace{\Phi'(t_o+\delta)}_{<-\delta^{-1}\xi/2}
$$

$$
< \xi/2 - 2\delta\delta^{-1}(\xi/2) = -\xi/2.
$$

- **Step 1.A + 2.A**: $\Phi \geq \xi/2$ or $\Phi \leq \xi/2$ on some interval $J \subset [t_o, t_o + 4\delta]$ with length $\delta$. Using this fact and (50), we have that $t \in J$

$$\widehat{\phi} - \phi = \Phi + \widetilde{\phi} - \phi \quad \text{or} \quad \phi - \widehat{\phi} = \phi - \widetilde{\phi} - \Phi,$$

is greater or equal than

$$\xi/2 - L(t - t_o)^\alpha/\alpha \geq \xi/2 - L(4\delta)^\alpha/\alpha,$$

which is greater or equal than $\xi/4$ provided that $\delta \leq \frac{1}{4} \left( \frac{\alpha\xi}{4L} \right)^{1/\alpha}$.

**Part B**

Now suppose that $(\phi - \widehat{\phi})(t_o) \geq \xi$ for some $\xi > 0$ and $t_o \in [a + \delta, b - \delta]$, where $0 < \delta \leq (b-a)/2$. By Assumption H.4 and concavity of both $\phi, \widehat{\phi}$ there exist numbers $\gamma, \widehat{\gamma}$ such that

$$\phi(t) \geq \phi(t_o) + \gamma(t - t_o) - L|t - t_o|^\alpha/\alpha,$$

and

$$\widehat{\phi}(t) \leq \widehat{\phi}(t_o) + \widehat{\gamma}(t - t_o).$$

Thus

$$(\phi - \widehat{\phi})(t) \geq \xi + (\gamma - \widehat{\gamma})(t - t_o) - L|t - t_o|^\alpha/\alpha \geq \xi - L\delta^\alpha,$$

for all $t$ in the interval $[t_o, t_o + \delta]$ or $[t_o - \delta, t_o]$, depending on the sign of $\gamma - \widehat{\gamma}$. Moreover, $\xi - L\delta^\alpha/\alpha \geq \xi/4$, provided that $\delta \leq \left( \frac{3\alpha\xi}{4L} \right)^{1/\alpha}$.

# I  REMARKS

## I.1  REMARK ON ASSUMPTION 3.2

In this section, we discuss sufficient conditions under which Assumption 3.2 holds. Under twice differentiability of $\psi_i$, the strong concavity of $\mathrm{rev}_i\left(\cdot \mid \mathbf{p}_{-i}\right)$ uniformly over $\mathbf{p}_{-i}$, is guaranteed if there exists a value $\mu_i > 0$, such that $-\partial_{p^2}^2 \mathrm{rev}_i(p \mid \mathbf{p}_{-i}) \geq \mu_i$ for every $p \in \mathcal{P}_i$, uniformly in $\mathbf{p}_{-i} \in \mathcal{P}_{-i}$. Uniformity in $\mathbf{p}_{-i} \in \mathcal{P}_{-i}$ means that $\mu_i$ does not depends on $\mathbf{p}_{-i}$. Here we give some examples where this condition is satisfied.

(1) **Linear demand models (Li et al., 2024)**. In this case $\mathrm{rev}_i(p \mid \mathbf{p}_{-i}) = p(\alpha_i - \beta_i p + \boldsymbol{\gamma}_i^\top \mathbf{p}_{-i})$, and $-\partial_{p^2}^2 \mathrm{rev}_i(p \mid \mathbf{p}_{-i}) = 2\beta_i > 0$.

(2) **Concave demand models**. If $\psi_i$ is such that $\psi_i' \geq B_{\psi_i'}$ for some $B_{\psi_i'} > 0$ as in Assumption 3.1 and is concave, which implies $\partial_p \mathrm{rev}_i(p \mid \mathbf{p}_{-i}) = \psi_i(-\beta_i p + \boldsymbol{\gamma}_i^\top \mathbf{p}_{-i}) - \beta_i p \psi_i'(-\beta_i p + \boldsymbol{\gamma}_i^\top \mathbf{p}_{-i})$, then

$$-\partial_{p^2}^2 \mathrm{rev}_i(p \mid \mathbf{p}_{-i}) = 2\psi_i'(-\beta_i p + \boldsymbol{\gamma}_i^\top \mathbf{p}_{-i}) - \beta_i^2 p \psi_i''(-\beta_i p + \boldsymbol{\gamma}_i^\top \mathbf{p}_{-i})$$
$$\geq 2\psi_i'(-\beta_i p + \boldsymbol{\gamma}_i^\top \mathbf{p}_{-i}) \geq 2B_{\psi_i'} > 0.$$

Note that requiring concavity of $\psi_i$ is equivalent to requiring 1-concavity of $\psi_i$, then, by the inclusion property of $s$-concave functions, any (positive and strictly increasing) $s_i$-concave $\psi_i$ with $s_i \geq 1$ satisfies Assumption 3.2.

(3) **s-concave demand functions**. Suppose that $\psi_i$ satisfies Assumption 3.1 and Assumption 3.4. Then, by Proposition 3.5, $\psi_i$ is $s_i$-concave for some $s_i > -1$, and by Lemma B.1 this is equivalent to $\psi_i'' \leq (1 - s_i)(\psi_i')^2/\psi_i$. This implies that

$$- \partial_{p^2}^2 \mathrm{rev}_i(p \mid \mathbf{p}_{-i})$$
$$= 2\psi_i'(-\beta_i p + \boldsymbol{\gamma}_i^\top \mathbf{p}_{-i}) - \beta_i^2 p \psi_i''(-\beta_i p + \boldsymbol{\gamma}_i^\top \mathbf{p}_{-i})$$
$$\geq 2\psi_i'(-\beta_i p + \boldsymbol{\gamma}_i^\top \mathbf{p}_{-i}) - \beta_i^2 p(1 - s_i)[\psi_i'(-\beta_i p + \boldsymbol{\gamma}_i^\top \mathbf{p}_{-i})]^2/\psi_i(-\beta_i p + \boldsymbol{\gamma}_i^\top \mathbf{p}_{-i})$$
$$= \psi_i'(-\beta_i p + \boldsymbol{\gamma}_i^\top \mathbf{p}_{-i}) \left\{2 - \beta_i^2 p(1 - s_i)\psi_i'(-\beta_i p + \boldsymbol{\gamma}_i^\top \mathbf{p}_{-i})/\psi_i(-\beta_i p + \boldsymbol{\gamma}_i^\top \mathbf{p}_{-i})\right\},$$

and since $\psi_i' \geq B_{\psi_i'} > 0$ a sufficient condition is that

$$2 - \beta_i^2 p(1 - s_i)\psi_i'(-\beta_i p + \boldsymbol{\gamma}_i^\top \mathbf{p}_{-i})/\psi_i(-\beta_i p + \boldsymbol{\gamma}_i^\top \mathbf{p}_{-i}) \geq C_i,$$

for some constant $C_i > 0$. Note that since $\psi_i'/\psi_i$ is non-negative, for $s_i \geq 1$ the condition is trivially satisfied with $C_i = 2$, indeed this condition coincides with the case studied before for concave demand functions. When $s_i \in (-1, 1)$ the condition becomes

$$p\frac{d}{dp}\left\{\log \circ \psi_i(-\beta_i p + \boldsymbol{\gamma}_i^\top \mathbf{p}_{-i})\right\} = -\beta_i p\frac{\psi_i'(-\beta_i p + \boldsymbol{\gamma}_i^\top \mathbf{p}_{-i})}{\psi_i(-\beta_i p + \boldsymbol{\gamma}_i^\top \mathbf{p}_{-i})} \geq -K_i,$$

where we set $K_i = \frac{2 - C_i}{\beta_i(1 - s_i)}$. Note that the left-hand side is always negative, then, a necessary condition such that the inequality holds is that the right-hand side is negative too, i.e., if $0 < C_i \leq 2$, or $K_i > 0$. For example, suppose that $\psi_i$ is 0-concave (i.e. log-concave), that is $\psi_i = e^{\phi_i}$ for some increasing concave function $\phi_i$, where $\phi_i$ needs to be increasing in order to guarantee that $\psi_i$ is increasing. The condition becomes

$$\beta_i p\phi'(-\beta_i p + \boldsymbol{\gamma}_i^\top \mathbf{p}_{-i}) = -p\frac{d}{dp}\log \psi_i(-\beta_i p + \boldsymbol{\gamma}_i^\top \mathbf{p}_{-i}) \leq K_i,$$

and it's easy to see that any $\phi_i(x) = ax$ for $a > 0$ sufficiently small works. Indeed the condition becomes $p \leq K_i/(\beta_i a)$ for all $p \in \mathcal{P}_i$, which holds as long as $K_i/(\beta_i a) \geq \overline{p}_i$, or $a \leq K_i/(\beta_i \overline{p}_i)$. Another example is $\psi_i(x) = -1/(x + b)$ for $b$ sufficiently large, indeed the condition becomes $\frac{p}{(-\beta_i p + \boldsymbol{\gamma}_i^\top \mathbf{p}_{-i} + b)^2} \leq K_i/\beta_i$ or $p \leq K_i^*$ for some $K_i^*(b) > 0$, and $b$ must by such that $K_i^* \geq \overline{p}_i$.

## I.2 REMARK 4.1 ON THE COMMON EXPLORATION PHASE

To highlight the importance of a common exploration phase, we demonstrate that no seller has an incentive to extend or shorten the exploration phase. If firm $i$ prolongs its exploration phase, it risks *incomplete learning* of its model parameters (Keskin & Zeevi, 2018), leading to inaccurate parameter estimation and, consequently, a loss in revenue. Conversely, if firm $i$ shortens its exploration phase, it may achieve consistent estimation but at the cost of higher regret due to insufficient exploratory data. For a better illustration, consider a scenario where firm $i$ extends its exploration phase to $\tau' > \tau$ while all other firms use a phase of length $\tau$. When firm $i$ estimates its parameters using data $\{(\mathbf{p}^{(t)}, y_i^{(t)})\}_{t \leq \tau'}$, the data from times $\tau + 1$ to $\tau'$ are no longer iid. In this later period, all other firms have already started their exploration, causing the prices $\{\mathbf{p}_{-i}^{(t)}\}_{t=\tau+1,\ldots,\tau'}$ to be dependent on the earlier data $\{\mathbf{p}_{-i}^{(t)}\}_{t \leq \tau}$. This dependency can result in inconsistent parameter estimation and a loss of efficiency for firm $i$. Conversely, if firm $i$ opts for a shorter exploration phase than $\tau$, it may achieve consistent estimation, but the reduced amount of exploratory data could lead to lower efficiency compared to firms that adhere to the full exploration phase.

## I.3 REMARK 4.2 ON THE NEED FOR TWO DIFFERENT PHASES FOR MODEL ESTIMATION

In principle, one could estimate $(\boldsymbol{\theta}_i, \psi_i)$ jointly using the full exploration phase. For instance, Balabdaoui et al. (2019) employ a profile least squares approach to achieve $L^2$ convergence for the joint LSE of $(\boldsymbol{\theta}_i, \psi_i)$ based on the data $\{(y_i^{(t)}, \mathbf{p}^{(t)})\}_{t \leq \tau}$. However, in our context, $L^2$ convergence alone does not suffice to establish an upper bound on the total expected regret. To elaborate, let $(\boldsymbol{\theta}_i^{JLS}, \psi_i^{JLS})$ denote the joint estimator proposed in Balabdaoui et al. (2019) based on the data $\{(y_i^{(t)}, \mathbf{p}^{(t)})\}_{t \leq \tau}$. Their result applied to our setting gives the following $L^2$ convergence rate:

$$\left( \int_{\mathcal{P}} \left( \psi_i^{JLS}(\langle \mathbf{p}, \boldsymbol{\theta}_i^{JLS} \rangle) - \psi_i(\langle \mathbf{p}, \boldsymbol{\theta}_i \rangle) \right)^2 d\mathscr{D}(\mathbf{p}) \right)^{1/2} = O_p\left( \tau^{-1/3} \right).$$

Here, the convergence is measured with respect to the measure $\mathscr{D}$, which governs the distribution of prices during the exploration phase. Consequently, the $L^2$ distance between $(\boldsymbol{\theta}_i, \psi_i)$ and $(\boldsymbol{\theta}_i^{JLS}, \psi_i^{JLS})$ can only be evaluated for prices distributed according to $\mathscr{D}$. However, during the exploitation phase ($t \geq \tau + 1$), the distribution of prices $\mathbf{p}^{(t)}$ may differ from $\mathscr{D}$, making it impossible to evaluate the performance of the estimators during this phase in an $L^2$ sense. A uniform convergence result, which does not depend on any specific price distribution, solves this issue. However, achieving uniform convergence of the form

$$\sup_{\mathbf{p} \in \mathcal{P}} \left| \psi_i^{JLS}(\langle \mathbf{p}, \boldsymbol{\theta}_i^{JLS} \rangle) - \psi_i(\langle \mathbf{p}, \boldsymbol{\theta}_i \rangle) \right| \tag{52}$$

is challenging when $(\boldsymbol{\theta}_i^{JLS}, \psi_i^{JLS})$ are estimated jointly. To overcome this difficulty, our approach is to first estimate $\boldsymbol{\theta}_i$ and then, conditional on this estimator, separately estimate $\psi_i$ using an independent dataset. This two-step procedure ensures conditional independence and yields consistent estimators, addressing the difficulties of the joint estimation strategy.

## J SIMULATIONS

### J.1 CONTRACTION CONSTANT VARYING REGIMES

We evaluate the performance of Algorithm 1 in the sequential price competition model with $N \in \{2, 4, 6\}$ sellers. Each seller $i \in \mathcal{N} = \{1, 2, \ldots, N\}$ sets prices supported on $\mathcal{P} = [0, 3]^{\times N}$. We examine how convergence behavior changes as the contraction constant $L_{\boldsymbol{\Gamma}}$ varies –specifically, when it is close to 0, near 0.5, or approaches 1.

Recall that the contraction constant of the Best Response operator is

$$L_{\boldsymbol{\Gamma}} = \sup_{i \in \mathcal{N}} \|g_i'\|_{\infty} \frac{\|\gamma_i\|_1}{\beta_i}$$

which simplifies to $L_{\boldsymbol{\Gamma}} = \sup_{i \in \mathcal{N}} \frac{\|\gamma_i\|_1}{\beta_i}$, when the link functions $\psi_i$ are log-concave (see Remark 3.8). In our experiments, we take

$$\psi_i(u) = \Phi\left(\frac{2}{i} u\right), \quad i = 1, 2, \ldots, N,$$

where $\Phi$ is the cumulative distribution function of a standard Gaussian random variable. Since $\Phi$ is log-concave, the simplification applies. We now identify the parameter regimes for $\beta_i$ that yield $L_{\boldsymbol{\Gamma}} \approx 0$ and $L_{\boldsymbol{\Gamma}} \approx 1$.

**Case 1:** $L_{\boldsymbol{\Gamma}} \downarrow 0$. In the extreme case $L_{\boldsymbol{\Gamma}} \downarrow 0$, we must have $\beta_i \uparrow 1$ for all $i \in \mathcal{N}$, which forces $\gamma_i \downarrow 0$ component-wise.

**Case 2:** $L_{\boldsymbol{\Gamma}} \uparrow 1$. To approach the opposite extreme, $L_{\boldsymbol{\Gamma}} \uparrow 1$, consider

$$\sup_{i \in \mathcal{N}} \sup_{\beta_i > 0, \gamma_i \in \mathbb{R}^{n-1}} \frac{\|\gamma_i\|_1}{\beta_i} < 1, \quad \text{such that} \quad \beta_i^2 + \|\gamma_i\|_2^2 = 1, \quad \forall i \in \mathcal{N}.$$

Without loss of generality, fix $i = 1$. By the Cauchy–Schwarz inequality,

$$\|\gamma_1\|_1 \leq \sqrt{N - 1} \|\gamma_1\|_2.$$

So the largest possible $\|\gamma_1\|_1$ occurs when entries of $\gamma_1$ are equally distributed. In that case, the requirement becomes

$$\sqrt{N - 1} \|\gamma_1\|_2 < \beta_1.$$

Since $\|\gamma_1\|_2 = \sqrt{1 - \beta_1^2}$, this becomes $\sqrt{N - 1} \sqrt{1 - \beta_1^2} < \beta_1$. Squaring both sides gives

$$(N - 1)(1 - \beta_1^2) < \beta_1^2 \quad \Longleftrightarrow \quad \beta_1^2 > \frac{N - 1}{N}.$$

Thus, $L_{\boldsymbol{\Gamma}} \uparrow 1$ as $\beta_1 \downarrow \sqrt{\frac{N-1}{N}}$. In particular, we have the following scheme:

$$N = 2: \quad \beta_1 \downarrow \sqrt{\frac{1}{2}} \approx 0.707 \implies L_{\boldsymbol{\Gamma}} \uparrow 1,$$

$$N = 4: \quad \beta_1 \downarrow \sqrt{\frac{3}{4}} \approx 0.866 \implies L_{\boldsymbol{\Gamma}} \uparrow 1,$$

$$N = 6: \quad \beta_1 \downarrow \sqrt{\frac{5}{6}} \approx 0.913 \implies L_{\boldsymbol{\Gamma}} \uparrow 1.$$

A summary of the two extreme cases can be found in Table 3.

**Simulation design.** Given the discussion above, and the table in Table 3, for each $N \in \{2, 4, 6\}$, we run simulations under three representative values of $\beta_i$, yielding a total of 9 settings:

- **N=2**. $\beta_i = 0.71, 0.84, 0.97$.
- **N=4**. $\beta_i = 0.87, 0.93, 0.99$.
- **N=6**. $\beta_i = 0.92, 0.955, 0.99$.

Table 3: Extreme cases for $\beta_i$ and corresponding behavior of $L_{\mathbf{\Gamma}}$.

| Case | Condition on $\beta_i$ | Condition on $\gamma_i$ |
|---|---|---|
| $L_{\mathbf{\Gamma}} \downarrow 0$ | $\beta_i \uparrow 1, \forall i \in \mathcal{N}$ | $\|\gamma_i\|_1 \downarrow 0, \forall i \in \mathcal{N}$ |
| $L_{\mathbf{\Gamma}} \uparrow 1$ | $\beta_1 \downarrow \sqrt{N-1/N}$ | $\gamma_{1j} \uparrow {}^1/\sqrt{N(N-1)}$ for $j \in \mathcal{N} \setminus \{1\}$ |
| $N = 2$ | $\beta_1 \downarrow \sqrt{1/2}$ | $\gamma_{1,2} \uparrow {}^1/\sqrt{2}$ |
| $N = 4$ | $\beta_1 \downarrow \sqrt{3/4}$ | $\gamma_{1,j} \uparrow {}^1/\sqrt{12}$ for $j = 2, 3, 4$ |
| $N = 6$ | $\beta_1 \downarrow \sqrt{5/6}$ | $\gamma_{1,j} \uparrow {}^1/\sqrt{30}$ for $j = 2, 3, 4, 5, 6$ |

In all experiments, demand noise follows $\mathrm{Unif}[-0.05, 0.05]$. The common exploration phase has length $\frac{2}{3}T^{5/7}$ and the length $\kappa_i$ of the first part of the exploration phase is determined using Equation (14) with $\mathscr{C}_i = 1$, $\mathscr{B}_i = 1$. During the first phase of the exploration phase, the values of $\mathbf{p}^{(t)}$ are samples from a multivariate Gaussian with mean

$$\mathbf{m} = \left(\frac{\bar{p}_i + \underline{p}_i}{2}\right)_{i \in \mathcal{N}}, \quad \Sigma = \mathrm{diag}\left(\sqrt{\frac{\bar{p}_i - \underline{p}_i}{2}}\right)_{i \in \mathcal{N}}.$$

For each horizon $T \in \{100, 400, 800, 1600, 3200\}$, we apply Algorithm 1 and repeat the simulation 30 times to obtain averages and $95\%$ confidence intervals.

The results are displayed in Figure 5, and a log-log version to highlight the convergence rate in Figure 6. As expected, convergence accelerates as the contraction constant approaches 1, in line with the intuition highlighted in (9).

## J.2 DIFFERENT EXPLORATION PHASES ACROSS SELLERS

In this experiment, we replicate the setup in Appendix J.1 but relax the common exploration phase assumption in order to test the robustness of our algorithm. An illustration of misaligned exploration phases is provided in Figure 4.

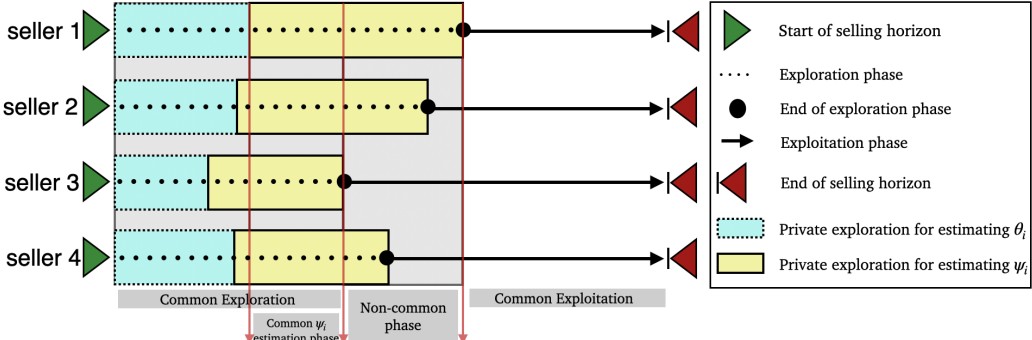

Figure 4: Illustration of our policy (Algorithm 1) with $N = 4$ sellers in sequential price competition under nonlinear demands. For each seller $i \in \{1, 2, 3, 4\}$, in their exploration phase (dotted line) of length $\tau_i$, they offer randomized prices following their distribution $\mathscr{D}_i$. Within the exploration phase, each seller has a private phase for estimating $\boldsymbol{\theta}_i$ (blue box with dotted border), with length $\tau_i \kappa_i$ and a private phase for estimating $\psi_i$ (yellow box with continued border line), with length $\tau_i(1 - \kappa_i)$. The $i$-th seller's price experiment ends at period $t = \tau_i$ (black circles). Subsequently, in their exploitation phase (represented by the continued black line), seller $i$ offers prices based on the estimators generated in the exploration phase.

We set the exploration length of each seller as

$$\tau_i := \tau_{\text{base}} + u_i \cdot \tau_{\text{base}}, \quad u_i \sim \mathrm{Unif}[0.25, 0.75],$$

where $\tau_{\text{base}} = \frac{2}{3}T^{5/7}$. The allocation parameter $\kappa_i$ for the first part of exploration is computed using Equation (14), with $\tau$ replaced by $\tau_i$ and fixed constants $\mathscr{C}_i = \mathscr{B}_i = 1$. For this simulation, we set $\beta_i = 0.8$ for $N = 2$, $\beta_i = 0.92$ for $N = 4$, and $\beta_i = 97$ for $N = 6$, which satisfy Assumption 3.6.

During the exploration phase, the prices $p_i^{(t)}$ are sampled independently across sellers from a Gaussian distribution $\mathcal{N}(m_i, \sigma_i)$ with

$$m_i = \frac{\bar{p}_i + \underline{p}_i}{2}, \quad \sigma_i = \sqrt{\frac{\bar{p}_i - \underline{p}_i}{2}}, \quad i \in \mathcal{N}.$$

For each horizon $T \in \{100, 400, 800, 1600, 3200\}$, we run Algorithm 1 and repeat the simulation 30 times, reporting averages and $95\%$ confidence intervals.

The results are reported in Figure 7, together with a log-log plot in Figure 6, which illustrate the convergence of prices to the Nash equilibrium and the decay of regret.

As anticipated, the estimation of $\psi_i$ deteriorates, in line with the intuition of Remark 4.1. Indeed, as the horizon $T$ grows, the exploration lengths $\tau_i$ increase for all sellers, which enlarges the non-common exploration phase (see Figure 7). Consequently, a larger number of price vectors $\mathbf{p}^{(t)}$ are no longer i.i.d., making parameter estimation increasingly inconsistent. Nevertheless, despite the adverse effect on the estimation of $\psi_i$, both convergence to the Nash equilibrium and regret performance remain robust, with empirical regret rates matching – or even surpassing – the theoretical benchmarks of $-1/7$ for equilibrium convergence and $5/7$ for regret.

## J.3 ROBUSTNESS TO MISSPECIFICATION OF $s_i$.

In this experiment, we replicate the setup of Appendix J.1, except that we fix $N = 4$ and set $\beta_i = 0.9$. We again take $\psi_i(u) = \Phi(2u/i)$ for $i = 1, 2, \ldots, N$, where $\Phi$ denotes the c.d.f. of a standard Gaussian random variable. Since $\Phi$ is log-concave, each $\psi_i$ is 0-concave, that is, $s_i = 0$. We then run Algorithm 1 under five different shape parameters $s_i' \in \{-0.2, -0.1, 0, 0.1, 0.2\}$. As shown in Figure 8, this misspecification of $s_i$ has no visible effect on either the NE convergence rate or the regret convergence rate. The small variations in the estimated slopes reported in the legend of Figure 8 are plausibly due to finite-sample noise ($T = 1600$).

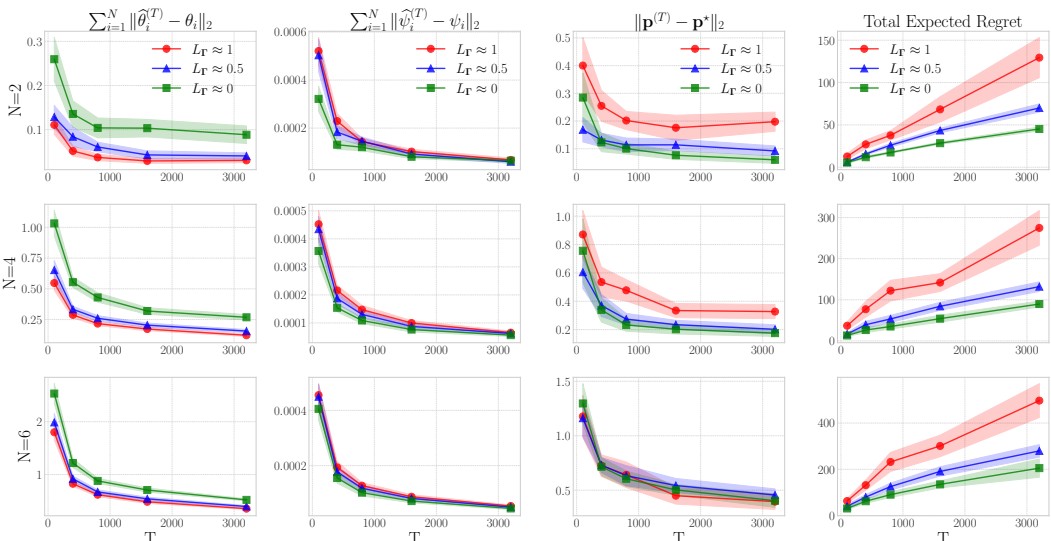

Figure 5: Performance of Algorithm 1 in sequential price competition with $N \in \{2, 4, 6\}$ sellers for different values of the contraction constant $L_{\Gamma}$.

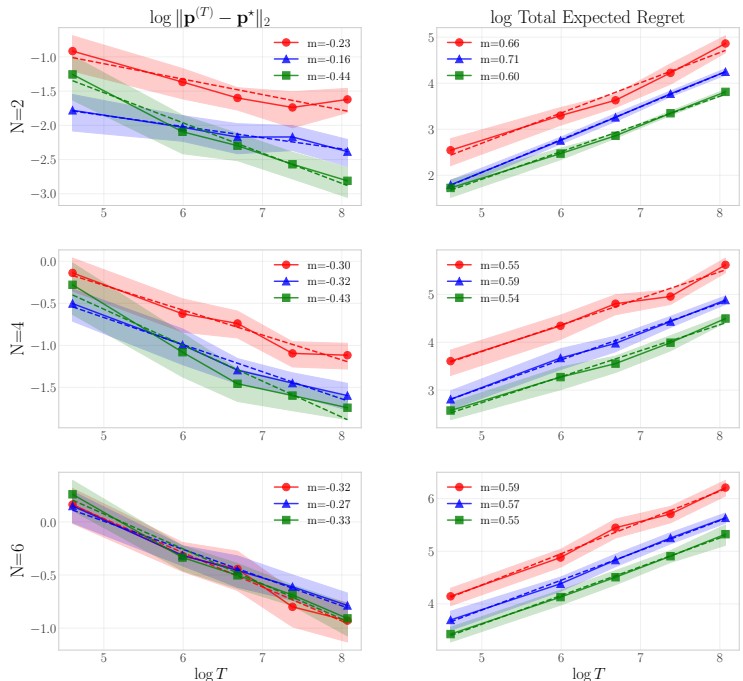

Figure 6: Log-log performance of Algorithm 1 in sequential price competition with $N \in \{2, 4, 6\}$ sellers for different values of the contraction constant $L_{\Gamma}$. The slopes, indicated as $m$, of the convergence to NE and the regret are always smaller than the corresponding theoretical upper bounds ($-1/7$ and $5/7$).

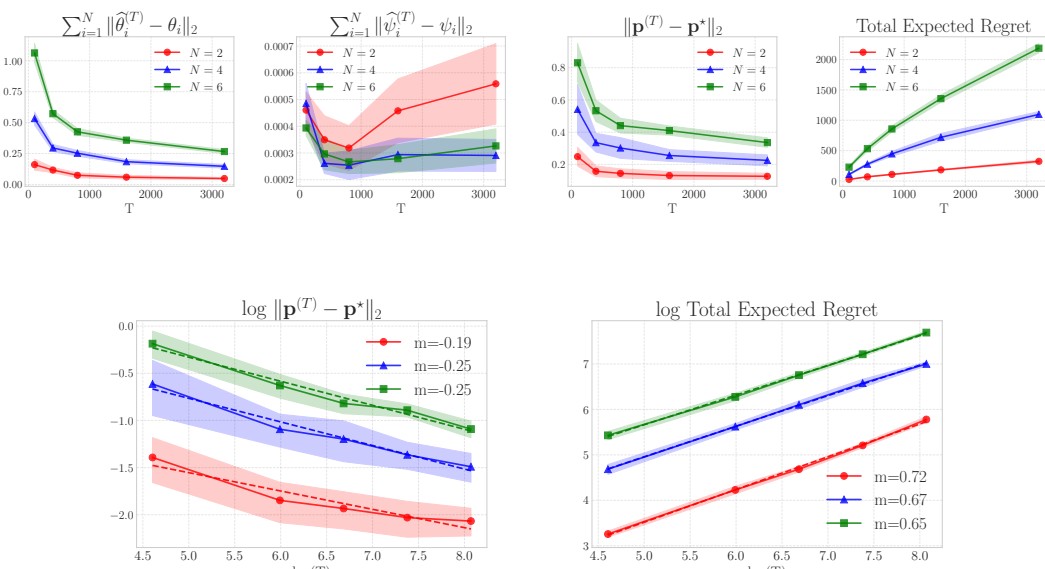

Figure 7: Performance in the different exploration phases.

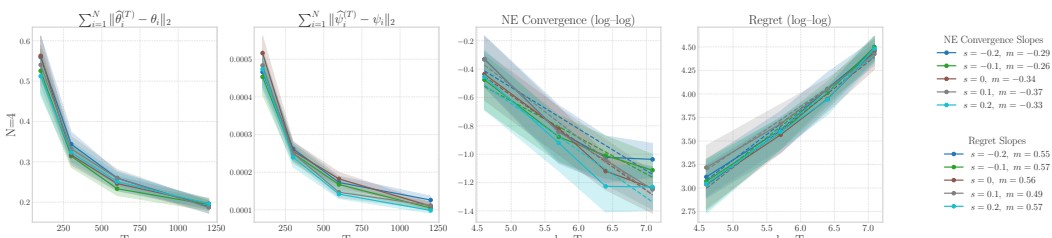

Figure 8: Log-log performance of Algorithm 1 in sequential price competition with $N = 4$ sellers for different values of misspecification of $s_i = 0$, specifically $\{-0.2, -0.1, 0, 0.1, 0.2\}$. The slopes, indicated as $m$, of the convergence to NE and the regret are close to each other. The small variations in the rates can be attributed to the finite sample experiment ($T = 1600$).

# K   ADDITIONAL FIGURES

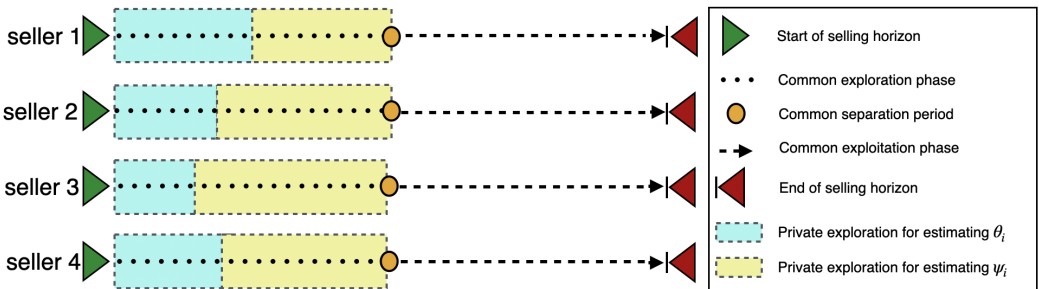

Figure 9: Illustration of our policy (Algorithm 1) with $N = 4$ sellers in sequential price competition under nonlinear demands. For each seller $i \in \{1, 2, 3, 4\}$, in their exploration phase (dotted line) of length $\tau$, they offer randomized prices following their distribution $\mathscr{D}_i$. Within the exploration phase, each seller has a private phase for estimating $\boldsymbol{\theta}_i$ (blue box), with length $\tau \kappa_i$ and a private phase for estimating $\psi_i$ (yellow box), with length $\tau(1 - \kappa_i)$. The seller's price experiment ends at period $t = \tau$ (orange circle). Subsequently, in their exploitation phase (dashed line), seller $i$ offers prices using the estimators generated in the exploration phase.

## L  EXAMPLES OF $s$-CONCAVE NON-DECREASING FUNCTIONS

The notion of $s$-concave functions generalizes concavity ($s = 1$) and log-concavity (which holds for $s = 0$). The class of log-concave functions has been extensively studied: see Bobkov & Madiman (2011); Dümbgen & Rufibach (2009); Cule & Samworth (2010); Borzadaran & Borzadaran (2011); Bagnoli & Bergstrom (2006). Since in this paper we consider a positive monotone $s$-concave function $\psi$, we first give some examples of monotone log-concave functions and generalize to $s$-concavity. Without loss of generality, we consider log-concave (and $s$-concave) CDFs $\psi$; indeed, if $\psi$ is monotone but its range is not contained in $[0, 1]$, it is always possible to rescale $\psi$, transforming it to a CDF: in fact, if $\psi$ is $s$-concave, then $c\psi$ is $s$-concave for every $c > 0$ because $\psi\left((1 - \lambda)u_0 + \lambda u_1\right) \geq M_s\left(\psi\left(u_0\right), \psi\left(u_1\right); \lambda\right)$ for all $u_0, u_1 \in \mathcal{U}$ if and only if $c \cdot \psi\left((1 - \lambda)u_0 + \lambda u_1\right) \geq M_s\left(c \cdot \psi\left(u_0\right), c \cdot \psi\left(u_1\right); \lambda\right)$ for all $u_0, u_1 \in \mathcal{U}$.

### L.1  LOG-CONCAVE CDF

In Bagnoli & Bergstrom (2006) we find a large class of log-concave CDF. They prove that if a density function $f$ is log-concave and continuously differentiable, then the CDF $F$ is also log-concave. This gives a way to generate log-concave CDFs. Figure 10, by Bagnoli & Bergstrom (2006), presents several widely used continuous univariate probability distributions whose densities are log-concave. Except for the Laplace distribution, the log-concavity of each density $f$ can be verified by checking that the second derivative of $\ln f(x)$ is non-positive across its support. For some families-such as the Weibull, power-function, beta, and gamma distributions-log-concavity holds only for specific parameter values. The table reports the parameter ranges under which these densities remain log-concave.

| Name of distribution | Support | Density function f(x) | Cumulative dist function F(x) | $(\ln f(x))''$ |
|---|---|---|---|---|
| Uniform | $[0, 1]$ | $1$ | x | $0$ |
| Normal | $(-\infty, \infty)$ | $\frac{1}{\sqrt{2\pi}}e^{-x^2/2}$ | $*$ | $-1$ |
| Exponential | $(0, \infty)$ | $\lambda e^{-\lambda x}$ | $1 - e^{-\lambda x}$ | $0$ |
| Logistic | $(-\infty, \infty)$ | $\frac{e^{-x}}{(1+e^{-x})^2}$ | $\frac{1}{(1+e^{-x})^2}$ | $-2f(x)$ |
| Extreme Value | $(-\infty, \infty)$ | $e^{-x}\exp\{-e^{-x}\}$ | $\exp\{-e^{-x}\}$ | $-e^{-x}$ |
| Laplace (Double Exponential) | $(-\infty, \infty)$ | $\frac{1}{2}e^{-|x|}$ | $e^{\lambda x}$ if $x \leq 0$ 
 $1 - \frac{1}{2}e^{-x}$ if $x \geq 0$ | $0$ for $x \neq 0$ |
| Power Function ($c \geq 1$) | $(0, 1]$ | $cx^{c-1}$ | $x^c$ | $\frac{1-c}{x^2}$ |
| Weibull ($c \geq 1$) | $[0, \infty)$ | $cx^{c-1}e^{-x^c}$ | $1 - e^{-x^c}$ | $\frac{1-c}{x^2}(1 + cx^c)$ |
| Gamma ($c \geq 1$) | $[0, \infty)$ | $\frac{x^{c-1}e^{-x}}{\Gamma(c)}$ | $*$ | $\frac{1-c}{x^2}$ |
| Chi-Squared ($c \geq 2$) | $[0, \infty)$ | $\frac{x^{(c-2)/2}e^{-x/2}}{2^{c/2}\Gamma(c/2)}$ | $*$ | $\frac{2-c}{2x^2}$ |
| Chi ($c \geq 1$) | $[0, \infty)$ | $\frac{x^{c-1}e^{-x^2/2}}{2^{(c-2)/2}\Gamma(c)}$ | $*$ | $\frac{1-c}{x^2} - 1$ |
| Beta ($\nu \geq 1$, $\omega \geq 1$) | $[0, 1]$ | $\frac{x^{\nu-1}(1-x)^{\omega-1}}{B(\nu,\omega)}$ | $*$ | $\frac{1-\nu}{x^2} + \frac{1-\omega}{(1-x)^2}$ |
| Maxwell | | This is a Chi distribution with $c = 3$ | | |
| Rayleigh | | This is a Weibull distribution with $c = 2$ | | |

Figure 10: (by Bagnoli & Bergstrom (2006)) Distributions with log-concave density functions (distribution functions marked $*$ lack a closed-form representation).

## L.2 Non log-concave CDF

When the density $f$ is not log-concave, analyzing the behavior of the associated cumulative distribution function $F$ is more complicated. Distributions with log-convex densities exhibit a wide range of behaviors: some have log-concave CDFs, others have log-convex CDFs, and still others have CDFs that are neither log-concave nor log-convex.

Figure 11, by Bagnoli & Bergstrom (2006) summarizes, for each of these cases, whether the density and the CDF are log-concave or log-convex.

| Name of distribution | Density function | c.d.f |
| --- | --- | --- |
| Power function $(0 < c < 1)$ | log-convex | log-concave |
| Weibull $(0 < c < 1)$ | log-convex | log-concave |
| Gamma $(0 < c < 1)$ | log-convex | log-concave |
| Arc-Sine | log-convex | neither |
| Pareto | log-convex | log-concave |
| Lognormal | neither | log-concave |
| Student's t | neither | neither |
| Cauchy | neither | neither |

Figure 11: (by Bagnoli & Bergstrom (2006)) Properties of distributions without log-concave density.

From Figure 11, we note that the CDFs of the *Pareto, Lognormal. Student's t, Cauchy* are not log-concave. In the next section, we prove that they are $s^*$-optimal-concave for some $s^* < 0$.

## L.3 $s$-concave CDF that are not log-concave

In this section, we present some properties of $s$-concave functions and give some examples. In this Section, we will widely use Lemma B.1 without explicit mention, which states that if $f$ is a positive function defined in an interval $(a, b)$ that is twice continuously differentiable, then $f$ is $s$-concave iff

$$f \cdot f'' + (s - 1)(f')^2 \leq 0 \quad \text{in } (a, b).$$

We start with the inclusion property.

**Proposition L.1.** *[Inclusion Property] Let $f : (a, b) \to (0, \infty)$, a twice differentiable function. If $f$ is $\beta$-concave, then $f$ is $s$-concave for $s \leq \beta$.*

*Proof.* If $f$ is $\beta$-concave, for $s \leq \beta$ it holds

$$f(u)f''(u) + (s - 1)(f'(u))^2 \leq f(u)f''(u) + (\beta - 1)(f'(u))^2 \leq 0,$$

i.e., $f$ is $s$-concave. $\qquad\square$

As a consequence, log-concavity implies $s$-concavity for every $s \leq 0$. An immediate consequence of the inclusion property is the following.

**Corollary L.2.** *Let $f : (a, b) \to (0, \infty)$, be a twice differentiable function. Suppose that $f$ is not $s_1$-concave, but is $s_0$-concave for some $s_0 < s_1$. Then, there exists a $s^\star \in [s_0, s_1)$ such that $f$ is $s$-concave for every $s \leq s^\star$ and is not $s$-concave for every $s > s^\star$.*

*Proof.* Let $s^\star = \sup\{s : f \cdot f'' + (s - 1)(f')^2 \leq 0\}$. By the inclusion property, $f$ is $s$-concave for all $s \leq s^\star$. However, there not exists a $s > s^\star$ such that $f$ is $s$-concave, because it contradicts the definition of $s^\star$. $\qquad\square$

**Definition L.3.** *From previous Corollary L.2, we say that a twice differentiable function $f : (a, b) \to (0, \infty)$ is $s^*$-optimal-concave if*

$$s^\star = \sup \left\{ s : \sup_{u \in (a,b)} [f(u)f''(u) + (s - 1)(f'(u))^2] \leq 0 \right\}.$$

*In this case, $s^\star$ is called optimal-concave parameter.*

In the following, we prove under which condition $s$-concavity is closed under multiplication.

**Proposition L.4.** *Let $f, g : (a, b) \to (0, \infty)$ be twice differentiable functions*

- *if $f$ and $g$ are $s$-concave with $s < 0$, such that $f' \cdot g' \geq 0$, then the product $f \cdot g$ is $s$-concave.*

- *if $f$ and $g$ are $0$-concave then the product $f \cdot g$ is $0$-concave.*

*Proof.* The second statement is the well-known result that the product of log-concave functions is log-concave; then, we only prove the first statement, which immediately follows from the equality

$$h \cdot h'' + (s-1)(h')^2 = g^2 \cdot (f \cdot f'' + (s-1)(f')^2) + f^2 \cdot (g \cdot g'' + (s-1)(g')^2) + 2sf \cdot g \cdot f' \cdot g'.$$

$\square$

In the next sections, we first show the existence of some well-known distributions whose density is $s$-concave for some $s < 0$ and not log-concave (i.e., $0$-concave). Then we identify a class of distributions for which if the density $f$ is $s$-concave, then the CDF $F$ is $\mu(s)$-concave for a fixed transformation $\mu$ that will be specified later.

### L.3.1 $s$-CONCAVE CONTINUOUS DENSITIES

From now on, we assume that the PDFs that we consider are continuous. As we showed in Appendix L.2, many density functions are *not log-concave*, such as Student's t, Cauchy, Pareto, and log-Normal. However, for all of them, we can show that the PDFs are $s$-concave for some $s < 0$.

**Remark L.5.** *Note that for PDF $f$ such that $\{u : f(u) > 0\} = \mathbb{R}$, it is necessary to impose $s \leq 0$. Indeed for $s > 0$ the function $\phi(u) = d_s \circ f(u) = f^s(u)$, which is strictly positive on $\mathbb{R}$, can not be concave. To see this note that, being $f$ a density function, necessarily $f(u) \to 0^+$ as $|u| \to +\infty$, and consequently, as $|u| \to +\infty$ we have $\phi(u) \to 0$ which is not possible if $\phi$ is concave and strictly positive on $\mathbb{R}$. The same applies to densities defined on any unbounded interval. This implies that if a density defined in an unbounded interval is log-concave (that is $0$-concave), then it is $0$-optimal-concave. However, when a density $f$ is compactly supported, there is no restriction on $s$.*

The proof of Proposition L.6 is deferred to Appendix L.3.5.

**Proposition L.6.** *1. The Student's t-distribution PDF is $\left(-\frac{1}{\nu+1}\right)$-optimal-concave where $\nu > 0$ is the degree of freedom.*

2. *The Cauchy PDF is $(-1/2)$-optimal-concave (independently of scaling and location parameters).*

3. *The Pareto PDF is $\left(-\frac{1}{\alpha+1}\right)$-optimal-concave, (independently of the location), where $\alpha > 0$ is the scaling factor.*

4. *The log-normal PDF with parameters $(\mu, \sigma^2)$ is $\left(-\frac{\sigma^2}{4}\right)$-optimal-concave, (independently of $\mu$).*

### L.3.2 $s$-CONCAVE CDFs AND SURVIVAL FUNCTIONS

Let $F$ be a CDF and $\bar{F} = 1 - F$ its survival function, and let $f = F'$. When $\{u : f(u) > 0\} = \mathbb{R}$, for similar reasoning as in Remark L.5, a necessary condition for having $\phi = d_s \circ F$ or $\phi = d_s \circ \bar{F}$ concave, is that $s \leq 0$. In the following proposition, we find a necessary condition on $s$ for which $F$ and $\bar{F}$ are $s$-concave when $\{u : f(u) > 0\} = (a, b)$ for $a, b \in \mathbb{R}$.

The proof of Proposition L.7 is deferred to Appendix L.3.6.

**Proposition L.7.** *Let $f : (a, b) \mapsto (0, \infty)$ be twice continuously differentiable function, and let $F(u) = \int_a^u f(t)dt$ for all $x \in (a, b)$ and define $f^{(k)}(a) = \lim_{u \to a} f^{(k)}(u)$ and $f^{(k)}(b) = \lim_{u \to b} f^{(k)}(u)$ for $k = 0, 1$.*

1) *If $f(a) = 0$, $f'(a) = 0$ and $F$ is $s$-concave, then $s \leq 1/2$.*

*2) If $f(b) = 0$, $f'(b) = 0$ and $\bar{F}$ is s-concave, then $s \leq 1/2$.*

In the following proposition, we prove that, under certain conditions, when a density function is $s$-concave, then the CDF and the survival function is $\mu$-concave for some $\mu = \mu(s)$. The proof of Proposition L.8 is deferred to Appendix L.3.7.

**Proposition L.8.** *Fix a function $f : (a,b) \mapsto (0,\infty)$ continuously differentiable, and let $F(u) = \int_a^u f(t)dt$ for all $x \in (a,b)$ and define $f(a) = \lim_{u \to a} f(u)$. Then:*

1) *If $f$ is s-concave on $(a,b)$ with $f(a) \neq 0$ and $s > -1$, then $F$ is $\mu$-concave for all $\mu \leq 1 - \frac{1}{s+1}$.*

2) *If $f$ is s-concave on $(a,b)$ with $f(a) = 0$ and $s \neq -1$, then $F$ is $\mu$-concave for all $\mu \leq 1 - \frac{1}{s+1}$.*

3) *If $f$ is monotone decreasing, then $F$ is s-concave for any $s < 1$.*

The proof of Proposition L.9 is deferred to Appendix L.3.8.

**Proposition L.9.** *Fix a function $f : (a,b) \mapsto [0,\infty)$ continuously differentiable, and let $F(u) = \int_a^u f(t)dt$ for all $x \in (a,b)$ and define $f(b) = \lim_{u \to b} f(u)$. Then:*

1) *If $f$ is s-concave on $(a,b)$ with $f(b) \neq 0$ and $s > -1$, then $\bar{F}$ is $\mu$-concave for all $\mu \leq 1 - \frac{1}{s+1}$.*

2) *If $f$ is s-concave on $(a,b)$ with $f(b) = 0$ and $s \neq -1$, then $\bar{F}$ is $\mu$-concave for all $\mu \leq 1 - \frac{1}{s+1}$.*

3) *If $f$ is monotone decreasing, then $\bar{F}$ is s-concave for any $s < 1$.*

The special case $s = 0$ was proved by Bagnoli & Bergstrom (2006).

### L.3.3    $s$-CONCAVE CDFS THAT ARE NOT LOG-CONCAVE

From Figure 11 we already know that the CDFs of the *Pareto, Lognormal. Student's t, Cauchy* are not log-concave. However, by our Proposition L.9 we immediately get that they are $s^*$-optimal-concave for some $s^* < 0$.

**Corollary L.10.**         *1. The Student's t-distribution CDF and survival function are $\mu$-concave for any $\mu \leq -\frac{1}{\nu}$ where $\nu > 0$ is the degree of freedom.*

2. *The Cauchy CDF and survival function are $\mu$-concave for any $\mu \leq 1$ (independently of scaling and location parameters).*

3. *The Pareto CDF and survival function are $\mu$-concave for any $\mu \leq -\frac{1}{\alpha}$, (independently of the location), where $\alpha$ is the scaling factor.*

4. *The log-normal CDF and survival function are with parameters $(\mu, \sigma^2)$ are $\mu$-concave for any $\mu \leq \frac{\sigma^2}{\sigma^2 - 4}$, (independently of $\mu$).*

From Proposition L.8 (and Proposition L.9), if $f$ is $\mu^\star$-optimal-concave it does not necessarily means that $F$ (and $\bar{F}$) is $s(\mu^*) = \left(1 - \frac{1}{\mu^*+1}\right)$-optimal concave. Indeed, the optimal concave value can be larger than $s(\mu^*)$. However, for the class of distributions in Corollary L.10, we know that $s^*$ has to be $< 0$.

In the next section, we construct a class of positive $s$-concave monotone functions that are not log-concave.

### L.3.4    GENERATE $s^*$-OPTIMAL-CONCAVE NON-DECREASING FUNCTIONS IN $[0,1]$, FOR $s^* < 0$.

**Proposition L.11.** *Let $\vartheta : (0,1) \to \mathbb{R}$ be a twice differentiable non-decreasing function (i.e., $\vartheta' \geq 0$) and strictly convex on a set of positive measure (i.e., $\vartheta'' > 0$ on a set of positive measure). Assume that there exists $s^* < 0$ such that*

$$s^* = \sup_s \{s : \vartheta''(x) + s(\vartheta'(x))^2 \leq 0 \quad \text{for all } x \in (0,1)\}.$$

*Define $F(x) = e^{\vartheta(x)}$. Then $F$ is s\*-concave, but $F$ is not log-concave (i.e., $F''(x)F(x) - (F'(x))^2 > 0$ for at least one $x \in (0,1)$).*

*Proof.* We compute

$$F'(x) = F(x)\vartheta'(x), \qquad F''(x) = F(x)(\vartheta'(x))^2 + F(x)\vartheta''(x).$$

Hence

$$\begin{aligned} F(x)F''(x) + (s-1)(F'(x))^2 &= F^2(x)\big[(\vartheta')^2 + \vartheta''\big] + (s-1)F^2(x)(\vartheta')^2 \\ &= F^2(x)\big[\vartheta'' + s(\vartheta')^2\big]. \end{aligned}$$

Since $F^2(x) > 0$, the inequality

$$FF'' + (s-1)(F')^2 \le 0$$

holds if and only if

$$\vartheta''(x) + s(\vartheta'(x))^2 \le 0.$$

Thus the assumed condition shows that $F$ is s\*-concave. On the other hand, log–concavity corresponds to the case $s = 0$, i.e.

$$\vartheta''(x) + 0 \cdot (\vartheta')^2 \le 0 \quad \Longleftrightarrow \quad \vartheta''(x) \le 0.$$

Since $\vartheta$ is strictly convex on a set of positive measure, we have $\vartheta''(x) > 0$ for the $x$ in this set, so this condition fails. Hence $F$ is *not* log-concave. $\square$

**Examples of admissible $\vartheta$.** We provide three explicit choices of strictly convex functions $\vartheta : (0,1) \to \mathbb{R}$ satisfying

$$\vartheta''(x) + s^*(\vartheta'(x))^2 \le 0 \quad \text{for some } s^* < 0,$$

so that $F(x) = e^{\vartheta(x)}$ is s\*-concave but not log-concave.

**1. Power function example.** Fix $\alpha > 1$ and set $\vartheta(x) = x^\alpha$. Then

$$\vartheta'(x) = \alpha x^{\alpha-1}, \qquad \vartheta''(x) = \alpha(\alpha-1)x^{\alpha-2},$$

and

$$\vartheta''(x) + s(\vartheta'(x))^2 = \alpha x^{\alpha-2}\Big[(\alpha-1) + s\alpha x^\alpha\Big].$$

Since $x^\alpha \le 1$ on $(0,1)$, a sufficient (and sharp) condition is $s \le -\frac{\alpha-1}{\alpha}$. Hence $F(x) = \exp(x^\alpha)$ is s\*-concave for $s^* = -\frac{\alpha-1}{\alpha}$, but not log-concave because $\vartheta''(x) > 0$.

**2. Exponential example.** Fix $k > 0$ and set $\vartheta(x) = e^{kx}$. Then

$$\vartheta'(x) = ke^{kx}, \qquad \vartheta''(x) = k^2 e^{kx},$$

and

$$\vartheta''(x) + s(\vartheta'(x))^2 = k^2 e^{kx}\big(1 + se^{kx}\big).$$

Since $e^{kx} \le e^k$ on $(0,1)$, a sufficient and sharp condition is $s \le -e^{-k}$. Thus $F(x) = \exp(e^{kx})$ is s\*-concave for $s^* = -e^{-k}$, but not log-concave since $\vartheta''(x) > 0$.

### L.3.5   PROOF OF PROPOSITION L.6

*Proof of a).* We have that $f(x) = \frac{\Gamma\left(\frac{\nu+1}{2}\right)}{\sqrt{\pi\nu}\Gamma\left(\frac{\nu}{2}\right)}\left(1 + \frac{x^2}{\nu}\right)^{-\frac{\nu+1}{2}}$ where $\nu > 0$ is the degree of freedom.

We need to find the values of $\nu$ such that $f \cdot f'' + (s-1)(f')^2 \le 0$ which reduces to

$$\left(1 + \frac{x^2}{\nu}\right)^{-\frac{\nu+1}{2}}\left[\frac{4x^2}{\nu^2}\left(-\frac{\nu+1}{2}\right)\left(-\frac{\nu+1}{2}-1\right)\left(1+\frac{x^2}{\nu}\right)^{-\frac{\nu+1}{2}-2} + \frac{2}{\nu}\left(-\frac{\nu+1}{2}\right)\left(1+\frac{x^2}{\nu}\right)^{-\frac{\nu+1}{2}-1}\right]$$

$$+ (s-1)\left[\frac{2x}{\nu}\left(-\frac{\nu+1}{2}\right)\left(1+\frac{x^2}{\nu}\right)^{-\frac{\nu+1}{2}-1}\right]^2 \le 0$$

iff

$$\frac{4x^2}{\nu^2}\left(-\frac{\nu+1}{2}\right)\left(-\frac{\nu+1}{2}-1\right)\left(1+\frac{x^2}{\nu}\right)^{-\frac{\nu+1}{2}-2} + \frac{2}{\nu}\left(-\frac{\nu+1}{2}\right)\left(1+\frac{x^2}{\nu}\right)^{-\frac{\nu+1}{2}-1}$$

$$+ (s-1)\frac{4x^2}{\nu^2}\left(-\frac{\nu+1}{2}\right)^2\left(1+\frac{x^2}{\nu}\right)^{-\frac{\nu+1}{2}-2} \le 0$$

iff

$$\frac{4x^2}{\nu^2}\left(-\frac{\nu+1}{2}\right)\left(-\frac{\nu+1}{2}-1\right) + \frac{2}{\nu}\left(-\frac{\nu+1}{2}\right)\left(1+\frac{x^2}{\nu}\right) + (s-1)\frac{4x^2}{\nu^2}\left(-\frac{\nu+1}{2}\right)^2 \le 0$$

iff

$$\frac{4x^2}{\nu^2}\left(-\frac{\nu+1}{2}-1\right) + \frac{2}{\nu}\left(1+\frac{x^2}{\nu}\right) + (s-1)\frac{4x^2}{\nu^2}\left(-\frac{\nu+1}{2}\right) \ge 0$$

iff

$$4x^2\left(-\frac{\nu+1}{2}-1\right) + 2\nu\left(1+\frac{x^2}{\nu}\right) + (s-1)4x^2\left(-\frac{\nu+1}{2}\right) \ge 0$$

iff

$$x^2\left(-2(\nu+3) - (s-1)2(\nu+1) + 2\nu\right) + 2\nu \ge 0.$$

As long as the coefficient of $x^2$ is positive the inequality is true for all $x$, i.e.

$$-(\nu+3) - (s-1)(\nu+1) + 1 \ge 0 \quad \Rightarrow \quad (s-1)(\nu+1) \le -\nu - 2 \quad \Rightarrow \quad s \le -\frac{1}{\nu+1}.$$

$\square$

*Proof of b).* The Cauchy has density $f(x) = (\pi\gamma)^{-1}\left[1+\left(\frac{x-x_0}{\gamma}\right)^2\right]^{-1}$ where $x_0 \in \mathbb{R}$ is the location and $\gamma > 0$ the scale parameter. We need to find the values of $\nu$ such that $f \cdot f'' + (s-1)(f')^2 \le 0$. However, since this inequality has to hold for all $x$, we can assume $x_0 = 0$. The characterization translates to

$$\left(1+\frac{x^2}{\gamma^2}\right)^{-1}\left[(-1)\frac{2}{\gamma^2}\left[1+\frac{x^2}{\gamma^2}\right]^{-2} + \frac{8x^2}{\gamma^4}\left[1+\frac{x^2}{\gamma^2}\right]^{-3}\right] + (s-1)\left((-1)\frac{2x}{\gamma^2}\left[1+\frac{x^2}{\gamma^2}\right]^{-2}\right)^2 \le 0$$

iff

$$(-1)\frac{2}{\gamma^2}\left[1+\frac{x^2}{\gamma^2}\right]^{-2} + \frac{8x^2}{\gamma^4}\left[1+\frac{x^2}{\gamma^2}\right]^{-3} + (s-1)\frac{4x^2}{\gamma^4}\left[1+\frac{x^2}{\gamma^2}\right]^{-3} \le 0$$

iff

$$(-1)\frac{2}{\gamma^2}\left[1+\frac{x^2}{\gamma^2}\right] + \frac{8x^2}{\gamma^4} + (s-1)\frac{4x^2}{\gamma^4} \le 0$$

iff

$$-\frac{2x^2}{\gamma^4} - \frac{2}{\gamma^2} + \frac{8x^2}{\gamma^4} + (s-1)\frac{4x^2}{\gamma^4} \le 0$$

iff

$$x^2\left(-\frac{2}{\gamma^4} + \frac{8}{\gamma^4} + (s-1)\frac{4}{\gamma^4}\right) - \frac{2}{\gamma^2} \le 0$$

the inequality is true for all $x$ as long as

$$6 + (s-1)4 \le 0 \Rightarrow s \le -\frac{1}{2}.$$

$\square$

*Proof of c).* $f(x) = \frac{\alpha x_m^\alpha}{x^{\alpha+1}}$, where $x_m > 0$ is the scale parameter and $\alpha > 0$ the shape. We need to find the values of $\nu$ such that $f \cdot f'' + (s-1)(f')^2 \leq 0$ which reduces to

$$x^{-(\alpha+1)}(-1)(\alpha+1)(-(\alpha+1)-1)x^{-(\alpha+1)-2} + (s-1)(\alpha+1)^2 x^{-2(\alpha+1)-2} \leq 0$$

iff

$$(-1)(\alpha+1)(-(\alpha+1)-1) + (s-1)(\alpha+1)^2 \leq 0$$

iff

$$(\alpha+1)(\alpha+2) + (s-1)(\alpha+1)^2 \leq 0 \Rightarrow s \leq -\frac{1}{\alpha+1}.$$

$\square$

*Proof of d).* $f(x) = \frac{1}{x\sigma\sqrt{2\pi}} \exp\left(-\frac{(\ln x - \mu)^2}{2\sigma^2}\right)$, where $x, \sigma > 0$ and $\mu \in \mathbb{R}$. Since $\frac{1}{\sigma\sqrt{2\pi}} > 0$ the problem reduces to prove that $f(x) = \exp\left(-\frac{(\ln x - \mu)^2}{2\sigma^2}\right)/x$ is $s$-concave. Note that

$$\begin{aligned}
f'(x) &= -\frac{1}{x^2}\exp\left(-\frac{(\ln x - \mu)^2}{2\sigma^2}\right) + \frac{1}{x}\exp\left(-\frac{(\ln x - \mu)^2}{2\sigma^2}\right)\left[-\frac{\ln x - \mu}{x\sigma^2}\right] \\
&= -\frac{1}{x^2}\exp\left(-\frac{(\ln x - \mu)^2}{2\sigma^2}\right)\left[1 + \frac{\ln x - \mu}{\sigma^2}\right] \\
&= -\frac{1}{\sigma^2 x^2}\exp\left(-\frac{u^2}{2\sigma^2}\right)(\sigma^2 + u)
\end{aligned}$$

where $u = \ln x - \mu$, and

$$\begin{aligned}
f''(x) &= -\frac{1}{x^2}\exp\left(-\frac{(\ln x - \mu)^2}{2\sigma^2}\right)\left[1 + \frac{\ln x - \mu}{\sigma^2}\right]\left[-\frac{\ln x - \mu}{x\sigma^2}\right] + \\
&\quad -\exp\left(-\frac{(\ln x - \mu)^2}{2\sigma^2}\right)\frac{\frac{1}{x\sigma^2}x^2 - 2x\left[1 + \frac{\ln x - \mu}{\sigma^2}\right]}{x^4} \\
&= \frac{1}{x^3}\exp\left(-\frac{(\ln x - \mu)^2}{2\sigma^2}\right)\left[1 + \frac{\ln x - \mu}{\sigma^2}\right]\frac{\ln x - \mu}{\sigma^2} + \\
&\quad -\exp\left(-\frac{(\ln x - \mu)^2}{2\sigma^2}\right)\frac{\frac{1}{\sigma^2} - 2\left[1 + \frac{\ln x - \mu}{\sigma^2}\right]}{x^3} \\
&= \frac{1}{x^3}\exp\left(-\frac{u^2}{2\sigma^2}\right)\left[1 + \frac{u}{\sigma^2}\right]\frac{u}{\sigma^2} - \exp\left(-\frac{u^2}{2\sigma^2}\right)\frac{\frac{1}{\sigma^2} - 2\left[1 + \frac{u}{\sigma^2}\right]}{x^3} \\
&= \frac{1}{\sigma^4 x^3}\exp\left(-\frac{u^2}{2\sigma^2}\right)\left[(\sigma^2 + u)u - \sigma^2 + 2\sigma^4 + 2\sigma^2 u\right] \\
&= \frac{1}{\sigma^4 x^3}\exp\left(-\frac{u^2}{2\sigma^2}\right)\left[u^2 + 3\sigma^2 u - \sigma^2(1 - 2\sigma^2)\right].
\end{aligned}$$

We need $f \cdot f'' + (s-1)(f')^2 \leq 0$, that is

$$\frac{1}{\sigma^4 x^4}\exp\left(-\frac{u^2}{\sigma^2}\right)\left[u^2 + 3\sigma^2 u - \sigma^2(1 - 2\sigma^2)\right] + (s-1)\frac{1}{\sigma^4 x^4}\exp\left(-\frac{u^2}{\sigma^2}\right)(\sigma^2 + u)^2 \leq 0$$

iff

$$u^2 + 3\sigma^2 u - \sigma^2(1 - 2\sigma^2) + (s-1)(\sigma^2 + u)^2 \leq 0$$

iff

$$su^2 + (1 + 2s)\sigma^2 u - \sigma^2(1 - 2\sigma^2) + (s - 1)\sigma^4 \leq 0$$

iff

$$su^2 + (1 + 2s)\sigma^2 u + s\sigma^4 + \sigma^4 - \sigma^2 \leq 0.$$

The determinant $\Delta = (1 + 2s)^2\sigma^4 - 4s(s\sigma^4 + \sigma^4 - \sigma^2) = \sigma^2(\sigma^2 + 4s)$. Then we have $s$-concavity for all $s \leq -\frac{\sigma^2}{4}$.

$\square$

### L.3.6  PROOF OF PROPOSITION L.7

*Proof.* We have that $F \cdot f' + (s - 1)f^2 \leq 0$, which implies $f'(u) \leq (1 - s)\frac{f^2(u)}{F(u)}$ for all $u \in (a, b)$. By sign conservation theorem this implies that $f'(a) \leq (1 - s)\lim_{u \to a}\frac{f^2(u)}{F(u)} = (1 - s)\lim_{u \to a}\frac{2f(u)f'(u)}{f(u)} = (1 - s)\lim_{u \to a}2f'(u) = (1 - s)2f'(a)$. Since $f'(a) \neq 0$ then $f'(a) > 0$ and then $1 \leq (1 - s)2$ which implies $s \leq \frac{1}{2}$. Similar proof hold for $\bar{F}$.

$\square$

### L.3.7  PROOF OF PROPOSITION L.8

*Proof.* We need to prove that $F \cdot f' + (\mu - 1)f^2 \leq 0$. Using that $f^{s-1} \cdot f'$ is non-increasing (because $(d_s \circ f)'' \leq 0$ i.e. $(d_s \circ f)' = f^{s-1} \cdot f$ is non-increasing) we have

$$\frac{f'(u)}{f(u)}F(u) = \frac{f^{s-1}(u)}{f(u)}\frac{f'(u)}{f^{s-1}(u)}\int_a^u f(t)dt \leq \frac{1}{f(u)f^{s-1}(u)}\int_a^u f^{s-1}(t)f'(t)f(t)dt$$

$$= \frac{1}{f(u)f^{s-1}(u)}\left[\frac{f^{s+1}(u) - f^{s+1}(a)}{s + 1}\right]$$

If $f(a) \neq 0$ then, being $s > -1$ we have

$$\frac{f'(u)}{f(u)}F(u) \leq \frac{1}{f(u)f^{s-1}(u)}\left[\frac{f^{s+1}(u) - f^{s+1}(a)}{s + 1}\right] \leq \frac{f(u)}{s + 1}$$

which implies $0 \geq F \cdot f' - \frac{1}{s+1}f^2 \geq F \cdot f' + (\mu - 1)f^2$, which proves 1). For 2), being $f(a) = 0$, note that $\frac{f'(u)}{f(u)}F(u) \leq \frac{f(u)}{s+1}$ holds with the only assumption that $s \neq -1$. For point 3), notice that since $F^{s-1}$ is monotone decreasing for $s < 1$, then $F^{s-1} \cdot f'$ is monotone decreasing, but $F^{s-1}(u)f'(u) = \frac{d}{du}(d_s(F(u)))$, which implies that $F$ is $s$-concave. $\square$

### L.3.8  PROOF OF PROPOSITION L.9

*Proof.* We need need to prove that $-\bar{F}f' + (\mu - 1)f^2 \leq 0$. Using that $f^{s-1} \cdot f'$ is non-increasing we have

$$\frac{f'(u)}{f(u)}\bar{F}(u) = \frac{f^{s-1}(u)}{f(u)}\frac{f'(u)}{f^{s-1}(u)}\int_u^b f(t)dt \geq \frac{1}{f(u)f^{s-1}(u)}\int_u^b f^{s-1}(t)f'(t)f(t)dt$$

$$= \frac{1}{f(u)f^{s-1}(u)}\left[\frac{f^{s+1}(b) - f^{s+1}(u)}{s + 1}\right]$$

If $f(b) \neq 0$ the, since $s > -1$, we have

$$\frac{f'(u)}{f(u)}\bar{F}(u) \geq \frac{1}{f(u)f^{s-1}(u)}\left[\frac{f^{s+1}(b) - f^{s+1}(u)}{s + 1}\right] \geq -\frac{f(u)}{s + 1}$$

which implies $0 \geq -\bar{F}f' - \frac{1}{s+1}f^2 \geq -\bar{F}f' + (\mu - 1)f^2$, which proves 1). For 2), being $f(b) = 0$, we have that $\frac{f'(u)}{f(u)}\bar{F}(u) \geq -\frac{f(u)}{s+1}$ holds with the only assumption that $s \neq -1$. For point 3), notice that since $-\bar{F}^{s-1}$ is monotone decreasing for $s < 1$, then $-\bar{F}^{s-1} \cdot f'$ is monotone decreasing, but $-\bar{F}^{s-1}(u)f'(u) = \frac{d}{du}(d_s(\bar{F}(u)))$, which implies that $F$ is $s$-concave for any $s < 1$. $\square$

