# OpenReview forum: "Revenue Maximization Under Sequential Price Competition Via The Estimation Of $s$-Concave Demand Functions"
_ICLR.cc/2026/Conference — ICLR 2026 Poster_

### Official Review · Reviewer_apYP · 2025-10-14

**Soundness:** 3
**Presentation:** 3
**Contribution:** 2
**Rating:** 4
**Confidence:** 3

**Summary:**

The paper studies a repeated price competition game with N sellers over T periods. In each period, sellers post prices simultaneously and publicly observe the joint price vector, but each seller only observes its own random demand. Seller i’s mean demand is modeled as a monotone single-index function $,\lambda_i(p)=\psi_i(\langle \theta_i, p\rangle),$ with $\theta_i=(-\beta_i,\gamma_i)$, $\beta_i>0$, and $\psi_i$ unknown, increasing, and $s_i$-concave. The domain of the index is $U=[-p_{\max},p_{\max}]$, and noises are sub-Gaussian. Each seller’s goal is to maximize revenue and minimize a regret metric defined against a dynamic benchmark best-response $\Gamma_i(p_{-i})=\arg\max_{p_i\in P_i} p_i \psi_i(-\beta_i p_i+\langle \gamma_i, p_{-i}\rangle)$ while treating rival prices as fixed. The policy uses a common exploration of length $\tau\propto T^\xi$ to estimate $(\theta_i,\psi_i)$ in two stages via centered least squares (for $\theta_i$ under elliptical designs) and shape-constrained nonparametric least squares (for $\psi_i$ with monotonicity and $s$-concavity), then an exploitation phase applying iterative best responses with the learned models. A key structural identity links virtual valuation curvature $\phi_i'(u)\ge c_i$ to $s$-concavity with $s_i=c_i-1$, enabling a tuning-free estimator and equilibrium analysis through a contraction of the best-response map.

In this paper, the authors prove (1) The existence of a unique Nash Equilibrium, characterized by $p_i^\star=\Pi_{P_i}, g_i(\langle \gamma_i,p_{-i}^\star\rangle)/\beta_i$ where $g_i(u)=u-\phi_i^{-1}(u)$, and the joint best-response operator $\Gamma$ is a contraction when $\sup_i |g_i'|_\infty |\gamma_i|_1/\beta_i<1$. (2) The concentration for parameters: $|\hat\theta_i-\theta_i|_2=O\big(\sqrt{(N\log n^{(1)}i)/n^{(1)}i}\big)$; sup-norm nonparametric error $\mathbb{E}\big[\sup{u\in K}|\hat\psi{i,\hat\theta_i}(u)-\psi_i(u)|\big]=O\big((\log n^{(2)}_i/n^{(2)}_i)^{2/5}\big)$. And (3) the regret rate $O\big(T^\xi+T^{1-2\xi/5} N^{3/2} (\log T)^{2/5}\big)$ and $\mathbb{E}|p^{(T)}-p^\star|_2^2=O\big(N^{3/2} T^{-2\xi/5}(\log T)^{2/5}\big)$, optimized at $\xi=5/7$ to yield $\tilde O(N^{3/2}T^{5/7})$ regret and $\tilde O(N^{3/4}T^{-1/7})$ equilibrium convergence.

The authors also conduct numerical experiments with $N=2,4,6$ and log-concave links show convergence to NE and sublinear regret aligned with the theoretical rates, and finally a robustness check. Compared to existing literature, this work generalizes linear-demand competition with $\tilde O(\sqrt{T})$ regret to semiparametric monotone SIM under shape constraints and matches the known $\tilde O(T^{5/7})$ rate in the monopolistic SIM case while introducing a tuning-free $s$-concave NPLS analysis and a new equivalence between $\phi_i'$ and $s$-concavity.

**Strengths:**

(1) Technical connection between $\phi_i'(u)\ge c_i$ and $s$-concavity ($s_i=c_i-1$), enabling both equilibrium characterization and estimation under shape constraints.

(2) Sup-norm concentration for $h\circ$concave regression with $O((\log n/n)^{2/5})$ rates; careful two-stage estimation to decouple dependencies.

(3) Broadens beyond linear or fixed parametric nonlinear demand; theory covers NE existence/uniqueness and convergence under strategic coupling.

(4) Clean regret/convergence bounds tied to estimation rates; simulations align with theory.

Overall, this is a good work.

**Weaknesses:**

There are some issues that I’m most concerned about:

(1) The assumed knowledge on $c_i$ or $s_i$, which is a very strong assumption.

(2) The optimality of $\tilde O(N^{3/2}T^{5/7})$ regret. A matching lower bound is not established.

(3) The condition of contraction $\sup_i |g_i'|_\infty |\gamma_i|_1/\beta_i<1$ is not well justified in real-world markets.

**Questions:**

(1) How should practitioners set $c_i$ (and $s_i=c_i-1$) if unknown? Is there a safe adaptive surrogate (under/overestimation) that preserves contraction and regret rates?

(2) Do you conjecture a minimax lower bound matching $T^{5/7}$ for monotone $s$-concave SIM in this feedback model? Where is the main slack otherwise?

(3) Please elaborate on the exploration distribution assumption (elliptical with $g(x+y)=g(x)g(y)$). Also, if exploration uses truncated Gaussians within $P$ (non-elliptical full support), how do concentration results and the decoupling arguments change?

Some typos to notice:
Line 239, “coponents”

---

> ### Author Response · Authors · 2025-11-23
> **Response to Reviewer apYP**
>
> Thank you for taking the time to review our paper and for the helpful comments and suggestions on our work. Please see our responses below.
>
> ## Q1) Considerations for the case of unknown $c_i$.
>
> Thanks for the insightful question. Please refer to Remark 5.7 in the updated version, where we discuss how the upper bound on regret and the convergence to the Nash equilibrium depend on $c_i$, or equivalently on $s_i=c_i-1$. We show that the algorithm is robust to a certain degree of misspecification of $s_i$: the contraction condition is preserved, and both the regret bound and the convergence rate to the NE remain unchanged. In Appendix J3, we also provide simulations with a fixed choice $s_i=0$, illustrating that the empirical regret and NE convergence rates remain essentially the same under modest misspecifications of $s_i$. Finally, in the conclusions of our manuscript, we outline a possible future direction: jointly estimating $s_i$ and $\psi_i$, which could lead to even stronger algorithmic guarantees. However, we note that this is a substantial statistical challenge for future work.
>
> ---
>
> ## Q2) Conjecturing the minimax lower bound.
>
> We conjecture that ***the uniform convergence rate in Theorem H. 5 for estimating $\psi_i$ is minimax-optimal for monotone $s$-concave regression***, so the statistical component of our analysis does not introduce slack. Indeed, the rate is determined entirely by the underlying concavity structure: $s$-concavity itself does not change the statistical difficulty, as it simply corresponds to applying a known monotone transformation that reduces the problem to concave regression. Under the smoothness Assumption H.4, the optimal uniform convergence rate for the convex regression problem is known to be $(\log(n)/n)^{2 / 5}$ (see e.g., [1], Eq. 1.2), which matches the rate we obtained.
>
> However, ***we do not conjecture that the overall *regret rate* $T^{5 / 7}$ is minimax-optimal*** for this feedback model. The main looseness comes from the two-phase design: explicit exploration forces the regret decomposition $T^{\xi}+ T^{1-2 \xi / 5}$, whose balancing yields $5 / 7$. We believe that algorithms avoiding explicit exploration-while still controlling the semiparametric estimation error-could potentially achieve faster theoretical regret.
>
> ## References
> [1] Wang, Xiao, and Jinglai Shen. "Uniform Convergence and Rate Adaptive Estimation of a Convex Function." arXiv preprint arXiv:1205.0023 (2012).

---

> > ### Author Response · Authors · 2025-11-23
> >
> > ## Q3) The choice of elliptically distributed prices during exploration.
> >
> > Thanks for the question.
> >
> > 1) **Discussion about Assumption 5.1**. In the revised version, we refer the reader to Remark D.1 and Remark D.2, which provide a detailed discussion of Assumption 5.1 and the choice of the price distribution during the exploration phase. Before describing the assumption here, we emphasize that selecting an appropriate distribution for the design points (here, the exploration prices) is standard in exploration–exploitation algorithms. This choice is crucial because it ensures that the learner can perform exploration design and obtain a consistent estimator. For example, both [1] and [2] employ uniformly randomized prices precisely to guarantee consistency of their estimators. We now describe the components of Assumption 5.1:
> >
> > * The requirement that exploration prices follow an elliptically symmetric distribution is used to guarantee the consistency of the estimator of $\boldsymbol{\theta}_i$ in the single-index model. In particular, this assumption is invoked in Lemma E.2 to derive the normal equations (the same assumption can be found in [3] and [4]). If the exploration distribution were not elliptically symmetric, the resulting estimator would fail to be consistent.
> >
> > * **The assumption g(x+y)=g(x)g(y)** (satisfied, for example, by Gaussian distributions) supports the estimation of $\psi_i$. It ensures that $\psi_{i,\boldsymbol{\theta}}(u)$ depends on $u$ solely through the argument of $\psi_i$ rather than that of $g$. This decoupling guarantees that $\psi_{i,\boldsymbol{\theta}}$ inherits key properties from $\psi_i$, such as smoothness and adherence to the prescribed shape constraints. In turn, these properties are crucial for establishing the consistency of the estimator of $\psi_i$. This argument appears in our Remark D.2.
> >
> > 2) **Regarding truncated Gaussians**. Truncation must occur along ellipsoidal level sets; otherwise, the resulting distribution is no longer elliptical, and the consistency guarantees for $\boldsymbol{\theta}_i$ no longer apply. Truncating a Gaussian to a rectangle, such as $P=\left[p_i, \bar{p}_i\right]^N$ destroys elliptical symmetry, and the theoretical argument used to control $\boldsymbol{\theta}_i$ 's estimation error does not go through. This is precisely why, in the theory, we avoid enforcing exploration on a rectangular domain: each seller's feasible set $P_i$ is axis-aligned and independent of the others, but the design distribution does not need to match the feasible pricing domain. This argument appears in our Remark D.1.
> >
> >
> > ## References
> >
> > [1] Jianqing Fan, Yongyi Guo, and Mengxin Yu. Policy optimization using semiparametric models for dynamic pricing. Journal of the American Statistical Association, 119(545):552–564, 2024.
> >
> > [2] Yiyun Luo, Will Wei Sun, and Yufeng Liu. Distribution-free contextual dynamic pricing. Mathematics of Operations Research, 49(1):599–618, 2024.
> >
> > [3] Fadoua Balabdaoui, Cecile Durot, and Hanna Jankowski. Least squares estimation in the monotone single index model. Bernoulli, 25(4B):3276–3310, 2019.
> >
> > [4] David R Brillinger. A generalized linear model with “gaussian” regressor variables. Selected Works of David Brillinger, pp. 589–606, 2012.

---

> > > ### Comment · Reviewer_apYP · 2025-11-23
> > > **Thanks for your reply**
> > >
> > > Thanks the authors for their careful explanation! I really appreciate the effort you have made to revise and improve the paper. For Question (1), combining to your general response above, I tend to understand the main contribution of this paper as a substantial extension of current feature-based dynamic pricing methods to a broader scope of s-concavity (a generalization of log-concavity). To this extent, I acknowledge that it is reasonable to start with assumptions of knowing $s_i$, and then a learning or searching approach of $s_i$ would be left for future studies. Combining this single point with my previous remark on this paper as a good work, I am happy to increase my score and confidence.
> > >
> > > I would like to also mention, regarding my Question (2), that the optimal estimation rate does not lead to a minimax regret rate. I am glad to see that the authors admitted this. While the regret is still sub-optimal due to the exploration-first phase (like Fan et al. 2024), I look forward to seeing future works converting this estimate method to minimax loss (analog to what Tullii et al. 2024 did on the top of Fan et al. 2024 and Luo et al. 2022). Besides, I am satisfied with the authors’ explanation on the elliptical distribution of explorative prices.
> > >
> > > Reference (in addition to those you've already mentioned):
> > >
> > > Tullii, Matilde, Solenne Gaucher, Nadav Merlis, and Vianney Perchet. "Improved algorithms for contextual dynamic pricing." Advances in Neural Information Processing Systems 37 (2024): 126088-126117.

---

### Official Review · Reviewer_1Gqn · 2025-10-27

**Soundness:** 3
**Presentation:** 3
**Contribution:** 3
**Rating:** 8
**Confidence:** 4

**Summary:**

The paper studies price competition among multiple sellers over a selling horizon of $T$ periods. Each seller’s demand depends on both their own and rivals’ prices. Sellers observe competitors’ prices but not competitors’ realized demand. The paper proposes a dynamic pricing policy that converges to the Nash equilibrium prices.

**Strengths:**

The paper proposes a novel semiparametric pricing policy for nonlinear mean demand. It adopts a more general framework of $s$-concavity.
The paper also establishes an upper bound on the total expected regret and analyze the convergence to the NE.

**Weaknesses:**

Major comments:

1. In Assumption 3.1, the constants $\underline{B}_{\psi_i}$  and

$\overline{B}_{\psi_i}$ are assumed known. How do sellers obtain these values in real-world application? You can also provide some references to justify this assumption.

2. In the exploration phase, price is randomly chosen. Can you provide some practical examples or some references to justify it?

3. In the experiments, when estimating $\phi$, how is $\mathcal{H}_i$ in Equation (11) set?

Minor Comments:
1. Line 59: There is an extra comma after “own price.”
2. Line 75: “Early work Kirman (1975)” should be “Early work (Kirman, 1975).”
3. Line 88: “economics literature Birge et al. (2024); Li et al. (2024)” should be “economics literature (Birge et al., 2024; Li et al., 2024).”
4. Line 96: It seems missing some words between “symmetric p(t)” and “Brillinger (2012)”

**Questions:**

See Weaknesses.

---

> ### Author Response · Authors · 2025-11-23
> **Response to Reviewer 1Gqn**
>
> Thank you for taking the time to review our paper, for the positive feedback and for your helpful comments and suggestions. Please see our responses below.
>
> ## Q1) Knowledge of $[\underline{B}_i, \bar{B}_i]$
>
> We kindly refer the reviewer to our response to Q2 in Reviewer neHp’s report.
>
> ---
>
> ## Q2) exploration - exploitation justification and examples.
>
> Several works using exploration-exploitation frameworks make the same assumption (see, e.g., [1], [2], and [3]). The purpose of randomized pricing during exploration is to guarantee the convergence of the estimators. For instance, [1] shows in Section 3.1, that drawing prices uniformly at random allows one to consistently recover the parameter $\theta$ via least squares; [2] follows a similar strategy. Likewise, [3] demonstrates that using i.i.d. exploration prices from any distribution with non-degenerate variance (see Assumption 1 in their work) is sufficient to obtain a consistent estimator of the model parameters. Randomized exploration is therefore a standard and well-motivated ingredient in dynamic pricing models with learning.
>
> A practical example of this idea is A/B testing, where a firm deliberately randomizes prices to learn the demand response. Without such variation, the platform cannot identify how demand changes with price, leading to biased estimates and suboptimal long-run revenue.
>
> ## References
> [1] Jianqing Fan, Yongyi Guo, and Mengxin Yu. Policy optimization using semiparametric models for dynamic pricing. Journal of the American Statistical Association, 119(545):552–564, 2024.
>
> [2] Yiyun Luo, Will Wei Sun, and Yufeng Liu. Distribution-free contextual dynamic pricing. Mathematics of Operations Research, 49(1):599–618, 2024.
>
> [3] Shukai Li, Cong Shi, and Sanjay Mehrotra. LEGO: Optimal online learning under sequential price competition. Major Revision at Operations Research. Available at SSRN 4803002, 2024
>
> ---
>
> ## Q3) Definition of $\mathcal{H}_i$.
>
> This class consists of all functions $\phi$ that are both monotone non-decreasing and concave. This can be found in lines 361-362 of the updated version.

---

### Official Review · Reviewer_TM1z · 2025-10-31

**Soundness:** 3
**Presentation:** 2
**Contribution:** 3
**Rating:** 4
**Confidence:** 3

**Summary:**

The paper considers a setting where sellers can observe each other’s prices but can only see their own demand. All sellers use the same learning algorithm, and the authors prove that each seller can achieve sublinear regret while the prices collectively converge to a Nash equilibrium. The theoretical results are both novel and solid, though they rely on several somewhat contrived assumptions. The paper uses simulations to validate the findings; it would be even better if the authors compared their approach with other algorithms.

**Strengths:**

The conclusions of this paper are novel and valuable. The study of demand learning under s-concavity extends the frontier of existing research on this topic. Moreover, the proofs are rigorous and nontrivial, making it an excellent theoretical contribution.

**Weaknesses:**

1.	In lines 54–56, it would be helpful to briefly explain how this assumption fundamentally differs from the previous linear model and what additional technical challenges it introduces in the analysis.
2.	Why don’t we restrict $\gamma_i \ge 0$? An increase in others’ prices usually leads to higher demand.
3.	In line 252, why is $\Gamma$ assumed to be Lipschitz? Is this an explicit assumption, or can it be derived from earlier conditions? It would also be helpful to clarify under what circumstances this assumption holds in practice.
4.	Section 3 contains numerous assumptions but lacks illustrative examples. For instance, the authors should provide some common examples of functions that satisfy s-concavity. Listing assumptions without explanation raises concerns about their practical relevance. In reality, demand functions rarely satisfy such properties. It would be valuable to discuss how regret behaves when these assumptions are violated, for example, how regret depends on the parameter $s$.
5.	The term “optimal” in Remark 5.6 is not precise. The paper lacks a lower-bound analysis, which is essential for assessing the tightness of the derived regret order. As mentioned in the experimental section, the observed rate in simulations is faster than the theoretical one, which likely stems from a loose upper bound. In such a case, the parameter can only be considered optimal with respect to the construction of this upper bound, rather than being a generally optimal $\xi$.
6.	The experimental section lacks comparisons with other algorithms. Even if alternative methods, such as those assuming a linear model with possible misspecification, are included, such comparisons would highlight the value of the additional assumptions introduced in this paper.
7.	It would be helpful for the authors to explain why exploration and exploitation are treated separately, rather than adopting optimism-based algorithms (e.g., UCB-type approaches) that integrate the two.

**Questions:**

See above.

---

> ### Author Response · Authors · 2025-11-22
> **Response to Reviewer TM1z**
>
> Thank you for taking the time to review our paper and for your helpful comments and suggestions. Please see our responses below.
>
> ## Q1) Difference with linear demand models.
>
> In the linear model of [1] the link function is assumed to be known, specifically, it is linear $\psi_i(u)=u+\alpha_i$. Therefore, in their setting, only the finite-dimensional parameter $\boldsymbol{\theta}_i$ must be estimated. In our setting, $\psi_i$ is an ***unknown nonlinear link satisfying only monotonicity and $s$-concavity*** (which generalizes the linear demand function that is monotone and log-concave, i.e. $0$-concave). This introduces two main technical challenges: (i) we must estimate $\psi_i$ nonparametrically and establish concentration rates, and (ii) the presence of estimation errors of $\boldsymbol{\theta}_i$ and $\psi_i$ that interact within the estimated best-response operator $\hat{\Gamma}_i$, requires additional control. These issues do not arise in the linear model (or in general in models where $\psi_i$ is known) and motivate the new analysis in our paper.
>
> ## References
>
> [1] Shukai Li, Cong Shi, and Sanjay Mehrotra. LEGO: Optimal online learning under sequential price competition. Major Revision at Operations Research. Available at SSRN 4803002, 2024
>
> ---
>
> ## Q2) The sign of the $\gamma_i$'s.
>
> While it is true that in many applications $\gamma_{ij}>0$ is the natural case, imposing this restriction is not necessary for our algorithm or for any of our theoretical guarantees. All results continue to hold unchanged when $\gamma_{ij}>0$. However, we adopt the more general specification that allows $\gamma_{ij}$ to be arbitrary. This accommodates settings with negative cross-effects, which arise, for example, in vertically differentiated goods, that is, when sellers offer products of different perceived quality. In such cases, increasing a competitor's price may reduce my demand if customers view the competitor's product as higher quality. To reflect this generality, we have added a clarification in the revised version of the paper (see lines 221 - 225 in the revised version).
>
> ---
>
> ## Q3) Lipschitzianity of $\boldsymbol{\Gamma}$.
>
> Thanks for the question. The Lipschitz property of the best-response operator $\boldsymbol{\Gamma}$ is not an assumption; it is derived in Lemma 3.7 in the revised version of the manuscript, and it guarantees the sublinearity of the regret. The contraction property of the best response map is also derived in the linear demand case by [1] for the same reason. The contraction property is derived from our Assumption 3.6 in the revised version of the manuscript, which in practice states that the influence of competitors' prices on seller i's optimal response is sufficiently small relative to the sensitivity to its own price. This condition generalizes the analogous requirement in [1] (Assumption 2 in their work), and admits a clear practical interpretation: a seller's optimal price should not react too strongly to competitors' prices, otherwise the best-response dynamics cannot contract. Under this regime, $\boldsymbol{\Gamma}$ becomes a contraction map and is therefore Lipschitz with constant $L_{\boldsymbol{\Gamma}}<1$. We made this explicit in the updated version after stating Assumption 3.6. We have clarified this in the revised version of the manuscript in lines 312-313.
>
> ## References
> [1] Shukai Li, Cong Shi, and Sanjay Mehrotra. LEGO: Optimal online learning under sequential price competition. Major Revision at Operations Research. Available at SSRN 4803002, 2024

---

> > ### Author Response · Authors · 2025-11-22
> >
> > ## Q4) Examples of s-concave functions and the dependence of regret on s.
> >
> > Thank you for the suggestion. We have added an initial motivation before our assumptions at the beginning of Section 3. Additionally, we have added examples of $s$ concave functions in Appendix L in the updated version. We show examples of CDF that are log-concave ($0$-concave) and $s$-concave for $s<0$. Specifically, we list a large class of log-concave CDFs which includes: Uniform, Gaussian, Exponential, Logistic, Extreme value, Laplace, Gamma, Chi-Square, Chi, Beta, and more. This list was presented by [1], where they also prove that the CDFs of the *Pareto, Lognormal. Student's t, Cauchy* are not log-concave; however, we prove that they are $s$-concave for some $s<0$. We also exhibit a class of functions that are $s$-concave for $s<0$ but not log-concave.
> >
> > We want to specify that ***$s$-concavity is not an artificial condition introduced by our work***. As we point out in lines 103-115 and formalize in Proposition 3.5, many existing results in dynamic pricing already impose the condition $\varphi_i^{\prime}(u) \geq c_i$ for some $c_i>0$. ***This monotonicity condition on the virtual valuation is equivalent to assuming that the demand link $\psi_i$ is $\left(c_i-1\right)$-concave (see Proposition 3.5)***. Our formulation merely makes this structural requirement explicit so that we can exploit the shape-constrained inference literature.
> >
> > For the discussion about "how regret behaves when these assumptions are violated, for example, how regret depends on the parameter $s$" please see our response to Question 1 in Reviewer apYP.
> >
> > ## References
> >
> > [1] Mark Bagnoli and Ted Bergstrom. Log-concave probability and its applications. In Rationality and Equilibrium: A Symposium in Honor of Marcel K. Richter, pp. 217–241. Springer, 2006.
> >
> > ---
> >
> > ## Q5) Clarifying the Meaning of “Optimality”.
> >
> > Thank you for this important observation. We agree that, in the absence of a matching lower bound, the regret upper bound is not necessarily tight. We have clarified the terminology in the revision. Specifically, "optimal" in Remark 4.3 refers only to the value of $\xi$ that minimizes the exponent of $T$ within our derived upper bound, i.e., it is optimal with respect to our proof technique, not in a minimax sense. We made this clarification explicit in Remark 4.3 of the updated version of the manuscript.
> >
> > ---
> >
> > ## Q6) Comparison with other algorithms.
> >
> > Thank you for the suggestion. To the best of our knowledge, no prior algorithm handles sequential price competition with an unknown link function. In this sense, our method is the first to address this general setting, and there is no established baseline operating under comparable assumptions.
> >
> > The model in [1] assumes a known and linear demand link $\psi_i$, whereas our setting allows $\psi_i$ to be unknown and nonlinear. Because our algorithm must estimate $\psi_i$ nonparametrically while [1] treats it as known, a direct comparison would not be meaningful: [1] would be operating under strictly stronger information assumptions and thus would not reflect the difficulty of the setting we study.
> >
> > ## References
> > [1] Shukai Li, Cong Shi, and Sanjay Mehrotra. LEGO: Optimal online learning under sequential price competition. Major Revision at Operations Research. Available at SSRN 4803002, 2024

---

> > > ### Author Response · Authors · 2025-11-22
> > >
> > > ## Q7) Separate treatments for exploration and exploitation.
> > >
> > > Thank you for the comment. As in related semiparametric pricing works (e.g., [1]) or even parametric works (e.g., [2] and [3]), we separate exploration and exploitation because consistent estimation of the model parameters requires i.i.d. price samples. In particular, both the estimation of the index parameter $\boldsymbol{\theta}_i$ and the nonparametric estimation of the link function $\psi_i$ rely on concentration results that hold only when prices are drawn from a fixed distribution during exploration.
> > >
> > > Optimism-based algorithms, such as UCB, interweave exploration and exploitation, but they generate adaptively chosen, highly dependent price sequences. This violates the independence conditions required for the uniform convergence results in our analysis (including the supremum-norm concentration for the s-concave LSE). Extending these guarantees to adaptive designs remains an open and technically challenging problem.
> > >
> > > For these reasons, separating exploration from exploitation is currently the only way to ensure reliable semiparametric estimation under nonlinear, shape-constrained demand.
> > >
> > > ## References
> > >
> > > [1] Jianqing Fan, Yongyi Guo, and Mengxin Yu. Policy optimization using semiparametric models for dynamic pricing. Journal of the American Statistical Association, 119(545):552–564, 2024.
> > >
> > > [2] Yiyun Luo, Will Wei Sun, and Yufeng Liu. Distribution-free contextual dynamic pricing. Mathematics of Operations Research, 49(1):599–618, 2024.
> > >
> > > [3] Shukai Li, Cong Shi, and Sanjay Mehrotra. LEGO: Optimal online learning under sequential price competition. Major Revision at Operations Research. Available at SSRN 4803002, 2024

---

> > > > ### Comment · Reviewer_TM1z · 2025-11-27
> > > >
> > > > Thank you for the additional information. I will maintain my rating.

---

> > > > > ### Author Response · Authors · 2025-11-28
> > > > >
> > > > > We thank the reviewer for reading our responses. Please let us know if any of our clarifications were insufficient or if you have any additional questions we can address. We would be glad to provide further details if that would help the reviewer reevaluate the manuscript.

---

### Official Review · Reviewer_neHp · 2025-11-01

**Soundness:** 3
**Presentation:** 3
**Contribution:** 2
**Rating:** 6
**Confidence:** 4

**Summary:**

This paper studies price competition among multiple sellers with public price information and private demand information. Under monotonicity and s-concave shape constraints on the demand function modelled by the single index model, the authors show that, using their proposed algorithm, the prices converge to NE at $O(T^{-1/7})$ rate and each seller incurs a regret of $O(T^{5/7})$.

The algorithm follows a simple two-step approach, where all the sellers explore for a short time period to gather data for estimation, and then exploit with the estimated prices. The authors also argue that the convergence to NE is closely related to sublinear regrets of individual sellers.

A key observation in the proof is that the monotonicity of the virtual valuation is equivalent to the s-concave property (Proposition 3.5). Furthermore, the s-concavity of mean demand also guarantees the strong concavity of the revenue function.

**Strengths:**

- The conceptual relation of virtual valuation and s-concavity is elegant and useful.
- The demand model is single-index, and its estimation leverages semi-parametric least square regression, which is of theoretical depth.
- The overall writing is good; the presentation of assumptions, results, and algorithms is clear.

**Weaknesses:**

- The motivation of the proposed setting is not particularly strong, as it assumes that all the sellers invoke the same pricing algorithm. This was justified using some factual evidence from the practice at some area in Colorado but broader applicability is not well addressed.
- As mentioned by the authors, the convergence to NE and sublinear regrets are closely related (Line 240-266). Combining with the previous point of all sellers using the same algorithm, it is hard to see how this result differs significantly from the setting without competition, e.g., the prior work by Fan et al. (2024).

**Questions:**

- The algorithm relies on the knowledge of $s_i$. I understand that this limitation is acknowledged in the future direction section. It would strengthen the result significantly if this assumption can be relaxed. Can the algorithm potentially be improved to allow an estimation of $s_i$, or maybe it can be adapted to select from a grid of $s_i$ values (e.g., Lepski method)? More exploration on the possibility/challenge of this issue would be really helpful.
- Similarly, $[\underline{B}_i, \bar{B}_i]$ is required to be known. Can the authors add some brief explanation/justification of this into the main text?
- The authors comment on the tightness of the result (Line 432 - 444) by comparing with related result. Can these argument regarding lower bound be more rigorously stated? Especially given the concavity of the revenue function, one would typically assume a faster rate like $O(T^{-1/2}$ in classical literature. What is the key insight that drives the current regret bound?
- Assumption 5.1 appears unnecessarily abstract. What scenarios does it cover beyond sub-Gaussian?
- A short discussion on how the presented work relates to algorithmic collusion and potential societal impact could be practically meaningful

---

> ### Author Response · Authors · 2025-11-22
> **Response to Reviewer neHp**
>
> Thank you for taking the time to review our paper and for your helpful comments and suggestions. Please see our responses below.
>
> **Answer to Motivation and Monopolistic Setting.** Our framework is not motivated by assuming that sellers must use the same pricing algorithm. While this assumption is indeed standard in the literature (see, e.g., the recent works [1] and [2]), it is not central to the motivation of our setting.
>
> Our primary objective is instead to reformulate the classical assumption that the virtual valuations satisfy $\varphi_i \geq c_i$ for some $c_i>0$, in terms of shape constraints on the demand functions, specifically $s$-concavity of $\psi_i$. This reformulation allows us to design an optimization procedure that is tuning-free, while still preserving no-regret guarantees. Importantly, this perspective extends naturally to unknown, nonlinear demand functions, which, to the best of our knowledge, have not been addressed in previous works.
>
> As for the comparison with Fan et al. (2024), i.e. [3], our results specialize seamlessly to the monopolistic case. In that setting, our assumptions reduce precisely to those used in Fan et al.: Assumption 3.6 in the revised version of the manuscript (which implies the contraction of the best-response operator) is no longer needed, and one only requires the existence of a revenue maximizer. In Fan et al.’s notation, this corresponds to requiring $\varphi_i \geq c_i$ for some $c_i>0$, which is exactly equivalent, in our framework, to imposing $s_i$-concavity of the demand function for some $s_i>-1$.
>
> ## References
>
> [1] Shukai Li, Cong Shi, and Sanjay Mehrotra. LEGO: Optimal online learning under sequential price
> competition. Major Revision at Operations Research. Available at SSRN 4803002, 2024
>
> [2] Goyal, V., Li, S., & Mehrotra, S. (2023). Learning to price under competition for multinomial logit demand. Available at SSRN 4572453.
>
> [3] Jianqing Fan, Yongyi Guo, and Mengxin Yu. Policy optimization using semiparametric models for dynamic pricing. Journal of the American Statistical Association, 119(545):552–564, 2024.
>
> ---
> ---
>
> # Answer to Questions
>
> ## Q1) Knowledge of $s_i$
>
> Thank you for the question. While there are several possible ways to incorporate the estimation of $s_i$ into the algorithm, we have chosen to leave this direction for future work. Estimating $s_i$ --- and, more ambitiously, estimating $(s_i, \psi_i)$ jointly --- is a nontrivial problem and poses significant challenges for the statistical literature in its own right. Nonetheless, potential strategies for estimating $s_i$ include:
>
> 1. **Grid search via MISE**: After the exploration phase, one may evaluate a grid of candidate values (e.g., an interval centered around 0) and, for each candidate $s_i$, compute the mean integrated squared error (MISE) of the corresponding estimator. The value of $s_i$ that minimizes the MISE can then be selected.
>
> 2. **Lepski's method**: As suggested by the reviewer, one could possibly also adopt the Lepski method to select $s_i$ in a data-driven manner over a finite grid of candidate $s$-values. To the best of our knowledge, developing a full theory for such adaptive shape selection would require additional nontrivial results, which remain an interesting open problem for future investigations.
>
> We also emphasize that when $s_i$ is treated as part of the underlying probability model, the algorithm is robust to moderate misspecification of $s_i$. As explained to Reviewer apYP in Question 1, such misspecification does not affect the convergence rates of the regret or the Nash equilibrium, but only modifies certain additive and multiplicative constants. To illustrate this empirically, we include new simulations in Appendix J3 of the revised version.
>
> ---
>
> ## Q2) Knowledge of $[\underline{B}_i, \bar{B}_i]$
>
> Thank you for pointing this out. We now clarify in the revised version of the manuscript (lines 216-219) why the bounds $\left[\underline{B}_i, \bar{B}_i\right]$, which represent lower and upper limits on feasible demand levels for seller $i$, are reasonable to assume known. First, these bounds do not need to be tight; even loose bounds suffice and do not affect either the regret rate or the convergence rate to the NE. Second, such quantities are typically available in practice: firms routinely monitor historical minimum and maximum daily (or hourly) sales volumes, and inventory or capacity constraints naturally impose upper limits on demand. Therefore, reasonable operational bounds can be inferred directly from historical data or standard domain knowledge, even when the functional form of $\psi_i$ is unknown.
>
> ---

---

> > ### Author Response · Authors · 2025-11-22
> >
> > ## Q3) Tightness of the result
> >
> > We thank the reviewer for the question. To clarify, we do not compare our result to any known minimax lower bound-such bounds are not available for semiparametric single-index models in competitive pricing. Instead, we compare our upper bound to the upper bound obtained in the monopolistic setting of [1]. We do not claim that our rate is minimax-optimal.
> >
> > * Regarding the reviewer's intuition about a $T^{-1 / 2}$ type convergence: such rates typically arise under fully parametric models with strong smoothness conditions. For instance, in the fully parametric setting of [2] and in the semiparametric framework of [1] with an infinitely differentiable link function, the authors establish regret bounds of order $\sqrt{T}$. In contrast, our model involves an unknown, nonparametric, shape-constrained link function, and the statistical difficulty is dictated by the uniform estimation rate of $\psi_i$. For concave (or $s$-concave) functions, this rate is known to be $T^{-2 / 5}$ (see, e.g., [3]). This inherent statistical bottleneck-rather than the dynamic nature of the pricing problem-is the main factor determining our regret rate.
> >
> > * The regret exponent $5 / 7$ arises from balancing the exploration cost with the $T^{-2 / 5}$ estimation accuracy of $\psi_i$. As discussed in answer 2 to Reviewer apYP, while we claim that the estimation rate of the non-parametric $s$-concave function is minimax-optimal, the overall regret rate need not be, and we do not conjecture it is minimax-optimal for this feedback model.
> >
> > ## References
> >
> > [1] Jianqing Fan, Yongyi Guo, and Mengxin Yu. Policy optimization using semiparametric models for dynamic pricing. Journal of the American Statistical Association, 119(545):552–564, 2024.
> >
> > [2] Shukai Li, Cong Shi, and Sanjay Mehrotra. LEGO: Optimal online learning under sequential price competition. Major Revision at Operations Research. Available at SSRN 4803002, 2024
> >
> > [3] Wang, Xiao, and Jinglai Shen. "Uniform Convergence and Rate Adaptive Estimation of a Convex Function." arXiv preprint arXiv:1205.0023 (2012).
> >
> > ---
> >
> > ## Q4) Sub-Gaussian prices during exploration
> >
> > We kindly refer the reviewer to our response to Q3 in Reviewer apYP’s report.
> >
> > ## Q5) Algorithminc Collusion
> >
> > Thanks for the suggestion connecting this to algorithmic collusion. Our results indicate that when all sellers employ the same class of learning algorithm, the dynamics do not give rise to collusive behavior: the learning process converges to the classical Nash equilibrium, and prices do not drift upward in a coordinated way. This is beneficial from a consumer perspective, as it prevents the emergence of tacit price increases. However, if some sellers deviate and adopt different algorithms or learning rules, then the interaction dynamics may create conditions under which algorithmic collusion can emerge.
> >
> > We are happy to include a more detailed discussion of this phenomenon in the final version, where space constraints will be less restrictive.

---

> > > ### Comment · Reviewer_neHp · 2025-11-27
> > > **Thank you for the response and score updated**
> > >
> > > I thank the authors for their detailed and thoughtful response. I am more convinced by their motivation/positioning and tightness of result after the rebuttal. My take is that this paper has nice theoretical contribution and is quite relevant as algorithmic pricing is increasingly prevalent in modern retailing businesses, especially with the rise of agentic applications in businesses. Considering all these, I have thus increased my score from 6 to 8.

---

### Author Response · Authors · 2025-11-22
**Thanking the Reviewers and highlighting the contribution about s-concavity**

We first want to thank the reviewers for taking the time to review our work and for their insightful questions. We have addressed all the points raised and incorporated the corresponding changes into the revised version of the manuscript. Here, we would also like to highlight what we see as a major high-level contribution of our work: connecting the $s$ concavity literature to a concrete economic application.

The notion of $s$-concavity has a long history in analysis and probability, originating in work on convex measures and generalized concavity (see, e.g., [1], [2], [3], [4]). These papers (and many that followed) investigate the structural and probabilistic properties of $s$-concave functions and measures, often emphasizing how $s$-concavity naturally extends log-concavity (the special case $s=0$). However, despite this rich theoretical development, the concept of $s$-concavity has found relatively few tangible applications in economic modeling.

In this paper, we identify an interesting connection that bridges this theory with the dynamic pricing and revenue-management literature. Specifically, we relate the common assumption on the valuation (or virtual valuation) function's derivative,

$$
\varphi_i^{\prime}(u) \geq c_i \quad \text { for some } c_i>0,
$$

used in, e.g., [5], [6], [7], [8], [9], [10] where log-concavity of $\psi_i$ and $1-\psi_i$ is often imposed (implying $c_i \geq 1$). Our key observation is that:

$$
\varphi_i^{\prime}(u) \geq c_i \quad \Longleftrightarrow \quad \psi_i \text { is }\left(c_i-1\right) \text {-concave },
$$

where $\psi_i$ is the the demand function, with log-concavity recovered as the special case $c_i=1$. This connection has two important consequences. First, it reframes a standard "regularity" assumption in dynamic pricing (a lower bound on $\varphi_i^{\prime}$) as a shape constraint on demand. Second, it enables a fully datadriven, tuning-free estimation strategy: we estimate $\psi_i$ via nonparametric least squares under $s$ concavity. In doing so, we preserve no-regret guarantees while allowing for nonlinear, unknown demand functions, thereby extending the scope of prior work beyond the log-concave (or fully parametric) setting.

# Refrences

[1] Borell, Christer. "Convex measures on locally convex spaces." Arkiv för matematik 12.1 (1974): 239-252.

[2] Brascamp, Herm Jan, and Elliott H. Lieb. "On extensions of the Brunn-Minkowski and Prékopa-Leindler theorems, including inequalities for log concave functions, and with an application to the diffusion equation." Journal of functional analysis 22.4 (1976): 366-389.

[3] Prékopa, András. "On logarithmic concave measures and functions." Acta Scientiarum Mathematicarum 34 (1973): 335-343.

[4] Dharmadhikari, Sudhakar, and Kumar Joag-Dev. Unimodality, convexity, and applications. Elsevier, 1988.

[5] Jianqing Fan, Yongyi Guo, and Mengxin Yu. Policy optimization using semiparametric models for dynamic pricing. Journal of the American Statistical Association, 119(545):552–564, 2024.

[6] Yiwei Chen and Vivek F Farias. Robust dynamic pricing with strategic customers. Mathematics of Operations Research, 43(4):1119–1142, 2018.

[7] Richard Cole and Tim Roughgarden. The sample complexity of revenue maximization. In Proceedings of the forty-sixth annual ACM symposium on Theory of computing, pp. 243–252, 2014.

[8] Negin Golrezaei, Adel Javanmard, and Vahab Mirrokni. Dynamic incentive-aware learning: Robust pricing in contextual auctions. Advances in Neural Information Processing Systems, 32, 2019.

[9] Adel Javanmard. Perishability of data: dynamic pricing under varying-coefficient models. Journal of Machine Learning Research, 18(53):1–31, 2017.

[10] Adel Javanmard and Hamid Nazerzadeh. Dynamic pricing in high-dimensions. Journal of Machine Learning Research, 20(9):1–49, 2019.

---

### Author Response · Authors · 2025-12-04
**Summary note for the newly assigned Area Chair**

Dear Area Chair,

We are writing regarding our paper submission in light of the recent OpenReview incident and the subsequent reassignment of area chairs. Given the complications created by the leak, our aim is to provide a concise, factual summary of the post-rebuttal reviewer updates, ensuring that you have the full context when determining your independent assessment.

After we posted our detailed responses and revisions on November 22, two reviewers explicitly indicated that our clarifications addressed their main concerns and stated that they were raising their scores:

* **Reviewer neHp** wrote a follow-up comment on November 27, saying "I am more convinced by their motivation/positioning and tightness of result after the rebuttal", and they were reaffirming the theoretical contribution and relevance of the paper. They concluded by stating: "Considering all these, I have thus increased my score from 6 to 8".

* **Reviewer apYP** posted a follow-up comment on November 22, thanking us for the "careful explanation" and revision, noting that they now "understand the main contribution of this paper as a substantial extension of current feature-based dynamic pricing methods to a broader scope of s-concavity" and indicating that, in light of this, they were increasing both their score and their confidence.

For the other two reviewers:

* **Reviewer 1Gqn** already had a positive view of the paper (rating 8) in the original review and did not post an additional comment after our rebuttal.

* **Reviewer TM1z** posted a brief follow-up acknowledging our additional explanations and indicated that they would maintain their original rating. We then replied to invite them to let us know if any of our clarifications were insufficient and to offer further details should that help them reconsider their assessment of the manuscript.

Thank you very much for your time and effort in handling the review process.

Best,

The Authors

---

### Meta-Review · Area_Chair_r4dR · 2026-01-05

**Summary:**

The reviewers raised several initial concerns regarding this paper: (1) insufficient motivation for the problem setting, (2) lack of clarity on technical novelty and challenges, (3) unrealistic or unjustified assumptions, and (4) uncertainty regarding whether the minimax bound is achieved. In the rebuttal, the authors addressed all these points, providing particularly persuasive arguments for the motivation and the validity of their assumptions. While the minimax bound remains unproven, the authors clarified this as an open problem, which was accepted by the reviewers. Although some doubts regarding technical non-triviality persist, the overall quality and clarity of the work have been significantly improved.

**Reviewer Concerns:**

Addressed by Rebuttal:
1. Motivation and Assumptions: The authors provided high-quality justifications and convincing arguments, effectively resolving these concerns.
2. Minimax Bound: By explicitly framing the minimax bound as an open problem in the revised manuscript, the authors reached a mutual understanding with the reviewers.
3. Technical Novelty: The discussion of technical non-triviality was strengthened, leading several reviewers to reconsider the depth of the contribution.

Outstanding Concerns:

 One reviewer (TM1z) remains skeptical about the technical non-triviality of the results, feeling that the "challenge" of the analysis has not been fully demonstrated.

**Reviewer Scores:**

If a full discussion period had occurred, the score distribution would likely have shifted positively:

- Reviewers neHP and apYP: Both expressed a clear intention to upgrade their scores following the persuasive rebuttal concerning the motivation and the justification of assumptions.

- Reviewer 1Gqn: This reviewer was already positive and would have maintained their high assessment.

- Reviewer TM1z: Due to persistent doubts regarding the technical non-triviality, this reviewer would likely not have changed their score.

Conclusion: Overall, the majority of the reviewers hold a positive view of the paper. Since the most critical concerns have been addressed and there are no strong dissenting opinions (no "reject" or "strong reject"), I recommend the paper for Acceptance.

---

### Decision · Program_Chairs · 2026-01-26

Accept (Poster)